# Random Features Outperform Linear Models: Effect of Strong Input-Label Correlation in Spiked Covariance Data

## Abstract

Random Feature Model (RFM) with a nonlinear activation function is instrumental in understanding training and generalization performance in high-dimensional learning. While existing research has established an asymptotic equivalence in performance between the RFM and noisy linear models under isotropic data assumptions, empirical observations indicate that the RFM frequently surpasses linear models in practical applications. To address this gap, we ask, *"When and how does the RFM outperform linear models?"* In practice, inputs often have additional structures that significantly influence learning. Therefore, we explore the RFM under anisotropic input data characterized by spiked covariance in the proportional asymptotic limit, where dimensions diverge jointly while maintaining finite ratios. Our analysis reveals that a high correlation between inputs and labels is a critical factor enabling the RFM to outperform linear models. Moreover, we show that the RFM performs equivalent to noisy polynomial models, where the polynomial degree depends on the strength of the correlation between inputs and labels. Our numerical simulations validate these theoretical insights, confirming the performance-wise superiority of RFM in scenarios characterized by strong input-label correlation.

## 1 Introduction

Random Feature Model (RFM) (Rahimi & Recht, 2007) has received significant attention in recent years due to its compelling theoretical properties and its connections to neural networks. In this work, we focus on a typical supervised learning framework utilizing the RFM, expressed as

$$\boldsymbol{\omega}^T \sigma(\mathbf{F}\mathbf{x}), \tag{1}$$

where $\mathbf{x} \in \mathbb{R}^n$ is an input vector, $\mathbf{F} \in \mathbb{R}^{k \times n}$ is called the random feature matrix, $\sigma : \mathbb{R} \to \mathbb{R}$ is an activation function, and $\boldsymbol{\omega} \in \mathbb{R}^k$ is a vector of learnable parameters. The feature matrix $\mathbf{F}$ is randomly sampled and remains fixed, allowing the RFM to be interpreted as a two-layer neural network with fixed weights in the first layer. The learning process involves optimizing the following objective function based on a set of training samples $\{(\mathbf{x}_i, y_i)\}_{i=1}^m$

$$\hat{\boldsymbol{\omega}}_\sigma := \underset{\boldsymbol{\omega} \in \mathbb{R}^k}{\operatorname{argmin}} \frac{1}{m} \sum_{i=1}^m (y_i - \boldsymbol{\omega}^T \sigma(\mathbf{F}\mathbf{x}_i))^2 + \lambda ||\boldsymbol{\omega}||_2^2, \tag{2}$$

where $\lambda \in \mathbb{R}^+$ is a regularization constant. While the RFM can be studied under various loss functions, we concentrate on squared loss for simplicity in our theoretical development. The training error of the RFM with activation function $\sigma$ is defined as

$$\mathcal{T}_\sigma := \frac{1}{m} \sum_{i=1}^m (y_i - \hat{\boldsymbol{\omega}}_\sigma^T \sigma(\mathbf{F}\mathbf{x}_i))^2 + \lambda ||\hat{\boldsymbol{\omega}}_\sigma||_2^2. \tag{3}$$

Similarly, one can measure the performance of a trained RFM via the generalization error defined as

$$\mathcal{G}_\sigma := \underset{(\mathbf{x}, y)}{\mathbb{E}} (y - \hat{\boldsymbol{\omega}}_\sigma^T \sigma(\mathbf{F}\mathbf{x}))^2. \tag{4}$$

We study the training and generalization errors of RFM (1) in the "proportional asymptotic limit," where the number of training samples $m$, the input dimension $n$, and the number of intermediate features $k$ jointly diverge: $m, n, k \to \infty$ with $n/m, n/k \in (0, \infty)$. Under an isotropic data assumption, the RFM has been shown to perform equivalent to the following noisy linear model (Montanari et al., 2019; Gerace et al., 2020; Goldt et al., 2022; Mei & Montanari, 2022; Hu & Lu, 2023):

$$\boldsymbol{\omega}^T(\mu_0 \mathbf{1} + \mu_1 \mathbf{F}\mathbf{x} + \mu_* \mathbf{z}), \tag{5}$$

where $\mathbf{1}$ is an all-one vector, $\mathbf{z} \sim \mathcal{N}(0, \mathbf{I}_k)$, and $\mu_0, \mu_1, \mu_*$ are constants depending on the activation function $\sigma$ as $\mu_0 = \mathbb{E}[\sigma(z)]$, $\mu_1 = \mathbb{E}[z\sigma(z)]$, and $\mu_* = \sqrt{\mathbb{E}[\sigma(z)^2] - \mu_1^2 - \mu_0^2}$ where $z \sim \mathcal{N}(0, 1)$. The noisy linear model enables the performance characterization of the RFM (Dhifallah & Lu, 2020). It is also useful for studying double-descent phenomenon (Belkin et al., 2019; Geiger et al., 2020) with regards to loss functions (Mei & Montanari, 2022), activation functions (Wang & Bento, 2022; Demir & Doğan, 2023), and regularization constant (Nakkiran et al., 2021). However, such a noisy linear model falls short in explaining the empirical performance of the RFM on real-world data (Ghorbani et al., 2020). This motivates us to investigate the following question:

*"Under which conditions does the random feature model outperform linear models?"*

The equivalence of the RFM to the noisy linear model is closely tied to the isotropic data assumption, i.e., $\mathbf{x} \sim \mathcal{N}(0, \mathbf{I}_n)$. However, in practice, inputs often exhibit additional structures (Facco et al., 2017). To address this limitation, we consider spiked covariance data model (Johnstone, 2001; Baik et al., 2005)

$$\mathbf{x} \sim \mathcal{N}(0, \mathbf{I}_n + \theta \boldsymbol{\gamma}\boldsymbol{\gamma}^T), \quad y := \sigma_* \left( \frac{\boldsymbol{\xi}^T \mathbf{x}}{\sqrt{1 + \theta \alpha^2}} \right), \tag{6}$$

which introduces anisotropic characteristics due to the structured covariance of $\mathbf{x}$. Here, $\boldsymbol{\gamma} \in \mathbb{R}^n$ serves as a fixed but unknown "spike signal" with $||\boldsymbol{\gamma}||_2 = 1$, $\boldsymbol{\xi} \in \mathbb{R}^n$ represents a fixed but unknown "label signal" also normalized to one, i.e., $||\boldsymbol{\xi}||_2 = 1$. The parameter $\theta$, termed "spike magnitude" scales with the dimension as $\theta \asymp n^\beta$ where $\beta \in [0, 1/2)$, and $\sigma_* : \mathbb{R} \to \mathbb{R}$ is a nonlinear target function. Note that the input of the target function $\sigma_*$ is scaled to have unit variance. Additionally, we define the alignment parameter $\alpha := \boldsymbol{\gamma}^T \boldsymbol{\xi}$, which governs the input-label correlation.

Under the spiked data model, we first establish a "universality" theorem indicating that the RFM exhibits equivalent performance across two different activation functions if the first two statistical moments of $(\sigma(\mathbf{F}\mathbf{x}), y)$ are consistent for both activations. Leveraging this theorem, we extend our analysis to demonstrate that high-order polynomial models perform equivalent to the RFM beyond traditional linear frameworks (5). Notably, we show that the equivalence to the noisy linear model still holds provided that at least one of the following is sufficiently small: (i) the cosine similarity between the rows of $\mathbf{F}$ and the $\boldsymbol{\gamma}$, (ii) the spike magnitude $\theta$, and (iii) the alignment parameter $\alpha$. A precise characterization of these conditions will be detailed in Section 4. Moreover, we introduce a new equivalence between the RFM and noisy polynomial models, generalizing previous results on noisy linear models (5). Finally, this framework allows us to compare different activation functions within the context of spiked data, highlighting how nonlinearity can enhance learning performance beyond linear models.

Overall, our contributions are as follows:

1. We show the conditions under which the RFM outperforms linear models under anisotropic data settings, which indicates a key role played by the input-label correlation in data.

2. We establish a theorem that shows high-order polynomial models perform equivalent to the RFM under spiked covariance data assumption.

3. We relax the isotropic data assumption previously utilized by Hu & Lu (2023) for the "universality of random features," which may be of particular interest on its own.

By addressing these aspects, our work not only clarifies when and how RFMs can excel in the spiked covariance model setting but also enriches the theoretical understanding of their performance relative to linear models.

## 2 RELATED WORK

The RFM has emerged as a vital tool in the realm of machine learning, initially introduced as a randomized approximation to kernel methods (Rahimi & Recht, 2007). Its significance has grown in parallel with the development of neural networks, particularly in how both the RFM and the Neural Tangent Kernel (NTK) offer distinct linear approximations of two-layer neural networks (Ghorbani et al., 2021). While empirical evidence shows that neural networks often outperform both the RFM and NTK (Ghorbani et al., 2020), analyzing the RFM provides critical insights into the behavior of nonlinear models, serving as a bridge toward understanding more complex architectures.

A substantial body of research has focused on the equivalence between the RFM and noisy linear models with regards to their performances (Montanari et al., 2019; Gerace et al., 2020; Goldt et al., 2020; 2022; Dhifallah & Lu, 2020; Mei & Montanari, 2022). For instance, Hu & Lu (2023) demonstrated this equivalence in terms of training and generalization performance using Lindeberg's method. Additionally, Montanari & Saeed (2022) showed that nonlinear features could be substituted with equivalent Gaussian features in empirical risk minimization settings. However, these studies mainly examined the RFM under isotropic data assumptions, which limits their applicability to real-world scenarios characterized by more complex data structures. Our work extends these findings by investigating the RFM under anisotropic data conditions, thereby addressing a significant gap in the literature. Recent works have indicated that equivalent Gaussian features may not consistently provide equivalent training and generalization errors for nonlinear models (Gerace et al., 2022; Pesce et al., 2023). This discrepancy has led researchers to explore alternative feature representations, including Gaussian mixtures (Dandi et al., 2023b). However, our study diverges from this line of inquiry by concentrating on identifying equivalent activation functions rather than seeking alternatives to Gaussian features.

In light of the RFM's limitations in surpassing linear models, recent research has turned to neural networks trained with one gradient step as a potential solution to bridge this performance gap (Ba et al., 2022; Damian et al., 2022). Studies by Ba et al. (2023) and Mousavi-Hosseini et al. (2023) have explored one-step gradient descent techniques in relation to sample complexity under spiked covariance data assumption. Furthermore, Moniri et al. (2023) explored neural networks trained with one gradient step in an isotropic data setting and showed that the update can be characterized by the appearance of a spike in the spectrum of the feature matrix. Dandi et al. (2023a); Cui et al. (2024) studied the one-step gradient descent with a higher learning rate in comparison to Moniri et al. (2023). Ba et al. (2022); Moniri et al. (2023); Dandi et al. (2023a); Cui et al. (2024) also stated Gaussian equivalence results in terms of training and generalization errors when studying the one gradient step technique under isotropic Gaussian data. Note that although the aforementioned works considered feature learning in two-layer neural networks, feature learning for three-layer neural networks is shown to have better sample complexities than the two-layer case (Nichani et al., 2023; Wang et al., 2024). While these works primarily focus on surpassing the performance of linear models through feature learning in neural networks, our research aims to clarify when and how the RFM can outperform linear models without relying on feature learning.

Additionally, there is an ongoing exploration into deep random features that integrate multiple layers of random projections interleaved with nonlinearities (Schröder et al., 2023; Bosch et al., 2023). Research by Zavatone-Veth & Pehlevan (2023) has also examined deep random features with anisotropic weight matrices using statistical physics techniques. Moreover, studies have investigated the RFM under polynomial scaling limits where $k, n, m \to \infty$ while maintaining specific ratios between these parameters, i.e., $k/n^{c_1}, m/k^{c_2} \in (0, \infty)$ for some constants $c_1, c_2 > 0$ (Hu et al., 2024), further providing equivalence results for the RFM under the polynomial scaling limit.

## 3 PRELIMINARIES

**Notations** We denote vectors and matrices using bold letters, with lowercase letters representing vectors and uppercase letters representing matrices. Scalars are indicated by non-bold letters. The notation $||\mathbf{x}||_2$ refers to the Euclidean norm of the vector $\mathbf{x}$ while $||\mathbf{F}||$ denotes to spectral norm of the matrix $\mathbf{F}$. We use the term polylog $k$ to represent any polylogarithmic function of $k$. The symbol $\xrightarrow{\mathbb{P}}$ indicates convergence in probability. We employ big-O notation, denoted $\mathcal{O}(.)$, and

small-o notation, represented as $o(.)$ with respect to the parameters $k, n, m$. These notations are

$$h(k) = \mathcal{O}(g(k)) \quad \Leftrightarrow \quad \limsup_{k \to \infty} \frac{h(k)}{g(k)} < \infty, \tag{7}$$

$$h(k) = o(g(k)) \quad \Leftrightarrow \quad \lim_{k \to \infty} \frac{h(k)}{g(k)} = 0. \tag{8}$$

It is important to note that $h(k) = o(1/\text{polylog } k)$ implies that $\lim_{k \to \infty} h(k) \text{ polylog } k = 0$ for all polylogarithmic functions. We also use the notation $\theta \asymp n^\beta$ to indicate that there exists some constants $C_1, C_2 > 0$ such that $C_1 n^\beta \le \theta \le C_2 n^\beta$. Finally, we denote the $i$-th Hadamard power of a vector $\mathbf{x}$ as $\mathbf{x}^{\circ i}$.

**Assumptions**  We establish our results based on the following assumptions.

(A.1) The spike and label signal vectors $\boldsymbol{\gamma}, \boldsymbol{\xi} \in \mathbb{R}^n$ are deterministic and $||\boldsymbol{\gamma}||_2 = ||\boldsymbol{\xi}||_2 = 1$.

(A.2) $\mathbf{x} \sim \mathcal{N}(0, \mathbf{I}_n + \theta \boldsymbol{\gamma} \boldsymbol{\gamma}^T)$ for spike magnitude $\theta \asymp n^\beta$ where $\beta \in [0, 1/2)$.

(A.3) $y := \sigma_* \left( \frac{\boldsymbol{\xi}^T \mathbf{x}}{\sqrt{1 + \theta \alpha^2}} \right)$ where $\sigma_* : \mathbb{R} \to \mathbb{R}$ is a function satisfying $|\sigma_*(x)| < C(1 + |x|^K)$ for all $x \in \mathbb{R}$ and some constants $C > 0$, $K \in \mathbb{Z}^+$.

(A.4) The number of training samples $m$, dimension of input vector $n$ and number of intermediate features $k$ jointly diverge: $m, n, k \to \infty$ with $n/m, n/k \in (0, \infty)$.

(A.5) $\mathbf{F} := [\mathbf{f}_1, \mathbf{f}_2, \ldots, \mathbf{f}_k]^T$ where $\mathbf{f}_i \sim \mathcal{N}(0, \frac{1}{n+\theta} \mathbf{I}_n)$. Note that the covariance of $\mathbf{f}_i$ is selected such that $\mathbb{E}_{\mathbf{x}, \mathbf{f}_i}[(\mathbf{f}_i^T \mathbf{x})^2] = 1$.

(A.6) The activation function $\sigma : \mathbb{R} \to \mathbb{R}$ is an odd function satisfying $|\sigma(x)| < C(1 + |x|^K)$ for all $x \in \mathbb{R}$ and some constants $C > 0$, $K \in \mathbb{Z}^+$.

**Discussion of Assumptions**  The assumptions outlined in (A.1)-(A.3) establish the foundational framework for our data model and setup. Specifically, (A.1) describes the general characteristics of the spike and label signals, while (A.2) introduces the spike magnitude $\theta$. Although there may be potential to extend the range of $\theta$ in future work, our current proofs necessitate that $\beta < 1/2$. This restriction is crucial for ensuring that the mathematical properties we derive remain valid within the specified limits. In (A.3), we scale the input of the target function $\sigma_*$ to get unit variance. This scaling not only simplifies our proofs but also aligns with standard practices in theoretical analysis. Assumption (A.4) specifies the proportional asymptotic limit, which is a critical aspect of our study. By requiring that the dimensions jointly diverge while maintaining finite ratios, we can effectively analyze the behavior of the RFM in high-dimensional settings. In (A.5), we detail the distribution of the feature matrix $\mathbf{F}$, emphasizing that the $1/(n + \theta)$ scaling is vital for our results to hold. This specific scaling ensures that the covariance structure of $\mathbf{F}$ is appropriately calibrated with respect to both the input data and the spike magnitude, facilitating accurate performance characterization. Assumption (A.6) pertains to the properties of the activation function $\sigma$, which warrants further elaboration. Previous research by Hu & Lu (2023) established a performance-wise equivalence between the RFM and noisy linear models under isotropic data conditions, relying on the premise that $\sigma$ is an odd function with bounded first, second, and third derivatives. In our case, (A.6) implies the derivatives of $\sigma(x)$ bounded by polylog $k$ if $|x| \le$ polylog $k$, which happens with high probability for $x \sim \mathcal{N}(0, 1)$. Finally, it is noteworthy that while ReLU (9) does not conform to the odd function assumption stipulated in (A.6), empirical evidence suggests that our findings remain valid even when using ReLU as an activation function.

**Activation Functions for Numerical Results**  In our numerical simulations, we employ three widely used activation functions: ReLU (Rectified Linear Unit), tanh (hyperbolic tangent), and Softplus. The mathematical definitions of these activation functions are as follows:

$$\sigma_{ReLU}(x) := \max(0, x), \quad \sigma_{tanh}(x) := \frac{e^{2x} - 1}{e^{2x} + 1}, \quad \sigma_{Softplus} := \log(1 + e^x). \tag{9}$$

## 4 MAIN RESULTS

In this section, we describe our main theoretical results, significantly enhancing the understanding of RFM under structured data conditions. In Section 4.1, we introduce a new "universality" theorem that establishes the performance-wise equivalence of the RFM when utilizing two distinct activation functions, $\sigma$ and $\hat{\sigma}$. This theorem asserts that if the first two statistical moments of the pairs $(\sigma(\mathbf{F}\mathbf{x}), y)$ and $(\hat{\sigma}(\mathbf{F}\mathbf{x}), y)$ are matched, the training and generalization performance of the RFM will remain equivalent across both functions. Building on this foundation, Section 4.2 explores the scenario where $\hat{\sigma}$ is represented as a finite-degree Hermite expansion of $\sigma$, demonstrating that the RFM retains equivalent performance regardless of the chosen activation function. Notably, we reveal that the degree of this expansion is intricately tied to the input-label correlation, offering insights into how data structure influences model performance. In Section 4.3, we establish conditions for the equivalence between the RFM and a noisy linear model, providing a crucial corollary to our earlier findings. When these conditions are not satisfied, we identify that a high-order polynomial model serves as an equivalent to the RFM, a topic further elaborated in Section 4.4.

### 4.1 UNIVERSALITY OF RANDOM FEATURES UNDER SPIKED DATA

Here, we extend the universality laws established by Hu & Lu (2023) for RFM to encompass spiked covariance data. The concept of "universality of random features" implies that one can replace $\sigma(\mathbf{F}\mathbf{x})$ in the problem with an equivalent Gaussian vector, as articulated in equation (5), without compromising the training and generalization performance of the model. While Hu & Lu (2023) demonstrated this result under mild assumptions regarding the feature matrix $\mathbf{F}$, the nonlinearity $\sigma$, and the loss function, their findings were constrained by an isotropic data assumption. To address these limitations, we propose a new universality theorem tailored for scenarios involving spiked covariance data, as outlined in equation (6).

**Theorem 1.** *Let $\sigma(x)$, $\hat{\sigma}(x)$ be two activation functions satisfying (A.6). Suppose that assumptions (A.1)-(A.6) given in Section 3 hold. If*

$$\left\| \mathbb{E}_{\mathbf{x}}[\sigma(\mathbf{F}\mathbf{x})\sigma(\mathbf{F}\mathbf{x})^T] - \mathbb{E}_{\mathbf{x}}[\hat{\sigma}(\mathbf{F}\mathbf{x})\hat{\sigma}(\mathbf{F}\mathbf{x})^T] \right\| = o\left(1/polylog\,k\right), \tag{10}$$

$$\left\| \mathbb{E}_{(\mathbf{x},y)}[\sigma(\mathbf{F}\mathbf{x})y] - \mathbb{E}_{(\mathbf{x},y)}[\hat{\sigma}(\mathbf{F}\mathbf{x})y] \right\| = o\left(1/polylog\,k\right), \tag{11}$$

*are satisfied, then*

  (i) *the training error $\mathcal{T}_\sigma$ of the RFM with activation $\sigma$, and the training error $\mathcal{T}_{\hat{\sigma}}$ of the RFM with activation $\hat{\sigma}$, both converge in probability to the same value $e_\mathcal{T} \geq 0$,*

  (ii) *the corresponding generalization errors $\mathcal{G}_\sigma$ and $\mathcal{G}_{\hat{\sigma}}$ also converge in probability to the same value $e_\mathcal{G} \geq 0$ under additional assumptions (A.7) and (A.8) provided in Appendix A.*

*Proof.* We provide a proof in Appendix A.

**Proof Approach** We follow the proof technique used by Hu & Lu (2023). First, we define a perturbed training objective that includes additional terms related to the generalization error. Then, we bound the expected difference in the perturbed training objective resulting from replacing $\sigma(\mathbf{F}\mathbf{x}_i)$ with $\hat{\sigma}(\mathbf{F}\mathbf{x}_i)$ for each training sample $\mathbf{x}_i$, following the principles of Lindeberg's method (Lindeberg, 1922; Korada & Montanari, 2011). □

A key advancement of our work is encapsulated in Theorem 1, which establishes the equivalence of the RFM across two different activation functions. This result is notably more general than previous findings that focused solely on the equivalence of specific models. Specifically, Theorem 1 asserts that substituting the activation function $\sigma$ with $\hat{\sigma}$ does not impact the training and generalization performance of the RFM, provided that conditions (10) and (11) are satisfied. In the subsequent sections, we will leverage this theorem to identify models that are equivalent to the RFM and outline the conditions necessary for these equivalences to hold, thereby broadening the applicability of our findings in various contexts.

## 4.2 Noisy Polynomial Models Equivalent to the Random Feature Model

Now, we consider generalizing the equivalent noisy linear model to equivalent high-order polynomial models by utilizing Theorem 1. To achieve this, we leverage Hermite polynomials (O'Donnell, 2014, Chapter 11.2). For $i \in \mathbb{Z}^+$, $i$-th Hermite polynomial is defined as

$$H_i(x) := (-1)^i e^{x^2/2} \frac{d^i}{dx^i} e^{-x^2/2} \tag{12}$$

for any $x \in \mathbb{R}$. There are two important properties of Hermite polynomials. First, they are orthogonal with respect to Gaussian measure, so $\mathbb{E}_{z \sim \mathcal{N}(0,1)}[H_i(z)H_j(z)] = 0$ for $i \neq j$. Second, they form an orthogonal basis of the Hilbert space of functions $\sigma : \mathbb{R} \to \mathbb{R}$ satisfying $\mathbb{E}_{z \sim \mathcal{N}(0,1)}[\sigma(z)^2] < \infty$. This means that the function $\sigma(x)$ can be expressed as an infinite sum of Hermite polynomials as

$$\sigma(x) = \sum_{i=0}^{\infty} \frac{1}{i!} \mu_i H_i(x), \quad \text{with} \quad \mu_i := \mathbb{E}_{z \sim \mathcal{N}(0,1)}[H_i(z)\sigma(z)]. \tag{13}$$

This sum is also known as "Hermite expansion" and $\mu_i$ is called $i$-th Hermite coefficient. Since we are working with Gaussian random variables, Hermite expansion simplifies the analysis. Indeed, we can construct a performance-wise equivalent activation function using the Hermite expansion. Specifically, the following theorem states that the RFM with activation function $\sigma(x)$ performs equivalent to the RFM with $\hat{\sigma}_l(x)$, which is a finite-degree Hermite expansion with additional noise.

**Theorem 2.** *Let $\sigma$ be any activation function satisfying (A.6). Define another activation function*

$$\hat{\sigma}_l(x) := \left( \sum_{j=0}^{l-1} \frac{1}{j!} \mu_j H_j(x) \right) + \mu_l^* z \quad with \quad z \sim \mathcal{N}(0,1), \tag{14}$$

*where $\mu_j := \mathbb{E}_{z \sim \mathcal{N}(0,1)}[H_j(z)\sigma(z)]$ and $\mu_l^* := \sqrt{\mathbb{E}_{z \sim \mathcal{N}(0,1)}[\sigma(z)^2] - \sum_{j=0}^{l-1} \mu_j^2/(j!)}$.*

*Suppose that assumptions (A.1)-(A.6) given in Section 3 hold. If*

$$\eta := \max_{1 \leq i \leq k} \left| \frac{(\boldsymbol{\xi} + \theta\alpha\boldsymbol{\gamma})^T \mathbf{f}_i}{\sqrt{1 + \theta\alpha^2}} \right| \leq \frac{C}{n^{1/l}}, \quad for some C > 0 and some l \in \mathbb{Z}^+, \tag{15}$$

*is satisfied, then*

> *(i) the training error $\mathcal{T}_\sigma$ of the RFM with activation $\sigma$, and the training error $\mathcal{T}_{\hat{\sigma}_l}$ of the RFM with activation $\hat{\sigma}_l$, both converge in probability to the same value $e_\mathcal{T} \geq 0$,*
>
> *(ii) the corresponding generalization errors $\mathcal{G}_\sigma$ and $\mathcal{G}_{\hat{\sigma}_l}$ also converge in probability to the same value $e_\mathcal{G} \geq 0$ under additional assumptions (A.7) and (A.8) provided in Appendix A.*

*Proof.* In Appendix C, we show

$$\left\| \mathbb{E}_{\mathbf{x}}[\sigma(\mathbf{Fx})\sigma(\mathbf{Fx})^T] - (\mu_1^2 \mathbf{F}(\mathbf{I}_n + \theta\boldsymbol{\gamma}\boldsymbol{\gamma}^T)\mathbf{F}^T + \mu_*^2 \mathbf{I}_k) \right\| = o\left(1/\text{polylog}\, k\right), \tag{16}$$

for any $\sigma$ satisfying (A.6), $\mu_1 = \mathbb{E}[z\sigma(z)]$ and $\mu_* = \sqrt{\mathbb{E}[\sigma(z)^2] - \mu_1^2}$ with $z \sim \mathcal{N}(0,1)$. By definition, we have $\mathbb{E}[z\sigma(z)] = \mathbb{E}[z\hat{\sigma}_l(z)]$. One can easily show $\mathbb{E}[\sigma(z)^2] = \mathbb{E}[\hat{\sigma}_l(z)^2]$. Therefore, (16) holds for $\sigma$ and $\hat{\sigma}_l$ with same $\mu_1$ and $\mu_*$ values. Using (16) and triangle inequality, we get

$$\left\| \mathbb{E}_{\mathbf{x}}[\sigma(\mathbf{Fx})\sigma(\mathbf{Fx})^T] - \mathbb{E}_{\mathbf{x}}[\hat{\sigma}_l(\mathbf{Fx})\hat{\sigma}_l(\mathbf{Fx})^T] \right\| = o\left(1/\text{polylog}\, k\right). \tag{17}$$

Similarly, in Appendix D, we prove

$$\left\| \mathbb{E}_{(\mathbf{x},y)}[\sigma(\mathbf{Fx})y] - \mathbb{E}_{(\mathbf{x},y)}[\hat{\sigma}_l(\mathbf{Fx})y] \right\| = o\left(1/\text{polylog}\, k\right), \tag{18}$$

when $|\eta| = \mathcal{O}(n^{-1/l})$ for some $l \in \mathbb{Z}^+$ and all $i \in \{1, \ldots, k\}$. Finally, using Theorem 1 with (17) and (18), we reach the statement of the theorem. □

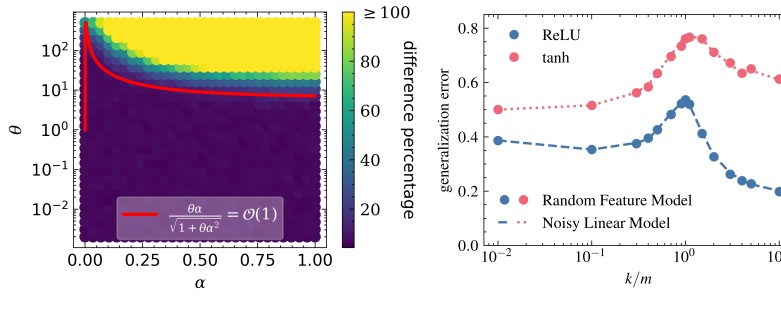

(a) Percentage difference between the generalization errors of the RFM and the noisy linear model (5). The red line is predicted by Corollary 3.

(b) Generalization errors of the RFM and the corresponding noisy linear model (5) for $\alpha = \mathcal{O}(1/\sqrt{n})$ (**misaligned**) and $\theta = n^{1/2}$.

Figure 1: *When does the noisy linear model (5) performs equivalent to the RFM under **spiked data**?* Left: the boundary of the equivalence with respect to spike magnitude $\theta$ and alignment $\alpha$ (controlling the input-label correlation). Right: an example case where the equivalence holds. For both of these figures, we use $\sigma_* = \sigma_{ReLU}$, $\lambda = 10^{-2}$, $n = 400$ and $m = 500$. The average of 50 Monte Carlo runs (numerical simulations) is plotted.

Theorem 2 specifies a finite-degree polynomial function (14) that is equivalent to any given activation function $\sigma(x)$. Furthermore, the degree of the polynomial is related to $\eta$ defined in (15). It is important to note that $\eta$ is related to the input-label correlation, which can be decomposed as

$$\mathbb{E}[(\mathbf{x} - \mathbb{E}[\mathbf{x}])(y - \mathbb{E}[y])] = \tilde{\mu}_1 \frac{\boldsymbol{\xi} + \theta\alpha\boldsymbol{\gamma}}{\sqrt{1 + \theta\alpha^2}} + \sum_{j>1} \tilde{\mu}_j \mathbb{E}_{\mathbf{x}}[\mathbf{x} H_j(\boldsymbol{\xi}^T \mathbf{x})], \tag{19}$$

where $\tilde{\mu}_j := \mathbb{E}_{z \sim \mathcal{N}(0,1)}[\sigma_*(z) H_j(z)]$ denotes $j$-th Hermite coefficient of $\sigma_*$. Here, $\eta$ depends on $(\boldsymbol{\xi} + \theta\alpha\boldsymbol{\gamma})/\sqrt{1 + \theta\alpha^2}$, which is the first term in the decomposition of the input-label correlation (19). Additionally, a higher value of $\|\boldsymbol{\xi} + \theta\alpha\boldsymbol{\gamma}\|/\sqrt{1 + \theta\alpha^2}$ implies a greater value of $\eta$ for any $\mathbf{F}$ sampled independent of $\boldsymbol{\gamma}$ and $\boldsymbol{\xi}$.

In the following sections, we will first examine the case where the equivalent activation function in equation (14) is linear, effectively reducing the model to the noisy linear model described in equation (5). Subsequently, we will focus on the more general case where the equivalent activation function is represented as a high-order polynomial.

### 4.3 CONDITION OF THE EQUIVALENCE TO THE NOISY LINEAR MODEL

In this section, we explore the "linear regime" of the RFM and delineate the conditions under which it aligns with the noisy linear model in the context of spiked data. Specifically, we examine the substitution of the activation function $\sigma(x)$ with $\hat{\sigma}_1(x) := (\mu_0 + \mu_1 x + \mu_* z)$ for $z \sim \mathcal{N}(0, 1)$. This transformation effectively simplifies the model to that of a noisy linear model, as described in equation (5). The conditions necessary for establishing equivalence between the noisy linear model and the RFM are articulated in the following corollary, which emerges directly from Theorem 2.

**Corollary 3.** *The noisy linear model (5) performs equivalent to the RFM with activation $\sigma$ in terms of training and generalization errors if $\eta = \mathcal{O}(n^{-1/2})$ in (15).*

Figure 1a vividly illustrates the disparity in generalization errors between the RFM and the noisy linear model across various degrees of alignment $\alpha$ and different values of spike magnitude $\theta$. The red line delineates the boundary of equivalence, as established by Corollary 3. Given that $\mathbf{f}_i$ is sampled independently of $\boldsymbol{\gamma}$ and $\boldsymbol{\xi}$, we observe that both $|\boldsymbol{\gamma}^T \mathbf{f}_i|$ and $|\boldsymbol{\xi}^T \mathbf{f}_i|$ scale as $\mathcal{O}(n^{-1/2})$. Consequently, the expression $|\theta\alpha/\sqrt{1 + \theta\alpha^2}| = \mathcal{O}(1)$ leads to the conclusion that $\eta = \mathcal{O}(n^{-1/2})$, which fulfills the condition outlined in Corollary 3.

As an illustrative example, we examine a misaligned scenario where $\alpha := \boldsymbol{\xi}^T \boldsymbol{\gamma} = \mathcal{O}(n^{-1/2})$. This condition is likely to occur when both $\boldsymbol{\gamma}$ and $\boldsymbol{\xi}$ are independently and identically sampled from $\mathcal{N}(0, \mathbf{I}_n)$ and are scaled such that $\|\boldsymbol{\gamma}\|_2 = \|\boldsymbol{\xi}\|_2 = 1$. Under these circumstances, we find that

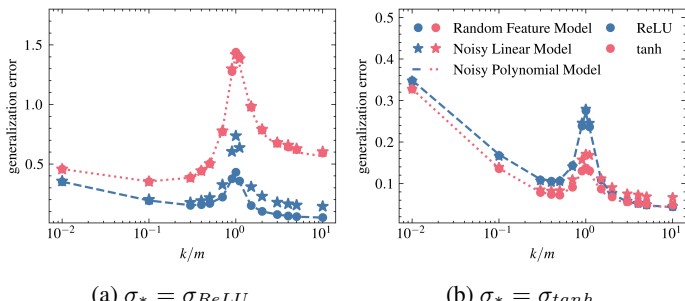

(a) $\sigma_* = \sigma_{ReLU}$        (b) $\sigma_* = \sigma_{tanh}$

Figure 2: *Equivalence to the **noisy polynomial model** (20) when the noisy linear model (5) is not enough.* Generalization errors of the RFM, the noisy linear model, and the noisy polynomial model for $\theta = n^{1/2}$ and $\alpha = 1$ (**aligned**). $\lambda = 10^{-3}$, $n = 400$ and $m = 500$. An average of 50 Monte Carlo runs is shown.

$\eta = \mathcal{O}(n^{-1/2})$ with high probability for values of $\theta$ that scale as $\theta \asymp n^{\beta}$, where $\beta \in [0, 1/2)$. Therefore, the RFM demonstrates performance equivalence to the noisy linear model with high probability in this context. Figure 1b visually represents this relationship, showcasing the training and generalization errors of both the RFM and the noisy linear model. Notably, the results indicate that this performance-wise equivalence is maintained across different activation functions.

### 4.4 Beyond the Noisy Linear Model: High-Order Polynomial Models

In this section, we explore the performance-wise equivalence of the RFM to the noisy polynomial model beyond the linear regime. Specifically, we focus on the model represented by

$$\boldsymbol{\omega}^T \hat{\sigma}_l(\mathbf{Fx}), \tag{20}$$

for $l > 2$, where $\hat{\sigma}_l$ is the equivalent polynomial function defined in (14).

If the condition in Corollary 3 does not hold, the equivalence between the RFM and the noisy linear model is no longer applicable. In this case, Theorem 2 provides high-order polynomial models (20) equivalent to the RFM. Thus, we delve into the "nonlinear regime" of the RFM under spiked data conditions. While much of the existing theoretical literature has concentrated on the linear regime of the RFM, Theorem 2 suggests that the nonlinear components of random features become increasingly significant when the spike signal $\boldsymbol{\gamma}$ and label signal $\boldsymbol{\xi}$ are aligned, indicating a strong input-label correlation. This relationship is illustrated in Figure 2, which demonstrates that the generalization errors of the RFM closely match those of the equivalent polynomial model in the aligned case ($\alpha := \boldsymbol{\gamma}^T \boldsymbol{\xi} = 1$). Notably, although the equivalence between the RFM and the noisy linear model may not hold due to violated conditions for linear equivalence, it is essential to recognize that such equivalence can still be achieved depending on the choice of activation function $\sigma$ and label function $\sigma_*$. This nuanced understanding is further characterized in the following remark.

*Remark* 4. When we assume that $\eta = \mathcal{O}(n^{-1/4})$, this allows the equivalent activation function to be represented as a third-degree polynomial. Let $\mu_0, \mu_1, \mu_2, \mu_3$ and $\tilde{\mu}_0, \tilde{\mu}_1, \tilde{\mu}_2, \tilde{\mu}_3$ denote the first four Hermite coefficients (13) of the functions $\sigma$ and $\sigma_*$, respectively. In this case, we can express $\mathbb{E}[\hat{\sigma}_l(\mathbf{f}_i^T \mathbf{x})y] = \sum_{j=0}^{3} \frac{1}{j!} \mu_j \tilde{\mu}_j \eta_i^j$ due to (112). This formulation reveals that the noisy polynomial model represented in (20) simplifies to a second-degree model if the product $\mu_3 \tilde{\mu}_3 = 0$. Furthermore, it reduces to the noisy linear model described in (5) under the conditions that both $\mu_2 \tilde{\mu}_2 = 0$ and $\mu_3 \tilde{\mu}_3 = 0$. These relationships highlight how specific properties of the Hermite coefficients dictate the behavior of the model, illustrating the nuanced interplay between activation functions and their corresponding polynomial representations.

The implications of Remark 4 are vividly illustrated in Figure 2. In 2a, we observe that the noisy linear model suffices for the tanh activation function in terms of equivalence, while a high-order polynomial model is necessary for the ReLU activation, given that $\sigma_*$ is ReLU. Conversely, 2b showcases the same simulation with $\sigma_*$ as tanh, demonstrating that a noisy polynomial model is essential to achieve equivalence to the RFM for tanh activation, whereas the noisy linear model remains adequate for ReLU activation. This distinction arises from the specific properties of the Hermite coefficients: for tanh, we have $\mu_2 = 0$ and $\mu_3 \neq 0$, while for ReLU, $\mu_2 \neq 0$ and $\mu_3 = 0$.

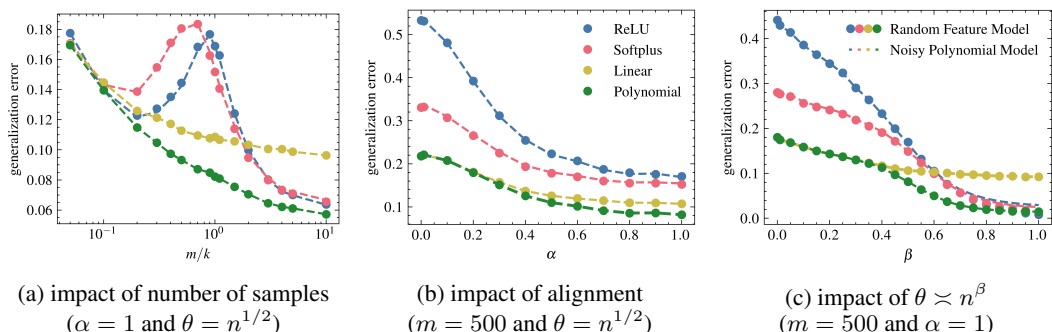

(a) impact of number of samples
($\alpha = 1$ and $\theta = n^{1/2}$)

(b) impact of alignment
($m = 500$ and $\theta = n^{1/2}$)

(c) impact of $\theta \asymp n^{\beta}$
($m = 500$ and $\alpha = 1$)

Figure 3: *Comparison of activation functions* - generalization errors of the RFM with different nonlinearities (linear, polynomial, Softplus, ReLU) with respect to number of samples (on the left), alignment (on the center), and spike magnitude $\theta \asymp n^{\beta}$ (on the right). Here, $n = 400$, $k = 500$, $\lambda = 10^{-2}$, and $\sigma_* = \sigma_{ReLU}$. We limit the degree of the equivalent polynomial model (14) to a maximum of $l = 4$ for numerical stability. We plot the average of 50 Monte Carlo runs.

Together, Theorem 2, Remark 4, and Figure 2 underscore the critical influence of the label generation process on the study of equivalent models.

## 5 NONLINEARITY IN RFM BENEFITS HIGH INPUT-LABEL CORRELATION

In this section, we conduct a comparative analysis of the RFM employing various activation functions, focusing specifically on their generalization performance. To establish a benchmark, we first consider the following "optimal" linear activation function for the RFM:

$$\sigma_{linear}(x) = a_0 + a_1 x, \tag{21}$$

where $a_0, a_1 \in \mathbb{R}$ are coefficients determined numerically to minimize the generalization error. This "optimal" linear activation serves as a critical reference point for understanding the conditions under which the RFM surpasses traditional linear models in performance.

According to Theorem 2, the RFM with any activation function that meets our assumptions performs equivalently to a noisy polynomial model. Notably, we find that $\eta = \mathcal{O}(n^{-1/4})$ in (15) holds with high probability when $\beta < 1/2$ for $\theta \asymp n^{\beta}$ and $\mathbf{F}$ is sampled independently of $\boldsymbol{\gamma}$ and $\boldsymbol{\xi}$. Therefore, in this context, we can define the following third-degree polynomial as an optimal activation function:

$$\sigma_{polynomial}(x) = b_0 + b_1 x + \frac{1}{2}b_2(x^2 - 1) + \frac{1}{6}b_3(x^3 - 3x) + b_4 z, \tag{22}$$

where $z \sim \mathcal{N}(0, 1)$ and the coefficients $b_0, b_1, b_2, b_3, b_4 \in \mathbb{R}$ are determined numerically to minimize the generalization error, akin to the approach taken for the optimal linear activation in (21).

We proceed to compare the generalization performance of the RFM across various activation functions: linear (21), polynomial (22), ReLU (9), and Softplus (9). Figure 3a presents this comparison in relation to the number of training samples. Notably, the generalization error of the RFM aligns closely with that of the equivalent polynomial model, as predicted by Theorem 2. Additionally, we observe the double-descent phenomenon (Belkin et al., 2019; Geiger et al., 2020), characterized by an initial decrease in generalization error, followed by an increase, and then a second decrease as the number of samples (or features) increases, particularly for the ReLU and Softplus activation functions. In contrast, this phenomenon is absent for the linear (21) and polynomial (22) activation functions, as their coefficients are numerically optimized. Similar findings have been documented under isotropic data conditions (Wang & Bento, 2022; Demir & Doğan, 2023). Interestingly, the RFM with linear activation outperforms both ReLU and Softplus in the mid-range of $k/m$ due to the double-descent behavior exhibited by those two functions. However, throughout the entire range of $k/m$, the polynomial RFM consistently demonstrates lower generalization error than its counterparts, highlighting its superior performance and robustness across different scenarios.

Next, we investigate the impact of alignment, defined as $\alpha := \boldsymbol{\gamma}^T \boldsymbol{\xi}$ controlling the input-label correlation, on the generalization performance of the RFM across the aforementioned activation functions in Figure 3b. In the misaligned scenario ($\alpha \leq 0.3$), we observe relatively high generalization errors,

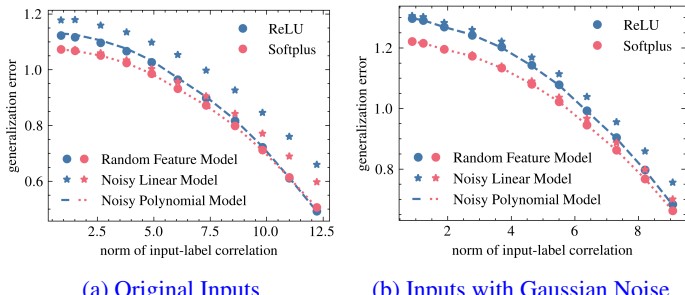

(a) Original Inputs         (b) Inputs with Gaussian Noise

Figure 4: *CIFAR-10 classification (airplane vs. automobile) - Impact of input-label correlation.* Input-label correlation is controlled by flipping the true labels. Here, the highest value of the correlation norm corresponds to the case with true labels, while the smallest value corresponds to random labels, and the rest is interpolation in between. $l = 5, \lambda = 10^{-1}$, $k = n = 3072$, and $m = 4000$. An average of 50 Monte Carlo runs is shown. Experimental details are explained in Appendix G.

with the errors for both linear and polynomial activation functions closely aligned. Specifically, for the range $0.3 < \alpha \leq 0.6$, we note a gradual separation in the generalization errors of the linear and polynomial functions. In cases of high alignment ($\alpha > 0.6$), the polynomial RFM distinctly outperforms the linear RFM, achieving significantly lower generalization errors. This observation corroborates our theoretical predictions that a strong input-label correlation—facilitated by high values of $\alpha$ —enables the RFM to outperform linear models.

In Figure 3c, we examine the influence of the spike magnitude scale, denoted as $\theta$, on the generalization errors of the RFM across the previously discussed activation functions. Notably, we observe that generalization errors for all activation functions decline as the scale of the spike magnitude increases. In the low spike magnitude regime ($\beta < 0.4$), both the RFM with linear and polynomial activation functions yield similar generalization errors, which remain lower than those of the RFM utilizing ReLU and Softplus activations. As we transition into the high spike magnitude regime ($\beta \geq 0.4$), the polynomial RFM consistently outperforms the linear RFM, aligning with our expectations. Although our analysis is confined to $\beta < 0.5$, we find that the generalization errors for ReLU and Softplus continue to align with those of their equivalent noisy polynomial models, even at higher values of $\beta$. Importantly, our findings indicate that RFM with nonlinear activation functions surpasses the optimal linear model in scenarios characterized by high spike magnitudes.

To illustrate how our results translate to real-world datasets, we study the effect of input-label correlation on the CIFAR-10 dataset Krizhevsky et al. (2009). Figure 4a illustrates our experimental results in this case. We observe that the noisy polynomial model performs equivalent to the RFM while the equivalence of the RFM to the linear model depends on factors like input-label correlation, input covariance, and activation function. To isolate the impact of the input-label correlation, we add standard Gaussian noise to the data such that the linear model performs equivalent to the RFM for the case of weak input-label correlation. For this case, Figure 4b shows our results when the input has additional Gaussian noise. We see that the generalization errors of the RFM and the linear model are gradually separated as the input-label correlation is increased, confirming our insights.

## 6 CONCLUSION

In conclusion, this study explored the random feature model within the context of spiked covariance data, establishing a significant "universality" theorem that underpins our findings. We demonstrated that a strong correlation between inputs and labels is essential for the RFM to surpass linear models in generalization performance. Additionally, we established that the RFM achieves performance equivalent to noisy polynomial models, with the polynomial degree influenced by the magnitude of the input-label correlation. Our numerical simulations corroborated these theoretical findings, highlighting the RFM's superior effectiveness in environments with strong input-label correlation. This work highlights the critical role of data modeling in understanding and enhancing model performance. Moving forward, we believe that extending the data model presents a promising avenue for future research, allowing for deeper exploration of complex relationships and further optimization of high-dimensional learning strategies.

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

## A    Proof of Universality of Random Features for the Spiked Data

Here, we provide a proof of Theorem 1 by adapting the proof used by Hu & Lu (2023). For a given set of training samples $\{(\mathbf{x}_i, y_i)\}_{i=1}^m$, let $\mathbf{R} := [\mathbf{r}_1, \mathbf{r}_2, \ldots, \mathbf{r}_m]^T$ where $\mathbf{r}_i := \sigma(\mathbf{F}\mathbf{x}_i)$ for all $i \in \{1, 2, \ldots, m\}$. For a given $\mathbf{R}$, we then introduce the following perturbed optimization related to the original problem:

$$\Phi_{\mathbf{R}}(\tau_1, \tau_2) := \inf_{\boldsymbol{\omega} \in \mathbb{R}^k} \left\{ \sum_{i=1}^m \left( \boldsymbol{\omega}^T \mathbf{r}_i - y_i \right)^2 + Q(\boldsymbol{\omega}) \right\}, \tag{23}$$

where $Q(\boldsymbol{\omega}) := m\lambda ||\boldsymbol{\omega}||_2^2 + \tau_1 k \boldsymbol{\omega}^T \Sigma_x \boldsymbol{\omega} + \tau_2 k \boldsymbol{\omega}^T \Sigma_{xy}$ with $\Sigma_x := \mathbb{E}_{(\mathbf{x},y)}[\sigma(\mathbf{F}\mathbf{x})\sigma(\mathbf{F}\mathbf{x})^T]$ and $\Sigma_{xy} := \mathbb{E}_{(\mathbf{x},y)}[\sigma(\mathbf{F}\mathbf{x})y]$. Here, $Q(\boldsymbol{\omega})$ is a function of $\tau_1, \tau_2$ (in addition to $\boldsymbol{\omega}$) but we do not include them in the notation for simplicity. Note that $\Phi_{\mathbf{R}}(0,0)/m$ is equal to the training error ($\mathcal{T}_\sigma$) of the original problem (3). The additional terms in $Q(\boldsymbol{\omega})$ are related to the generalization error and we discuss them later. Since we are interested in two different functions $\sigma$ and $\hat{\sigma}$, we define $\mathbf{A} := [\mathbf{a}_1, \mathbf{a}_2, \ldots, \mathbf{a}_m]^T$ and $\mathbf{B} := [\mathbf{b}_1, \mathbf{b}_2, \ldots, \mathbf{b}_m]^T$ where $\mathbf{a}_i := \sigma(\mathbf{F}\mathbf{x}_i)$ and $\mathbf{b}_i := \hat{\sigma}(\mathbf{F}\mathbf{x}_i)$ for all $i \in \{1, 2, \ldots, m\}$. Then, the training error $\mathcal{T}_\sigma$ for the RFM with activation $\sigma$, and the training error $\mathcal{T}_{\hat{\sigma}}$ for the case of activation $\hat{\sigma}$ converge in probability to the same value $e_\mathcal{T} \geq 0$ if the following holds

$$\frac{\Phi_{\mathbf{A}}(0,0)}{m} \xrightarrow{\mathbb{P}} e_\mathcal{T} \quad \text{and} \quad \frac{\Phi_{\mathbf{B}}(0,0)}{m} \xrightarrow{\mathbb{P}} e_\mathcal{T}. \tag{24}$$

Since $\Phi_{\mathbf{R}}(0,0)/m = (k/m)(\Phi_{\mathbf{R}}(0,0)/k)$ and $(k/m) \in (0,\infty)$ by assumption (A.4), we focus on $\Phi_{\mathbf{R}}(0,0)/k$ for the rest of the proof in order to write the bounds in terms of $k$ only.

Now, we describe how we handle the generalization error. We introduce $\tau_1 k \boldsymbol{\omega}^T \Sigma_x \boldsymbol{\omega} + \tau_2 k \boldsymbol{\omega}^T \Sigma_{xy}$ into $Q(\boldsymbol{\omega})$ to be used in the calculation of the generalization error. Before describing the relationship between the additional terms and the generalization error, we need the following two additional assumptions for the study of generalization error.

**Additional Assumptions for Generalization Error**

(A.7) There exists $\tau_* = \mathcal{O}(1/\sqrt{k})$ and $s \in (0, \infty)$ such that $Q(\boldsymbol{\omega})/m$ is $s$-strongly convex for all $\tau_1, \tau_2 \in (-\tau_*, \tau_*)$.

(A.8) There exists a limiting function $q^*(\tau_1, \tau_2)$ such that $\Phi_{\mathbf{R}}(\tau_1, \tau_2)/k \xrightarrow{\mathbb{P}} q^*(\tau_1, \tau_2)$ for all $\tau_1, \tau_2 \in (-\tau_*, \tau_*)$. Furthermore, the partial derivatives of $q^*(\tau_1, \tau_2)$ exist at $\tau_1 = \tau_2 = 0$, denoted as $\frac{\partial}{\partial \tau_1} q^*(0,0) = \rho^*$ and $\frac{\partial}{\partial \tau_2} q^*(0,0) = \pi^*$.

Assumption (A.7) is required to keep the perturbed objective (23) convex and bound the norm of optimal $\boldsymbol{\omega}$ of (23). Based on the assumption (A.8), the generalization error of the RFM with $\sigma$ can be specified as follows $\mathcal{G}_\sigma \xrightarrow{\mathbb{P}} \rho^* - 2\pi^* + \mathbb{E}_y[y^2]$. Furthermore, Lemma 10 in Appendix E states that the generalization error $\mathcal{G}_\sigma$ for the RFM with activation $\sigma$ and the generalization error $\mathcal{G}_{\hat{\sigma}}$ for the case of activation $\hat{\sigma}$ converge in probability to the same value $e_\mathcal{G} \geq 0$ if the following holds

$$\frac{\Phi_{\mathbf{A}}(\tau_1, \tau_2)}{k} \xrightarrow{\mathbb{P}} \phi \quad \text{and} \quad \frac{\Phi_{\mathbf{B}}(\tau_1, \tau_2)}{k} \xrightarrow{\mathbb{P}} \phi, \tag{25}$$

for any $\phi \in \mathbb{R}$ and any $\tau_1, \tau_2 \in (-\tau_*, \tau_*)$. The rest of this appendix shows that (25) holds, which then implies the statement of Theorem 1. To show the convergence in probability (25), we take advantage of a test function $\varphi(x)$. Specifically, Lemma 11 in Appendix E states that (25) holds if

$$\left| \mathbb{E}\, \varphi\left( \frac{1}{k} \Phi_{\mathbf{A}} \right) - \mathbb{E}\, \varphi\left( \frac{1}{k} \Phi_{\mathbf{B}} \right) \right| \leq \max\left\{ ||\varphi||_\infty, ||\varphi'||_\infty, ||\varphi''||_\infty \right\} \kappa_k, \tag{26}$$

for some $\kappa_k = o(1)$ and every bounded test function $\varphi(x)$ which has bounded first two derivatives. From now on, we use $\Phi_{\mathbf{R}}$ to denote $\Phi_{\mathbf{R}}(\tau_1, \tau_2)$. We can continue as follows

$$
\left| \mathbb{E}\varphi\left(\frac{1}{k}\Phi_{\boldsymbol{A}}\right) - \mathbb{E}\varphi\left(\frac{1}{k}\Phi_{\boldsymbol{B}}\right) \right| \leq \mathbb{E}\left| \mathbb{E}_{\backslash \boldsymbol{F}}\left[ \varphi\left(\frac{1}{k}\Phi_{\boldsymbol{A}}\right) \right] - \mathbb{E}_{\backslash \boldsymbol{F}}\left[ \varphi\left(\frac{1}{k}\Phi_{\boldsymbol{B}}\right) \right] \right|,
$$

$$
= \mathbb{E}\left[ \left| \mathbb{E}_{\backslash \boldsymbol{F}}\left[ \varphi\left(\frac{1}{k}\Phi_{\boldsymbol{A}}\right) \right] - \mathbb{E}_{\backslash \boldsymbol{F}}\left[ \varphi\left(\frac{1}{k}\Phi_{\boldsymbol{B}}\right) \right] \right| \left( \mathbb{1}_{\mathcal{A}}(\boldsymbol{F}) + \mathbb{1}_{\mathcal{A}^c}(\boldsymbol{F}) \right) \right],
$$

$$
\leq \sup_{\boldsymbol{F} \in \mathcal{A}} \left| \mathbb{E}_{\backslash \boldsymbol{F}}\left[ \varphi\left(\frac{1}{k}\Phi_{\boldsymbol{A}}\right) \right] - \mathbb{E}_{\backslash \boldsymbol{F}}\left[ \varphi\left(\frac{1}{k}\Phi_{\boldsymbol{B}}\right) \right] \right| + 2\|\varphi\|_{\infty}\mathbb{P}\left(\mathcal{A}^c\right),
\tag{27}
$$

where $\mathbb{E}_{\backslash \boldsymbol{F}}[\cdot]$ denotes the conditional expectation for a fixed feature matrix $\boldsymbol{F}$ and $\mathbb{1}_{\mathcal{A}}(\boldsymbol{F})$ is a indicator function (1 if $\mathbf{F} \in \mathcal{A}$). We refer to $\mathcal{A}$ as the admissible set of feature matrices defined in Appendix B. We also have $\mathbb{P}\left(\mathcal{A}^c\right) = o(1)$, proof of which is also given in Appendix B. Now, we need to show

$$
\sup_{\boldsymbol{F} \in \mathcal{A}} \left| \mathbb{E}_{\backslash \boldsymbol{F}}\varphi\left(\frac{1}{k}\Phi_{\boldsymbol{A}}\right) - \mathbb{E}_{\backslash \boldsymbol{F}}\varphi\left(\frac{1}{k}\Phi_{\boldsymbol{B}}\right) \right| \leq \max\{\|\varphi'\|_{\infty}, \|\varphi''\|_{\infty}\}\kappa_k,
\tag{28}
$$

for some $\kappa_k = o(1)$. To find a useful bound for (28), we interpolate the path from $\Phi_{\mathbf{A}}$ to $\Phi_{\mathbf{B}}$ following Lindeberg's method (Lindeberg, 1922; Korada & Montanari, 2011)

$$
\Phi_t := \min_{\boldsymbol{\omega} \in \mathbb{R}^k} \left\{ \sum_{i=1}^{t} \left(\boldsymbol{\omega}^T\mathbf{b}_i - y_i\right)^2 + \sum_{i=t+1}^{m} \left(\boldsymbol{\omega}^T\mathbf{a}_i - y_i\right)^2 + Q(\boldsymbol{\omega}) \right\},
\tag{29}
$$

$$
\hat{\boldsymbol{\omega}}_t := \arg\min_{\boldsymbol{\omega} \in \mathbb{R}^k} \left\{ \sum_{i=1}^{t} \left(\boldsymbol{\omega}^T\mathbf{b}_i - y_i\right)^2 + \sum_{i=t+1}^{m} \left(\boldsymbol{\omega}^T\mathbf{a}_i - y_i\right)^2 + Q(\boldsymbol{\omega}) \right\},
\tag{30}
$$

for $0 \leq t \leq m$. Then,

$$
\left| \mathbb{E}_{\backslash \boldsymbol{F}}\varphi\left(\frac{1}{k}\Phi_{\mathbf{A}}\right) - \mathbb{E}_{\backslash \boldsymbol{F}}\varphi\left(\frac{1}{k}\Phi_{\mathbf{B}}\right) \right| \leq \sum_{t=1}^{m} \left| \mathbb{E}_{\backslash \boldsymbol{F}}\varphi\left(\frac{1}{k}\Phi_t\right) - \mathbb{E}_{\backslash \boldsymbol{F}}\varphi\left(\frac{1}{k}\Phi_{t-1}\right) \right|,
\tag{31}
$$

due to triangle inequality. Therefore, we focus on showing the following

$$
\left| \mathbb{E}_{\backslash \boldsymbol{F}}\varphi\left(\frac{1}{k}\Phi_t\right) - \mathbb{E}_{\backslash \boldsymbol{F}}\varphi\left(\frac{1}{k}\Phi_{t-1}\right) \right| \leq \max\{\|\varphi'\|_{\infty}, \|\varphi''\|_{\infty}\}\frac{\kappa_k}{k},
\tag{32}
$$

since $k/m \in (0, \infty)$. Here, $\Phi_t$ and $\Phi_{t-1}$ can be seen as perturbations of a common "leave-one-out" problem defined as

$$
\Phi_{\backslash t} := \min_{\boldsymbol{\omega} \in \mathbb{R}^k} \left\{ \sum_{i=1}^{t-1} \left(\boldsymbol{\omega}^T\mathbf{b}_i - y_i\right)^2 + \sum_{i=t+1}^{m} \left(\boldsymbol{\omega}^T\mathbf{a}_i - y_i\right)^2 + Q(\boldsymbol{\omega}) \right\},
\tag{33}
$$

$$
\hat{\boldsymbol{\omega}}_{\backslash t} := \arg\min_{\boldsymbol{\omega} \in \mathbb{R}^k} \left\{ \sum_{i=1}^{t-1} \left(\boldsymbol{\omega}^T\mathbf{b}_i - y_i\right)^2 + \sum_{i=t+1}^{m} \left(\boldsymbol{\omega}^T\mathbf{a}_i - y_i\right)^2 + Q(\boldsymbol{\omega}) \right\}.
\tag{34}
$$

Note that $t$-th sample is left out in the definition of $\Phi_{\backslash t}$ and $\boldsymbol{\omega}_{\backslash t}$. Since $\Phi_t \approx \Phi_{\backslash t}$, we apply Taylor's expansion around $\Phi_{\backslash t}$

$$
\varphi\left(\frac{1}{k}\Phi_t\right) = \varphi\left(\frac{1}{k}\Phi_{\backslash t}\right) + \frac{1}{k}\varphi'\left(\frac{1}{k}\Phi_{\backslash t}\right)\left(\Phi_t - \Phi_{\backslash t}\right) + \frac{1}{2k^2}\varphi''(\zeta)\left(\Phi_t - \Phi_{\backslash t}\right)^2,
\tag{35}
$$

where $\zeta$ is some value that lies between $\frac{1}{k}\Phi_t$ and $\frac{1}{k}\Phi_{\backslash t}$. Similarly, applying Taylor's expansion to $\varphi\left(\Phi_{t-1}\right)$ around $\Phi_{\backslash t}$, and then subtracting it from the last equation, we get

$$
\left| \mathbb{E}_{\backslash \boldsymbol{F}}\left[ \varphi\left(\frac{1}{k}\Phi_t\right) \right] - \mathbb{E}_{\backslash \boldsymbol{F}}\left[ \varphi\left(\frac{1}{k}\Phi_{t-1}\right) \right] \right|
$$

$$
\leq \frac{\|\varphi'(x)\|_{\infty}}{k}\mathbb{E}_{\backslash \boldsymbol{F}}\left| \mathbb{E}_t\left(\Phi_t - \Phi_{t-1}\right) \right| + \frac{\|\varphi''(x)\|_{\infty}}{2k}\frac{\left(\mathbb{E}_{\backslash \boldsymbol{F}}\left(\Phi_t - \Phi_{\backslash t}\right)^2 + \mathbb{E}_{\backslash \boldsymbol{F}}\left(\Phi_{t-1} - \Phi_{\backslash t}\right)^2\right)}{k},
\tag{36}
$$

where $\mathbb{E}_t$ denotes the conditional expectation over the random vectors $\{\boldsymbol{a}_t, \boldsymbol{b}_t\}$ associated with the $t$-th training sample, while $\{\boldsymbol{a}_i, \boldsymbol{b}_i\}_{i \neq t}$ and $\boldsymbol{F}$ are fixed.

To bound the terms on the right side of (36), we first define a surrogate optimization problem:

$$\Psi_t(\mathbf{r}) := \Phi_{\backslash t} + \min_{\boldsymbol{\omega} \in \mathbb{R}^k} \left\{ \frac{1}{2} \left( \boldsymbol{\omega} - \hat{\boldsymbol{\omega}}_{\backslash t} \right)^T \boldsymbol{H}_{\backslash t} \left( \boldsymbol{\omega} - \hat{\boldsymbol{\omega}}_{\backslash t} \right) + \left( \boldsymbol{\omega}^T \mathbf{r} - y_t \right)^2 \right\}, \tag{37}$$

where

$$\boldsymbol{H}_{\backslash t} := 2 \sum_{i=1}^{t-1} \mathbf{b}_i \mathbf{b}_i^T + 2 \sum_{i=t+1}^{m} \mathbf{a}_i \mathbf{a}_i^T + \nabla^2 Q \left( \hat{\boldsymbol{\omega}}_{\backslash t} \right) \tag{38}$$

is the the Hessian matrix of the objective function $\Phi_{\backslash t}$ evaluated at $\hat{\boldsymbol{\omega}}_{\backslash t}$. Due to Lemma 12 in Appendix E, we have $\Phi_{t-1} = \Psi_t(\mathbf{a}_t)$ and $\Phi_t = \Psi_t(\mathbf{b}_t)$, which makes $\Psi_t(\mathbf{r})$ particularly interesting. We continue by simplifying $\Psi_t(\mathbf{r})$ as follows:

$$\Psi_t(\mathbf{r}) : = \Phi_{\backslash t} + \min_{\boldsymbol{\omega} \in \mathbb{R}^k} \left\{ \frac{1}{2} \left( \boldsymbol{\omega} - \hat{\boldsymbol{\omega}}_{\backslash t} \right)^T \boldsymbol{H}_{\backslash t} \left( \boldsymbol{\omega} - \hat{\boldsymbol{\omega}}_{\backslash t} \right) + \left( \boldsymbol{\omega}^T \mathbf{r} - y_t \right)^2 \right\}, \tag{39}$$

$$= \Phi_{\backslash t} + \min_{\tau} \min_{\mathbf{r}^T(\boldsymbol{\omega} - \hat{\boldsymbol{\omega}}_{\backslash t}) = \tau} \left\{ \frac{1}{2} \left( \boldsymbol{\omega} - \hat{\boldsymbol{\omega}}_{\backslash t} \right)^T \boldsymbol{H}_{\backslash t} \left( \boldsymbol{\omega} - \hat{\boldsymbol{\omega}}_{\backslash t} \right) + \left( \hat{\boldsymbol{\omega}}_{\backslash t}^T \mathbf{r} + \tau - y_t \right)^2 \right\}, \tag{40}$$

$$= \Phi_{\backslash t} + \min_{\tau} \left\{ \frac{\tau^2}{2\nu_t(\mathbf{r})} + \left( \hat{\boldsymbol{\omega}}_{\backslash t}^T \mathbf{r} + \tau - y_t \right)^2 \right\}, \quad \text{with} \quad \nu_t(\mathbf{r}) := \mathbf{r}^T \boldsymbol{H}_{\backslash t}^{-1} \mathbf{r}, \tag{41}$$

$$= \Phi_{\backslash t} + \frac{\left( \hat{\boldsymbol{\omega}}_{\backslash t}^T \mathbf{r} - y_t \right)^2}{2\nu_t(\mathbf{r}) + 1}, \tag{42}$$

$$\leq \Phi_{\backslash t} + \left( \hat{\boldsymbol{\omega}}_{\backslash t}^T \mathbf{r} - y_t \right)^2, \tag{43}$$

where the inequality in the last line is achieved by using $\tau = 0$ in (41) while the rest of the steps are trivial. Next, the following is due to Lemma 13 in Appendix E

$$\max \left\{ \mathbb{E}_{\backslash \boldsymbol{F}} \left( \Psi_t(\boldsymbol{b}_t) - \Phi_{\backslash t} \right)^2, \mathbb{E}_{\backslash \boldsymbol{F}} \left( \Psi_t(\boldsymbol{a}_t) - \Phi_{\backslash t} \right)^2 \right\} \leq k^{1-\epsilon} \text{ polylog } k, \tag{44}$$

which holds uniformly over $\boldsymbol{F} \in \mathcal{A}$ and $t \in \{1, 2, \ldots, m\}$ for some $\epsilon > 0$ satisfying $\theta^2 \leq n^{1-\epsilon}$. Then, we can reach

$$\frac{1}{k} \mathbb{E}_{\backslash \boldsymbol{F}} \left( \Phi_t - \Phi_{\backslash t} \right)^2 = \frac{1}{k} \mathbb{E}_{\backslash \boldsymbol{F}} \left( \Psi_t(\boldsymbol{b}_t) - \Phi_{\backslash t} \right)^2 = o(1), \tag{45}$$

and similarly,

$$\frac{1}{k} \mathbb{E}_{\backslash \boldsymbol{F}} \left( \Phi_{t-1} - \Phi_{\backslash t} \right)^2 = \frac{1}{k} \mathbb{E}_{\backslash \boldsymbol{F}} \left( \Psi_t(\boldsymbol{a}_t) - \Phi_{\backslash t} \right)^2 = o(1). \tag{46}$$

Next, by Lemma 12 and equation (41), we have

$$\mathbb{E}_{\backslash \boldsymbol{F}} \left| \mathbb{E}_k \left( \Phi_t - \Phi_{t-1} \right) \right| = \mathbb{E}_{\backslash \boldsymbol{F}} \left| \mathbb{E}_t \left[ \Psi_t(\boldsymbol{b}_t) - \Psi_t(\boldsymbol{a}_t) \right] \right|, \tag{47}$$

$$= \mathbb{E}_{\backslash \boldsymbol{F}} \left| \mathbb{E}_t \left[ \frac{\left( \hat{\boldsymbol{\omega}}_{\backslash t}^T \mathbf{b}_t - y_t \right)^2}{2\nu_t(\mathbf{b}_t) + 1} - \frac{\left( \hat{\boldsymbol{\omega}}_{\backslash t}^T \mathbf{a}_t - y_t \right)^2}{2\nu_t(\mathbf{a}_t) + 1} \right] \right|, \tag{48}$$

$$\leq \frac{1}{2\bar{\nu}_t + 1} \mathbb{E}_{\backslash \boldsymbol{F}} \left| \mathbb{E}_t \left[ \left( \hat{\boldsymbol{\omega}}_{\backslash t}^T \mathbf{b}_t - y_t \right)^2 - \left( \hat{\boldsymbol{\omega}}_{\backslash t}^T \mathbf{a}_t - y_t \right)^2 \right] \right| + \Delta_t(\mathbf{a}_t) + \Delta_t(\mathbf{b}_t), \tag{49}$$

where $\bar{\nu}_t := \mathbb{E}_t \nu_t(\mathbf{a}_t)$ and $\Delta_t(\mathbf{r})$ is defined as

$$\Delta_t(\mathbf{r}) := \mathbb{E}_{\backslash \boldsymbol{F}} \left| \mathbb{E}_t \left( \frac{1}{2\nu_t(\mathbf{r}) + 1} - \frac{1}{2\bar{\nu}_t + 1} \right) \left( \hat{\boldsymbol{\omega}}_{\backslash t}^T \mathbf{r} - y_t \right)^2 \right|, \tag{50}$$

for which the following is expected to hold

$$\max\{\Delta_t(\mathbf{a}_t), \Delta_t(\mathbf{b}_t)\} = o(1), \tag{51}$$

due to the concentration of $\nu_t(\mathbf{a}_t)$ and $\nu_t(\mathbf{b}_t)$ around $\bar{\nu}_t$ (Lemma 17) and the fact that $\frac{1}{2\bar{\nu}_t+1} \le C$ for some $C > 0$ by (43). Next, we bound the remaining term in (49) as

$$\mathbb{E}_{\backslash \mathbf{F}} \left| \mathbb{E}_t \left[ \left( \hat{\boldsymbol{\omega}}_{\backslash t}^T \mathbf{b}_t - y_t \right)^2 \right] - \mathbb{E}_t \left[ \left( \hat{\boldsymbol{\omega}}_{\backslash t}^T \mathbf{a}_t - y_t \right)^2 \right] \right|, \tag{52}$$

$$= \mathbb{E}_{\backslash \mathbf{F}} \left| \hat{\boldsymbol{\omega}}_{\backslash t}^T \left( \mathbb{E}_t[\mathbf{b}_t \mathbf{b}_t^T] - \mathbb{E}_t[\mathbf{a}_t \mathbf{a}_t^T] \right) \hat{\boldsymbol{\omega}}_{\backslash t} - 2\hat{\boldsymbol{\omega}}_{\backslash t}^T \left( \mathbb{E}_t[\mathbf{b}_t y_t] - \mathbb{E}_t[\mathbf{a}_t y_t] \right) \right| \tag{53}$$

$$\overset{(a)}{\le} \mathbb{E}_{\backslash \mathbf{F}} \left| \hat{\boldsymbol{\omega}}_{\backslash t}^T \left( \mathbb{E}_t[\mathbf{b}_t \mathbf{b}_t^T] - \mathbb{E}_t[\mathbf{a}_t \mathbf{a}_t^T] \right) \hat{\boldsymbol{\omega}}_{\backslash t} \right| + 2\mathbb{E}_{\backslash \mathbf{F}} \left| \hat{\boldsymbol{\omega}}_{\backslash t}^T \left( \mathbb{E}_t[\mathbf{b}_t y_t] - \mathbb{E}_t[\mathbf{a}_t y_t] \right) \right| \tag{54}$$

$$\overset{(b)}{\le} \mathbb{E}_{\backslash \mathbf{F}} \left\| \hat{\boldsymbol{\omega}}_{\backslash t} \right\|^2 \mathbb{E}_{\backslash \mathbf{F}} \left\| \mathbb{E}_t[\mathbf{b}_t \mathbf{b}_t^T] - \mathbb{E}_t[\mathbf{a}_t \mathbf{a}_t^T] \right\| + 2\mathbb{E}_{\backslash \mathbf{F}} \left\| \hat{\boldsymbol{\omega}}_{\backslash t} \right\| \mathbb{E}_{\backslash \mathbf{F}} \left\| \mathbb{E}_t[\mathbf{b}_t y_t] - \mathbb{E}_t[\mathbf{a}_t y_t] \right\| \tag{55}$$

$$\overset{(c)}{=} o(1), \tag{56}$$

where we use triangle inequality to reach (a). Then, we apply Cauchy-Schwarz inequality in order to reach (b). In the last step, we use a bound on $\mathbb{E}_{\backslash \mathbf{F}} \left\| \hat{\boldsymbol{\omega}}_{\backslash t} \right\|^l$ (Lemma 18) with the equivalence of covariance matrices (10) and the equivalence of cross-covariance vectors (11) to reach (c). We can use (56) and (51) to bound the right-hand side of (49). Then, (45)-(49) together with (36) imply (32). Finally, (31)-(32) imply (28) and consequently (26), which completes our proof.

# B   ADMISSIBLE SET OF FEATURE MATRICES

A feature matrix $\mathbf{F} = [\mathbf{f}_1, \mathbf{f}_2, \dots, \mathbf{f}_k]^T$ is called admissible ($\mathbf{F} \in \mathcal{A}$) if it satisfies

$$\max_{1 \le i,j \le k} \left| \mathbf{f}_i^T \mathbf{f}_j + \theta \boldsymbol{\gamma}^T \mathbf{f}_j \mathbf{f}_i^T \boldsymbol{\gamma} - \delta_{ij} \right| \le \frac{\text{polylog } k}{\sqrt{k}}, \tag{57}$$

$$\|\mathbf{F}(\mathbf{I}_n + (\sqrt{1+\theta} - 1)\boldsymbol{\gamma}\boldsymbol{\gamma}^T)\| \le \sqrt{\theta} \text{ polylog } k, \tag{58}$$

where $\delta_{ij}$ denote the Kronecker delta. Note that (57) (see Lemma 5) and (58) (see Lemma 6) hold with high probability for $\mathbf{f}_i \sim \mathcal{N}\left(0, \frac{1}{n+\theta}\mathbf{I}_n\right)$ which is sampled independent of $\boldsymbol{\gamma}, \boldsymbol{\xi}$ and $\mathbf{f}_j$ for $i \neq j$.

**Lemma 5.** *Under assumptions (A.1)-(A.6), the following holds with high probability*

$$\max_{1 \le i < j \le k} \left| \mathbf{f}_i^T \mathbf{f}_j + \theta \boldsymbol{\gamma}^T \mathbf{f}_j \mathbf{f}_i^T \boldsymbol{\gamma} \right| \le \frac{polylog \ k}{\sqrt{k}}, \tag{59}$$

$$\max_{1 \le i \le k} \left| \|\mathbf{f}_i\|^2 + \theta(\boldsymbol{\gamma}^T \mathbf{f}_i)^2 - 1 \right| \le \frac{polylog \ k}{\sqrt{k}}. \tag{60}$$

*Proof.* We first apply triangle equality to reach

$$\left| \mathbf{f}_i^T \mathbf{f}_j + \theta \boldsymbol{\gamma}^T \mathbf{f}_j \mathbf{f}_i^T \boldsymbol{\gamma} \right| \le |\mathbf{f}_i^T \mathbf{f}_j| + \theta |\boldsymbol{\gamma}^T \mathbf{f}_j \mathbf{f}_i^T \boldsymbol{\gamma}|, \tag{61}$$

$$\left| \|\mathbf{f}_i\|^2 + \theta(\boldsymbol{\gamma}^T \mathbf{f}_i)^2 - 1 \right| \le \left| \|\mathbf{f}_i\|^2 - \frac{n}{n+\theta} \right| + \theta \left| (\boldsymbol{\gamma}^T \mathbf{f}_i)^2 - \frac{1}{n+\theta} \right|. \tag{62}$$

Next, we find probability bounds for each of the terms on the right-hand side of (61) and (62). Let's start with $|\mathbf{f}_i^T \mathbf{f}_j|$. Remember that $\mathbf{f}_i \sim \mathcal{N}\left(0, \frac{1}{n+\theta}\mathbf{I}_n\right)$. Then, for any $l \in \{1, \dots, n\}$, $f_{i,l}$ and $f_{j,l}$ are sub-Gaussian random variables with sub-Gaussian norm bounded by $\frac{C}{\sqrt{n+\theta}}$ for some $C > 0$ (Vershynin, 2018, Example 2.5.8). Therefore, $f_{i,l}f_{j,l}$ is a sub-exponential random variable with sub-exponential norm bounded by $\frac{C^2}{n+\theta}$ (Vershynin, 2018, Lemma 2.7.7). Since $\mathbf{f}_i^T \mathbf{f}_j = \sum_{l=1}^n f_{i,l}f_{j,l}$, we can apply Bernstein's inequality (Vershynin, 2018, Theorem 2.8.2)

$$\mathbb{P}\left(|\mathbf{f}_i^T \mathbf{f}_j| \ge \epsilon\right) \le 2e^{-c\min\left(\frac{\epsilon^2}{K^2 n}, \frac{\epsilon}{K}\right)}, \tag{63}$$

where $K = C^2/(n + \theta)$ is the sub-exponential norm bound and $c > 0$ is a constant. We set $\epsilon = (\log k)^2/\sqrt{k}$ and apply union bound so that we get

$$\mathbb{P}\left(\max_{1 \leq i < j \leq k} |\mathbf{f}_i^T \mathbf{f}_j| \geq \frac{(logk)^2}{\sqrt{k}}\right) \leq c_1 e^{-(\log k)^2/c_2}, \tag{64}$$

for some $c_1, c_2 > 0$. Note that we use $n/k \in (0, \infty)$ and $\theta \leq \sqrt{n}$ to write $n$ and $\theta$ in terms of $k$. Also, we may reach the following inequality using the analogous steps:

$$\mathbb{P}\left(\max_{1 \leq i \leq k} \left|\|\mathbf{f}_i\|^2 - \frac{n}{n + \theta}\right| \geq \frac{(logk)^2}{\sqrt{k}}\right) \leq c_1 e^{-(\log k)^2/c_2}, \tag{65}$$

for some $c_1, c_2 > 0$. Note that $n/(n + \theta)$ appears in the equation since $\mathbb{E}[\|\mathbf{f}_i\|^2] = n/(n + \theta)$. Now, we focus on $|\boldsymbol{\gamma}^T \mathbf{f}_j \mathbf{f}_i^T \boldsymbol{\gamma}|$. First, note that $(\mathbf{f}_i^T \boldsymbol{\gamma}) \sim \mathcal{N}(0, 1/(n + \theta))$ since $\boldsymbol{\gamma}$ is a deterministic vector with $\|\boldsymbol{\gamma}\| = 1$. Therefore, $(\mathbf{f}_i^T \boldsymbol{\gamma})$ and $(\mathbf{f}_j^T \boldsymbol{\gamma})$ are sub-Gaussian random variables with sub-Gaussian norm bounded by $\frac{C}{\sqrt{n+\theta}}$ for some $C > 0$ (Vershynin, 2018, Example 2.5.8) so $(\boldsymbol{\gamma}^T \mathbf{f}_j \mathbf{f}_i^T \boldsymbol{\gamma})$ is a sub-exponential random variable with sub-exponential norm bounded by $\frac{C^2}{n+\theta}$ (Vershynin, 2018, Lemma 2.7.7). Thus, we have sub-exponential tail bound (Vershynin, 2018, Proposition 2.7.1) as follows

$$\mathbb{P}\left(|\boldsymbol{\gamma}^T \mathbf{f}_j \mathbf{f}_i^T \boldsymbol{\gamma}| \geq \epsilon\right) \leq 2e^{-C\epsilon(n+\theta)}, \tag{66}$$

for some $C > 0$. Setting $\epsilon = (\log k)^2/k$, we reach

$$\mathbb{P}\left(\max_{1 \leq i < j \leq k} |\boldsymbol{\gamma}^T \mathbf{f}_j \mathbf{f}_i^T \boldsymbol{\gamma}| \geq \frac{(logk)^2}{k}\right) \leq 2e^{-C(\log k)^2}, \tag{67}$$

for some $C > 0$. Following the analogous steps, we also get

$$\mathbb{P}\left(\max_{1 \leq i \leq k} \left|(\mathbf{f}_i^T \boldsymbol{\gamma})^2 - \frac{1}{n + \theta}\right| \geq \frac{(logk)^2}{k}\right) \leq 2e^{-C(\log k)^2}, \tag{68}$$

for some $C > 0$ since $\mathbb{E}[(\mathbf{f}_i^T \boldsymbol{\gamma})^2] = 1/(n + \theta)$. Using the found probability bounds for each of the terms on the right-hand side of (61) and (62), we complete the proof. $\square$

**Lemma 6.** *Under assumptions (A.1)-(A.6), the following hold with high probability*

$$\|\mathbf{F}(\mathbf{I}_n + \theta\boldsymbol{\gamma}\boldsymbol{\gamma}^T)\mathbf{F}^T\| \leq \theta \, polylog \, k. \tag{69}$$

*Proof.* Using the triangle inequality, we get

$$\|\mathbf{F}(\mathbf{I}_n + \theta\boldsymbol{\gamma}\boldsymbol{\gamma}^T)\mathbf{F}^T\| \leq \|\mathbf{F}\mathbf{F}^T\| + \theta\|\mathbf{F}\boldsymbol{\gamma}\boldsymbol{\gamma}^T\mathbf{F}^T\|. \tag{70}$$

First, we focus on $\|\mathbf{F}\mathbf{F}^T\| = \|\mathbf{F}\|^2$. Here, we can take advantage of a well-known result on the spectral norm of Gaussian random matrices (Vershynin, 2018, Corollary 7.3.3):

$$\mathbb{P}\left(\|\mathbf{F}\| \geq \sqrt{\frac{n}{n + \theta}} + \sqrt{\frac{k}{n + \theta}} + \epsilon\right) \leq 2e^{-c(n+\theta)\epsilon^2}, \tag{71}$$

$$\mathbb{P}\left(\|\mathbf{F}\| \geq \sqrt{\frac{n}{n + \theta}} + 2\sqrt{\frac{k}{n + \theta}}\right) \leq 2e^{-ck} \tag{72}$$

for any $\epsilon > 0$ and some $c > 0$. This result indicates $\|\mathbf{F}\mathbf{F}^T\| \leq$ polylog $k$ with high probability. Next, we work on $\|\mathbf{F}\boldsymbol{\gamma}\boldsymbol{\gamma}^T\mathbf{F}^T\|$. Since $(\mathbf{F}\boldsymbol{\gamma}\boldsymbol{\gamma}^T\mathbf{F}^T)_{i,j} = \mathbf{f}_i^T \boldsymbol{\gamma}\boldsymbol{\gamma}^T \mathbf{f}_j$, we may use the related results (67) - (68) from the proof of Lemma 5. Specifically, we have the following inequality with high probability

$$\max_{1 \leq i,j \leq k} \left|\boldsymbol{\gamma}^T \mathbf{f}_j \mathbf{f}_i^T \boldsymbol{\gamma} - \frac{\delta_{ij}}{n + \theta}\right| \leq \frac{\text{polylog } k}{k}, \tag{73}$$

$$\max_{1 \leq i,j \leq k} \left|\boldsymbol{\gamma}^T \mathbf{f}_j \mathbf{f}_i^T \boldsymbol{\gamma}\right| \leq \frac{\text{polylog } k}{k}, \tag{74}$$

where $\delta_{ij}$ denote the Kronecker delta function and. we use $1/(n + \theta) < C/k$ for some $C > 0$ in reaching the last line. It follows that the following holds with high probability

$$\|\mathbf{F}\boldsymbol{\gamma}\boldsymbol{\gamma}^T\mathbf{F}^T\| \leq \|\mathbf{F}\boldsymbol{\gamma}\boldsymbol{\gamma}^T\mathbf{F}^T\|_F = \sqrt{\sum_{i=1}^{k}\sum_{j=1}^{k}(\mathbf{f}_i^T \boldsymbol{\gamma}\boldsymbol{\gamma}^T \mathbf{f}_j)^2} \leq \text{polylog } k. \tag{75}$$

Using (70), we combine the results and reach the statement of the lemma. $\square$

## C  EQUIVALENCE OF COVARIANCE MATRICES

Here, we show that (10) holds for any activation function $\sigma$ satisfying assumption (A.6) and a second activation function defined as $\hat{\sigma}(x) := \mu_1 x + \mu_* z$ where $z \sim \mathcal{N}(0, 1)$. Note that trivially we have $\mathbb{E}_{\mathbf{x}}[\hat{\sigma}(\mathbf{Fx})\hat{\sigma}(\mathbf{Fx})^T] = \mu_1^2 \mathbf{F}(\mathbf{I}_n + \theta \boldsymbol{\gamma}\boldsymbol{\gamma}^T \mathbf{F}^T) + \mu_*^2 \mathbf{I}_k$. The following lemma specifies the equivalence of covariance matrices for the activation functions $\sigma$ and $\hat{\sigma}$.

**Lemma 7.** *Under assumptions (A.1)-(A.6) given in Section 3, the following holds*

$$\left\| \mathbb{E}_{\mathbf{x}}[\sigma(\mathbf{Fx})\sigma(\mathbf{Fx})^T] - (\mu_1^2 \hat{\mathbf{F}}\hat{\mathbf{F}}^T + \mu_*^2 \mathbf{I}_k) \right\| = o\left(1/\text{polylog } k\right), \tag{76}$$

*where* $\hat{\mathbf{F}} := \mathbf{F}(\mathbf{I}_n + (\sqrt{1+\theta} - 1)\boldsymbol{\gamma}\boldsymbol{\gamma}^T)$.

*Proof.* (Hu & Lu, 2023, Lemma 5) showed $\| \mathbb{E}_{\mathbf{g}}[\sigma(\mathbf{Fg})\sigma(\mathbf{Fg})^T] - (\mu_1^2 \mathbf{FF}^T + \mu_*^2 \mathbf{I}_k)\| = o\left(1/\text{polylog } k\right)$ under $\mathbf{g} \sim \mathcal{N}(0, \mathbf{I}_n)$ and some other assumptions on $\mathbf{F}$ matrix and activation function $\sigma$. Here, we adapt their proof to our case. First, note that we can consider $\sigma(\hat{\mathbf{F}}\mathbf{g})$ instead of $\sigma(\mathbf{Fx})$ since $\sigma(\hat{\mathbf{F}}\mathbf{g})$ is equal (in distribution) to $\sigma(\mathbf{Fx})$. Now, our goal is to show $\| \mathbb{E}_{\mathbf{g}}[\sigma(\hat{\mathbf{F}}\mathbf{g})\sigma(\hat{\mathbf{F}}\mathbf{g})^T] - (\mu_1^2 \hat{\mathbf{F}}\hat{\mathbf{F}}^T + \mu_*^2 \mathbf{I}_k)\| = o\left(1/\text{polylog } k\right)$. The $(i, j)$-th entry of $\mathbb{E}_{\mathbf{g}}[\sigma(\hat{\mathbf{F}}\mathbf{g})\sigma(\hat{\mathbf{F}}\mathbf{x})^T]$ is $\mathbb{E}\left[\sigma\left(\hat{\mathbf{f}}_i^T \mathbf{g}\right)\sigma\left(\hat{\mathbf{f}}_j^T \mathbf{g}\right)\right]$. Since $(\hat{\mathbf{f}}_i^T \mathbf{g}, \hat{\mathbf{f}}_j^T \mathbf{g})$ are jointly Gaussian, we can rewrite their joint distribution as that of $(z_i, \rho_{ij} z_i + \sqrt{1 - \rho_{ij}\rho_{ji}} z_j)$, where $z_i \sim \mathcal{N}(0, \|\hat{\mathbf{f}}_i\|^2)$, $z_j \sim \mathcal{N}(0, \|\hat{\mathbf{f}}_j\|^2)$ are two independent Gaussian variables and $\rho_{ij} := \hat{\mathbf{f}}_i^T \hat{\mathbf{f}}_j / \|\hat{\mathbf{f}}_i\|^2$. Then, for $i \neq j$, we have,

$$\mathbb{E}\left[\sigma\left(\hat{\mathbf{f}}_i^T \mathbf{g}\right)\sigma\left(\hat{\mathbf{f}}_j^T \mathbf{g}\right)\right] = \mathbb{E}\left[\sigma(z_i)\sigma\left(\rho_{ij} z_i + \sqrt{1 - \rho_{ij}\rho_{ji}} z_j\right)\right], \tag{77}$$

$$\overset{(a)}{=} \mathbb{E}[\sigma(z_i)\sigma\left(\sqrt{1 - \rho_{ij}\rho_{ji}} z_j\right)] + \rho_{ij}\mathbb{E}\left[\sigma(z_i) z_i \sigma'\left(\sqrt{1 - \rho_{ij}\rho_{ji}} z_j\right)\right]$$
$$\quad + \frac{1}{2}\rho_{ij}^2 \mathbb{E}\left[\sigma(z_i) z_i^2 \sigma''\left(\sqrt{1 - \rho_{ij}\rho_{ji}} z_j\right)\right] + \frac{1}{6}\rho_{ij}^3 \mathbb{E}\left[\sigma(z_i) z_i^3 \sigma'''(\zeta_{ij})\right], \tag{78}$$

$$\overset{(b)}{=} \hat{\mathbf{f}}_i^T \hat{\mathbf{f}}_j \mathbb{E}\sigma'(z_i)\mathbb{E}\sigma'\left(\sqrt{1 - \rho_{ij}\rho_{ji}} z_j\right) + \frac{1}{6}\rho_{ij}^3 \mathbb{E}\left[\sigma(z_i) z_i^3 \sigma'''(\zeta_{ij})\right], \tag{79}$$

$$\overset{(c)}{=} \hat{\mathbf{f}}_i^T \hat{\mathbf{f}}_j \mathbb{E}\sigma'(z_i)\mathbb{E}\sigma'(z_j) + R_{ij}, \tag{80}$$

where $\zeta_{ij}$ is some point between $\sqrt{1 - \rho_{ij}\rho_{ji}} z_j$ and $\rho_{ij} z_i + \sqrt{1 - \rho_{ij}\rho_{ji}} z_j$. Here, we apply Taylor's series expansion of $\sigma(\rho_{ij} z_i + \sqrt{1 - \rho_{ij}\rho_{ji}} z_j)$ around $\sqrt{1 - \rho_{ij}\rho_{ji}} z_j$ to reach (a). Then, we use the independence between $z_i$ and $z_j$, and the following identities: $\mathbb{E}\sigma(z_i) = \mathbb{E}[\sigma(z_i) z_i^2] = 0$ (due to $\sigma$ being an odd function) and $\mathbb{E}[\sigma(z_i) z_i] = \|\hat{\mathbf{f}}_i\|^2 \mathbb{E}[\sigma'(z_i)]$ (by Stein's lemma). To reach (c), we define the remainder term $R_{ij}$ as

$$R_{ij} = \hat{\mathbf{f}}_i^T \hat{\mathbf{f}}_j \mathbb{E}\sigma'(z_i)\left(\mathbb{E}\sigma'\left(\sqrt{1 - \rho_{ij}\rho_{ji}} z_j\right) - \mathbb{E}\sigma'(z_j)\right) + \frac{1}{6}\rho_{ij}^3 \mathbb{E}\left[\sigma(z_i) z_i^3 \sigma'''(\zeta_{ij})\right]. \tag{81}$$

For $i = j$, we define $R_{ii} = 0$. Then, we can verify the following decomposition using (80):

$$\mathbb{E}[\sigma(\hat{\mathbf{F}}\mathbf{g})\sigma(\hat{\mathbf{F}}\mathbf{g})^T] = (\mu_1 \mathbf{I}_k + \mathbf{D}_1)\hat{\mathbf{F}}\hat{\mathbf{F}}^T(\mu_1 \mathbf{I}_k + \mathbf{D}_1) + \mu_*^2 \mathbf{I}_k + \mathbf{D}_2 + \mathbf{D}_3 + \mathbf{R} \tag{82}$$

where we define $\mathbf{D}_1, \mathbf{D}_2, \mathbf{D}_3$ to be diagonal matrices as follows: $\mathbf{D}_1 := \text{diag}\left(\mathbb{E}\sigma'(z_i) - \mu_1\right)$, $\mathbf{D}_2 = \text{diag}\left(\mu_1^2 - \|\hat{\mathbf{f}}_i\|^2(\mathbb{E}\sigma'(z_i))^2\right)$, and $\mathbf{D}_3 = \text{diag}\left(\mathbb{E}\sigma^2(z_i) - \mu_1^2 - \mu_*^2\right)$.

Then, we have

$$\left\| \mathbb{E}[\sigma(\mathbf{Fx})\sigma(\mathbf{Fx})^T] - (\mu_1^2 \hat{\mathbf{F}}\hat{\mathbf{F}}^T + \mu_*^2 \mathbf{I}_k) \right\| \leq (2\mu_1 + \|\mathbf{D}_1\|)\|\hat{\mathbf{F}}\|^2 \|\mathbf{D}_1\| + \|\mathbf{D}_2\| + \|\mathbf{D}_3\| + \|\mathbf{R}\|, \tag{83}$$

$$\leq \frac{\|\hat{\mathbf{F}}\|^2 \text{ polylog} k}{\sqrt{k}}, \tag{84}$$

$$= o\left(1/\text{polylog } k\right), \tag{85}$$

where Lemma 8 is used for the bounds of $\|\mathbf{D}_1\|, \|\mathbf{D}_2\|, \|\mathbf{D}_3\|$ and $\|\mathbf{R}\|$. To reach the conclusion, we then take advantage of $\|\hat{\mathbf{F}}\|^2 \leq k^{1/2 - \epsilon/2}$ polylog $k$ (for some $\epsilon > 0$ satisfying $\theta \leq n^{1/2 - \epsilon/2}$), which is due to (58). $\qquad\square$

**Lemma 8.** *Suppose the setting in Lemma 7 and the definitions in its proof. Then,*

$$\max\left\{\|\mathbf{D}_1\|, \|\mathbf{D}_2\|, \|\mathbf{D}_3\|, \|\mathbf{R}\|\right\} \leq \frac{polylog\,k}{\sqrt{k}}. \tag{86}$$

*Proof.* Before bounding the terms, we define a truncated version of the activation function $\sigma$ as

$$\sigma_\epsilon(x) := \begin{cases} \sigma(x), & \text{if } |x| < \epsilon, \\ 0, & \text{otherwise}, \end{cases} \tag{87}$$

for $\epsilon = 2logk$, which is useful due to Lemma 19. Furthermore, $\sigma_\epsilon(x)$, its finite powers and its derivatives are bounded due to assumption (A.6). First, we focus on $\mathbf{D}_1$

$$\|\mathbf{D}_1\| \leq \max_i \mathbb{E}|\sigma'(z_i) - \mu_1| \tag{88}$$

$$\leq \max_i \mathbb{E}_{z \sim \mathcal{N}(0,1)}|\sigma'(\|\hat{\mathbf{f}}_i\|z) - \sigma'(z)| \tag{89}$$

$$\leq \max_i \mathbb{E}_{z \sim \mathcal{N}(0,1)}|\sigma'_\epsilon(\|\hat{\mathbf{f}}_i\|z) - \sigma'_\epsilon(z)| + \frac{polylog\,k}{k^{logk}} \tag{90}$$

$$\leq (\|\sigma''_\epsilon(x)\|_\infty \mathbb{E}[|z|]) \max_i \left|\|\hat{\mathbf{f}}_i\| - 1\right| + \frac{polylog\,k}{k^{logk}} \tag{91}$$

$$\leq (\|\sigma''_\epsilon(x)\|_\infty \mathbb{E}[|z|]) \max_i \left|\|\hat{\mathbf{f}}_i\|^2 - 1\right| + \frac{polylog\,k}{k^{logk}} \tag{92}$$

$$\leq \frac{polylog\,k}{\sqrt{k}}, \tag{93}$$

where we use Lemma 19 to get (90) and while the last line is due to (57) (Lemma 5) and bounded second derivative of $\sigma_\epsilon$. Similarly, one can show $\|\mathbf{D}_2\| \leq polylog\,k/\sqrt{k}$. Next, we study $\|\mathbf{D}_3\|$ as

$$\|\mathbf{D}_3\| \leq \max_i |\mathbb{E}\sigma^2(z_i) - \mu_1^2 - \mu_*^2| \tag{94}$$

$$= \max_i \mathbb{E}_{z \sim \mathcal{N}(0,1)}|\sigma^2(\|\hat{\mathbf{f}}_i\|z) - \sigma^2(z)| \tag{95}$$

$$= \max_i \mathbb{E}_{z \sim \mathcal{N}(0,1)}|\sigma_\epsilon^2(\|\hat{\mathbf{f}}_i\|z) - \sigma_\epsilon^2(z)| + \frac{polylog\,k}{k^{logk}} \tag{96}$$

$$\leq polylog\,k \max_i \left|\|\hat{\mathbf{f}}_i\| - 1\right| + \frac{polylog\,k}{k^{logk}} \tag{97}$$

$$\leq polylog\,k \max_i \left|\|\hat{\mathbf{f}}_i\|^2 - 1\right| + \frac{polylog\,k}{k^{logk}} \tag{98}$$

$$\leq \frac{polylog\,k}{\sqrt{k}}, \tag{99}$$

where we use the bounded derivative of $\sigma_\epsilon^2$ to reach (97) while the rest is similar to the bounding of $\|\mathbf{D}_1\|$. Lastly, to bound $\|\mathbf{R}\|$, one can first easily show that

$$\max_{1 \leq i,j \leq k} |R_{ij}| \leq \frac{polylog\,k}{k\sqrt{k}}. \tag{100}$$

using (57) (Lemma 5) and the definition of $R_{ij}$ in (81). Then, we have

$$\|\mathbf{R}\| \leq \|\mathbf{R}\|_F = \left(\sum_{i=1}^k \sum_{j=1}^k R_{ij}^2\right)^{1/2} \leq \frac{polylog\,k}{\sqrt{k}}, \tag{101}$$

which concludes our proof. $\square$

## D  EQUIVALENCE OF CROSS-COVARIANCE VECTORS

Here, we are interested in the cross-covariance $\mathbb{E}[\sigma(\mathbf{Fx})y]$. The following lemma specifies the equivalence of the cross-covariance vectors for activation function $\sigma$ and its finite-degree Hermite expansion $\hat{\sigma}_l$.

**Lemma 9.** *Suppose assumptions (A.1)-(A.6). Let $\eta_i := \mathbb{E}\left[\frac{(\mathbf{f}_i \mathbf{x})(\boldsymbol{\xi}^T \mathbf{x})}{\sqrt{1+\theta\alpha^2}}\right] = \frac{\mathbf{f}_i^T \boldsymbol{\xi} + \theta\alpha \mathbf{f}_i^T \boldsymbol{\gamma}}{\sqrt{1+\theta\alpha^2}}$. Assume that*

$$\eta := \max_{1 \leq i \leq k} |\eta_i| \leq \frac{C}{n^{1/l}}, \quad \text{for some } C > 0 \text{ and some } l \in \mathbb{Z}^+. \tag{102}$$

*Then, the following holds*

$$\left\| \mathbb{E}_{(\mathbf{x},y)} \left[\sigma(\mathbf{Fx})y\right] - \mathbb{E}_{(\mathbf{x},y)} \left[\hat{\sigma}_l(\mathbf{Fx})y\right] \right\| = o\left(1/polylog\, k\right), \tag{103}$$

*for $\hat{\sigma}_l(x)$ defined in (14).*

*Proof.* Let $\mathbf{x} \sim \mathcal{N}(0, \mathbf{I}_k + \theta\boldsymbol{\gamma}\boldsymbol{\gamma}^T)$. By assumption (A.3), we have

$$\mathbb{E}_{(\mathbf{x},y)} \left[\sigma(\mathbf{Fx})y\right] = \mathbb{E}_{\mathbf{x}} \left[\sigma(\mathbf{Fx})\sigma_*\left(\frac{\boldsymbol{\xi}^T \mathbf{x}}{\sqrt{1+\theta\alpha^2}}\right)\right] = \mathbb{E}_{\mathbf{g}} \left[\sigma\left(\hat{\mathbf{F}}\mathbf{g}\right)\sigma_*\left(\hat{\boldsymbol{\xi}}^T \mathbf{g}\right)\right], \tag{104}$$

where $\mathbf{g} \sim \mathcal{N}(0, \mathbf{I}_n)$, $\hat{\mathbf{F}} := \mathbf{F}(\mathbf{I}_n + (\sqrt{1+\theta} - 1)\boldsymbol{\gamma}\boldsymbol{\gamma}^T)$ and $\hat{\boldsymbol{\xi}} := \frac{(\mathbf{I}_n + (\sqrt{1+\theta}-1)\boldsymbol{\gamma}\boldsymbol{\gamma}^T)\boldsymbol{\xi}}{\sqrt{1+\theta\alpha^2}}$. Before dealing with the functions $\sigma$ and $\sigma_*$, we consider the joint distribution of $(\hat{\mathbf{f}}_i^T \mathbf{g}, \hat{\boldsymbol{\xi}}^T \mathbf{g})$, which is jointly Gaussian

$$(\hat{\mathbf{f}}_i^T \mathbf{g}, \hat{\boldsymbol{\xi}}^T \mathbf{g}) \sim \mathcal{N}\left(0, \begin{bmatrix} \|\hat{\mathbf{f}}_i\|^2 & \eta_i \\ \eta_i & 1 \end{bmatrix}\right), \tag{105}$$

Indeed, we can write an equivalent of $(\hat{\mathbf{f}}_i^T \mathbf{g}, \hat{\boldsymbol{\xi}}^T \mathbf{g})$ as follows

$$(\eta_i z + \sqrt{\|\hat{\mathbf{f}}_i\|^2 - \eta_i^2}\, z_i, z), \tag{106}$$

where $z_i, z \overset{i.i.d}{\sim} \mathcal{N}(0,1)$. This decomposition will be useful since we split correlated part $\eta_i z$ and uncorrelated part $\sqrt{\|\hat{\mathbf{f}}_i\|^2 - \eta_i^2}\, z_i$. Now, we use Hermite expansions of $\sigma$ and $\sigma_*$

$$\sigma(x) = \sum_{j=0}^{\infty} \frac{1}{j!} \mu_j H_j(x), \quad \text{and} \quad \sigma_*(x) = \sum_{j=0}^{\infty} \frac{1}{j!} \tilde{\mu}_j H_j(x), \tag{107}$$

where we define $H_j(z)$ to be the probabilist's $j$-th Hermite polynomial, $\mu_j := \mathbb{E}_{z \sim \mathcal{N}(0,1)}[H_j(z)\sigma(z)]$ and $\tilde{\mu}_j := \mathbb{E}_{z \sim \mathcal{N}(0,1)}[H_j(z)\sigma_*(z)]$. Then, we have

$$\mathbb{E}_{(\mathbf{x},y)} \left[\sigma(\mathbf{f}_i^T \mathbf{x})y\right] = \mathbb{E}_{\mathbf{g}} \left[\sigma(\hat{\mathbf{f}}_i^T \mathbf{g})\sigma_*(\hat{\boldsymbol{\xi}}^T \mathbf{g})\right], \tag{108}$$

$$= \mathbb{E}_{z_i, z} \left[\sigma\left(\eta_i z + \sqrt{\|\hat{\mathbf{f}}_i\|^2 - \eta_i^2}\, z_i\right)\sigma_*(z)\right] \tag{109}$$

$$= \mathbb{E}_{z_i, z} \left[\left(\sum_{j=0}^{\infty} \frac{1}{j!} \mu_j H_j\left(\eta_i z + \sqrt{\|\hat{\mathbf{f}}_i\|^2 - \eta_i^2}\, z_i\right)\right)\left(\sum_{j=0}^{\infty} \frac{1}{j!} \tilde{\mu}_j H_j(z)\right)\right], \tag{110}$$

$$\overset{(a)}{=} \sum_{j=0}^{\infty} \frac{1}{(j!)^2} \mu_j \tilde{\mu}_j \mathbb{E}_{z_i, z} \left[H_j\left(\eta_i z + \sqrt{\|\hat{\mathbf{f}}_i\|^2 - \eta_i^2}\, z_i\right) H_j(z)\right] \tag{111}$$

$$\overset{(b)}{=} \sum_{j=0}^{\infty} \frac{1}{j!} \mu_j \tilde{\mu}_j \eta_i^j \tag{112}$$

$$\overset{(c)}{=} \left(\sum_{j=0}^{l-1} \frac{1}{j!} \mu_j \tilde{\mu}_j \eta_i^j\right) + R_i, \tag{113}$$

$$\overset{(d)}{=} \mathbb{E}_{\mathbf{x},y} \left[\hat{\sigma}_l(\mathbf{f}_i^T \mathbf{x})y\right] + R_i, \tag{114}$$

where $R_i := \sum_{j=l}^{\infty} \frac{1}{j!} \mu_j \tilde{\mu}_j \eta_i^j$ to reach (c). To reach (a)-(b), we use the orthogonality of Hermite polynomials (Lemma 20). Finally, we get (d) by the definition of $\hat{\sigma}_l$. For $|R_i|$, we derive an upper bound as follows

$$|R_i| = \left| \sum_{j=l}^{\infty} \frac{1}{j!} \mu_j \tilde{\mu}_j \eta_i^j \right|, \tag{115}$$

$$\overset{(a)}{\leq} \sum_{j=l}^{\infty} \frac{1}{j!} |\mu_j \tilde{\mu}_j| |\eta_i^j|, \tag{116}$$

$$\overset{(b)}{\leq} \frac{C}{k}, \tag{117}$$

for some $C > 0$. The triangle inequality is used to reach (a) while we get (b) using $|\eta_i^l| = \mathcal{O}(1/k)$ (by the assumption of the lemma). Finally, we have

$$\left\| \underset{(\mathbf{x},y)}{\mathbb{E}} [\sigma(\mathbf{F}\mathbf{x})y] - \underset{(\mathbf{x},y)}{\mathbb{E}} [\hat{\sigma}_l(\mathbf{F}\mathbf{x})y] \right\| \leq \sqrt{\sum_{i=1}^{k} \left| \mathbb{E}[\sigma(\mathbf{f}_i^T\mathbf{x})y] - \mathbb{E}[\hat{\sigma}_l(\mathbf{f}_i^T\mathbf{x})y] \right|^2} = \sqrt{\sum_{i=1}^{k} R_i^2} \leq \frac{C}{\sqrt{k}}, \tag{118}$$

which completes the proof. $\qquad\square$

# E  AUXILIARY RESULTS FOR THE PROOFS

In this section, we provide auxiliary results that are used in the proofs given in the previous sections.

**Lemma 10.** *Suppose assumptions (A.1)-(A.6) in Section 3 and additional assumptions (A.7)-(A.8) in Appendix A. Then,*

$$\mathcal{G}_\sigma \overset{\mathbb{P}}{\to} e_{\mathcal{G}} \quad \textit{if and only if} \quad \mathcal{G}_{\hat{\sigma}} \overset{\mathbb{P}}{\to} e_{\mathcal{G}}, \tag{119}$$

*if the following holds,*

$$\frac{\Phi_{\mathbf{A}}(\tau_1, \tau_2)}{k} \overset{\mathbb{P}}{\to} \phi \quad \textit{if and only if} \quad \frac{\Phi_{\mathbf{B}}(\tau_1, \tau_2)}{k} \overset{\mathbb{P}}{\to} \phi, \tag{120}$$

*where $\Phi_{\mathbf{R}}(\tau_1, \tau_2)$ is the perturbed problem defined in (23). Furthermore, $\mathbf{A} := [\mathbf{a}_1, \mathbf{a}_2, \ldots, \mathbf{a}_m]^T$ and $\mathbf{B} := [\mathbf{b}_1, \mathbf{b}_2, \ldots, \mathbf{b}_m]^T$ where $\mathbf{a}_i := \sigma(\mathbf{F}\mathbf{x}_i)$ and $\mathbf{b}_i := \hat{\sigma}(\mathbf{F}\mathbf{x}_i)$ for all $i \in \{1, 2, \ldots, m\}$.*

*Proof.* We start by writing $\mathcal{G}_\sigma$ as follows

$$\mathcal{G}_\sigma := \underset{(\mathbf{x},y)}{\mathbb{E}} (y - \hat{\boldsymbol{\omega}}_\sigma^T \sigma(\mathbf{F}\mathbf{x}))^2, \tag{121}$$

$$= \mathbb{E}[y^2] - 2\hat{\boldsymbol{\omega}}_\sigma^T \mathbb{E}[\sigma(\mathbf{F}\mathbf{x})y] + \hat{\boldsymbol{\omega}}_\sigma^T \mathbb{E}[\sigma(\mathbf{F}\mathbf{x})\sigma(\mathbf{F}\mathbf{x})^T]\hat{\boldsymbol{\omega}}_\sigma, \tag{122}$$

$$= \mathbb{E}[y^2] - 2\hat{\boldsymbol{\omega}}_\sigma^T \Sigma_{xy} + \hat{\boldsymbol{\omega}}_\sigma^T \Sigma_x \hat{\boldsymbol{\omega}}_\sigma, \tag{123}$$

$$= \mathbb{E}[y^2] - 2\pi_{\mathbf{A}} + \rho_{\mathbf{A}}, \tag{124}$$

where we define $\pi_{\mathbf{A}} := \hat{\boldsymbol{\omega}}_0^T \Sigma_{xy}$ and $\rho_{\mathbf{A}} := \hat{\boldsymbol{\omega}}_0^T \Sigma_x \hat{\boldsymbol{\omega}}_0$ in the last step. Note that $\hat{\boldsymbol{\omega}}_\sigma = \hat{\boldsymbol{\omega}}_0$ and $\hat{\boldsymbol{\omega}}_{\hat{\sigma}} = \hat{\boldsymbol{\omega}}_m$ by definition (30). Similarly, we can arrive at $\mathcal{G}_{\hat{\sigma}} = \mathbb{E}[y^2] - 2\pi_{\mathbf{B}} + \rho_{\mathbf{B}} + R$ for activation function $\hat{\sigma}$ where $\pi_{\mathbf{B}} := \hat{\boldsymbol{\omega}}_m^T \Sigma_{xy}$ and $\rho_{\mathbf{B}} := \hat{\boldsymbol{\omega}}_m^T \Sigma_x \hat{\boldsymbol{\omega}}_m$. Furthermore, we define $R$ as the remainder term as follows

$$R := \hat{\boldsymbol{\omega}}_m^T \left( \mathbb{E}[\hat{\sigma}(\mathbf{F}\mathbf{x})\hat{\sigma}(\mathbf{F}\mathbf{x})^T] - \Sigma_x \right) \hat{\boldsymbol{\omega}}_m - 2\hat{\boldsymbol{\omega}}_m^T \left( \mathbb{E}[\hat{\sigma}(\mathbf{F}\mathbf{x})y] - \Sigma_{xy} \right), \tag{125}$$

which can be bounded as

$$\mathbb{E}_{\backslash \boldsymbol{F}}|R| \leq \mathbb{E}_{\backslash \boldsymbol{F}} \left( \|\hat{\boldsymbol{\omega}}_m\|^2 \right) \left\| \mathbb{E}[\hat{\sigma}(\mathbf{F}\mathbf{x})\hat{\sigma}(\mathbf{F}\mathbf{x})^T] - \Sigma_x \right\| + 2\mathbb{E}_{\backslash \boldsymbol{F}} \left( \|\hat{\boldsymbol{\omega}}_m\| \right) \left\| \mathbb{E}[\hat{\sigma}(\mathbf{F}\mathbf{x})y] - \Sigma_{xy} \right\|, \tag{126}$$

$$= o(1), \tag{127}$$

where we use a bound on $\mathbb{E}_{\backslash F}[\|\hat{\boldsymbol{\omega}}_m\|^l]$ (Lemma 18) with the equivalence of covariance matrices (10) and the equivalence of cross-covariance vectors (11) to reach the last line. Now, to prove (119), we need to show $\rho_{\mathbf{A}} \xrightarrow{\mathbb{P}} \rho^* = \frac{\partial}{\partial \tau_1} q^*(0,0)$ while $\rho_{\mathbf{B}} \xrightarrow{\mathbb{P}} \rho^*$ and $\pi_{\mathbf{A}} \xrightarrow{\mathbb{P}} \pi^* = \frac{\partial}{\partial \tau_2} q^*(0,0)$ while $\pi_{\mathbf{B}} \xrightarrow{\mathbb{P}} \pi^*$. We show $\rho_{\mathbf{A}} \xrightarrow{\mathbb{P}} \rho^*$ in the following while the rest can be shown analogously. First, observe that, for any $\tau_1, \tau_2 \in (-\tau_*, \tau_*)$,

$$\Phi_{\mathbf{A}}(\tau_1, \tau_2) \leq \Phi_{\mathbf{A}}(0,0) + \tau_1 k \hat{\boldsymbol{\omega}}_0^T \Sigma_x \hat{\boldsymbol{\omega}}_0 + \tau_2 k \hat{\boldsymbol{\omega}}_0^T \Sigma_{xy}, \tag{128}$$

which we arrive at by using the definition of the perturbed optimization (23). Then, for any $\tau \in (0, \tau_*)$,

$$\frac{\Phi_{\mathbf{A}}(\tau, 0) - \Phi_{\mathbf{A}}(0,0)}{k\tau} \leq \rho_{\mathbf{A}} \leq \frac{\Phi_{\mathbf{A}}(-\tau, 0) - \Phi_{\mathbf{A}}(0,0)}{-k\tau}. \tag{129}$$

For a fixed $\epsilon > 0$, there exists $\delta > 0$ such that

$$\left| \frac{q^*(\delta, 0) - q^*(0,0)}{\delta} - \rho^* \right| \leq \epsilon/3, \tag{130}$$

due to assumption (A.8) in Appendix A. Focusing on the inequality on the left in (129), we get:

$$\mathbb{P}(\rho_{\mathbf{A}} - \rho^* < -\epsilon) \leq \mathbb{P}\left( \frac{\Phi_{\mathbf{A}}(\delta, 0) - \Phi_{\mathbf{A}}(0,0)}{k\delta} - \rho^* < -\epsilon \right), \tag{131}$$

$$\leq \mathbb{P}(|\Phi_{\mathbf{A}}(\delta, 0)/k - q^*(\delta, 0)| > \delta\epsilon/3) + \mathbb{P}(|\Phi_{\mathbf{A}}(0,0)/k - q^*(0,0)| > \delta\epsilon/3), \tag{132}$$

$$= 0, \tag{133}$$

where we use (130) in the middle line and $\Phi_{\mathbf{A}}(\tau_1, \tau_2)/k \xrightarrow{\mathbb{P}} q^*(\tau_1, \tau_2)$ is used in the last line. Applying the same steps to the inequality on the right in (129), we arrive at $\mathbb{P}(\rho_{\mathbf{A}} - \rho^* > \epsilon) = 0$. Therefore, we conclude that $\rho_{\mathbf{A}} \xrightarrow{\mathbb{P}} \rho^*$. Using the same reasoning that we used to show $\rho_{\mathbf{A}} \xrightarrow{\mathbb{P}} \rho^*$, we may also show $\rho_{\mathbf{B}} \xrightarrow{\mathbb{P}} \rho^*$, $\pi_{\mathbf{A}} \xrightarrow{\mathbb{P}} \pi^*$, and $\pi_{\mathbf{B}} \xrightarrow{\mathbb{P}} \pi^*$ but we omit them here. Since we have $\rho_{\mathbf{A}} \xrightarrow{\mathbb{P}} \rho^*$, $\rho_{\mathbf{B}} \xrightarrow{\mathbb{P}} \rho^*$, $\pi_{\mathbf{A}} \xrightarrow{\mathbb{P}} \pi^*$, and $\pi_{\mathbf{B}} \xrightarrow{\mathbb{P}} \pi^*$, we reach (119) by using (124). □

**Lemma 11.** *Suppose the setting in Appendix A. Then, for any $\phi \in \mathbb{R}$,*

$$\frac{\Phi_{\mathbf{A}}(\tau_1, \tau_2)}{k} \xrightarrow{\mathbb{P}} \phi \quad \text{if and only if} \quad \frac{\Phi_{\mathbf{B}}(\tau_1, \tau_2)}{k} \xrightarrow{\mathbb{P}} \phi, \tag{134}$$

*if the following holds:*

$$\left| \mathbb{E}\, \varphi\left( \frac{1}{k}\Phi_{\mathbf{A}} \right) - \mathbb{E}\, \varphi\left( \frac{1}{k}\Phi_{\mathbf{B}} \right) \right| \leq \max \{\|\varphi\|_\infty, \|\varphi'\|_\infty, \|\varphi''\|_\infty\} \kappa_k, \tag{135}$$

*for some $\kappa_k = o(1)$ and every bounded test function $\varphi(x)$ which has bounded first two derivatives.*

*Proof.* The proof is adapted from Section II-D in (Hu & Lu, 2023). We need to select an appropriate test function to start the proof. For any fixed $\epsilon > 0$ and $\phi$, let

$$\varphi_\epsilon(x) := \mathbb{1}_{|x| \geq 3\epsilon/2} * \zeta_{\epsilon/2}(x - \phi), \tag{136}$$

where $*$ denotes the convolution operation and $\zeta_\epsilon(x) := \epsilon^{-1} \zeta(x/\epsilon)$ is a scaled mollifier. We define a standard mollifier as:

$$\zeta(x) := \begin{cases} c e^{-1/(1-x^2)}, & \text{if } |x| < 1, \\ 0, & \text{otherwise,} \end{cases} \tag{137}$$

for some constant $c$ ensuring $\int_{\mathbb{R}} \zeta(x)dx = 1$. We may easily verify that $\|\varphi_\epsilon'(x)\|_\infty < C/\epsilon$ and $\|\varphi_\epsilon''(x)\|_\infty < C/\epsilon^2$ for some constant $C$. Furthermore,

$$\mathbb{1}_{|x-\phi| \geq 2\epsilon} \leq \varphi_\epsilon(x) \leq \mathbb{1}_{|x-\phi| \geq \epsilon}. \tag{138}$$

Let $x = \Phi_{\mathbf{A}}/k$ in (138) and take the expectation:

$$\mathbb{P}(|\Phi_{\mathbf{A}}/k - \phi| \geq 2\epsilon) \leq \mathbb{E}[\varphi_\epsilon(\Phi_{\mathbf{A}}/k)]. \tag{139}$$

Similarly, let $x = \Phi_{\mathbf{B}}/k$ in (138) and take the expectation:

$$\mathbb{E}[\varphi_\epsilon(\Phi_{\mathbf{B}}/k)] \leq \mathbb{P}(|\Phi_{\mathbf{B}}/k - \phi| \geq \epsilon). \tag{140}$$

Using (135) together with the last two equations, we arrive at

$$\mathbb{P}(|\Phi_{\mathbf{A}}/k - \phi| \geq 2\epsilon) \leq \mathbb{P}(|\Phi_{\mathbf{B}}/k - \phi| \geq \epsilon) + \max\left\{C, C/\epsilon, C/\epsilon^2\right\}\kappa_k, \tag{141}$$

which leads to

$$\mathbb{P}(|\Phi_{\mathbf{A}}/k - \phi| \geq 2\epsilon) \leq \mathbb{P}(|\Phi_{\mathbf{B}}/k - \phi| \geq \epsilon) + \frac{C\kappa_k}{\epsilon^2}, \tag{142}$$

for $\epsilon \in (0,1)$ satisfying $\kappa_k/\epsilon^2 = o(1)$. Applying the same steps after switching $\mathbf{A}$ with $\mathbf{B}$, we get

$$\mathbb{P}(|\Phi_{\mathbf{B}}/k - \phi| \geq 2\epsilon) \leq \mathbb{P}(|\Phi_{\mathbf{A}}/k - \phi| \geq \epsilon) + \frac{C\kappa_k}{\epsilon^2}. \tag{143}$$

Combining the last two results and letting $\epsilon \to 0^+$ with $\kappa_k/\epsilon^2 = o(1)$, we reach (134). $\qquad\square$

**Lemma 12.** *Suppose the definitions in Appendix A. Then, the following holds:*

$$\Phi_{t-1} = \Psi_t(\mathbf{a}_t), \qquad and \qquad \Phi_t = \Psi_t(\mathbf{b}_t). \tag{144}$$

*Proof.* We apply Taylor expansion to $\Phi_{t-1}$ around $\Phi_{\backslash t}$:

$$\Phi_{t-1} = \Phi_{\backslash t} + \min_{\boldsymbol{\omega} \in \mathbb{R}^k}\left\{(\boldsymbol{\omega} - \hat{\boldsymbol{\omega}}_{\backslash t})^T \boldsymbol{d}_{\backslash t} + \frac{1}{2}\left(\boldsymbol{\omega} - \hat{\boldsymbol{\omega}}_{\backslash t}\right)^T \boldsymbol{H}_{\backslash t}\left(\boldsymbol{\omega} - \hat{\boldsymbol{\omega}}_{\backslash t}\right) + \left(\boldsymbol{\omega}^T \mathbf{a}_t - y_t\right)^2\right\}, \tag{145}$$

where $\boldsymbol{d}_{\backslash t}$ and $\boldsymbol{H}_{\backslash t}$ are respectively gradient vector and Hession matrix of $\Phi_{\backslash t}$ evaluated at $\hat{\boldsymbol{\omega}}_{\backslash t}$. Note that we do not need higher order terms since $\Phi_{\backslash t}$ only involves quadratic terms, which means higher order terms are equal to $0$. Furthermore, the gradient vector $\boldsymbol{d}_{\backslash t}$ is equal to $0$ due to the first-order optimality condition in (34). This leads us to:

$$\Phi_{t-1} = \Phi_{\backslash t} + \min_{\boldsymbol{\omega} \in \mathbb{R}^k}\left\{\frac{1}{2}\left(\boldsymbol{\omega} - \hat{\boldsymbol{\omega}}_{\backslash t}\right)^T \boldsymbol{H}_{\backslash t}\left(\boldsymbol{\omega} - \hat{\boldsymbol{\omega}}_{\backslash t}\right) + \left(\boldsymbol{\omega}^T \mathbf{a}_t - y_t\right)^2\right\} = \Psi_t(\mathbf{a}_t). \tag{146}$$

Similarly, one can show $\Phi_t = \Psi_t(\mathbf{b}_t)$ using the same reasoning, which is omitted here. $\qquad\square$

**Lemma 13.** *Suppose the definitions in Appendix A. Then, the following holds:*

$$\max\left\{\mathbb{E}_{\backslash \boldsymbol{F}}\left(\Psi_t\left(\boldsymbol{b}_t\right) - \Phi_{\backslash t}\right)^2, \mathbb{E}_{\backslash \boldsymbol{F}}\left(\Psi_t\left(\boldsymbol{a}_t\right) - \Phi_{\backslash t}\right)^2\right\} \leq polylog\ k. \tag{147}$$

*Proof.* Let $\mathbf{r}_t = \mathbf{a}_t$ or $\mathbf{r}_t = \mathbf{b}_t$. Then,

$$\mathbb{E}_{\backslash \boldsymbol{F}}\left(\Psi_t\left(\mathbf{r}_t\right) - \Phi_{\backslash t}\right)^2 \leq \mathbb{E}_{\backslash \boldsymbol{F}}\left(\hat{\boldsymbol{\omega}}_{\backslash t}^T \mathbf{r}_t - y_t\right)^4, \tag{148}$$

$$\leq \sqrt{\mathbb{E}_{\backslash \boldsymbol{F}}\left(\left|\hat{\boldsymbol{\omega}}_{\backslash t}^T \mathbf{r}_t\right| + 1\right)^8}\ polylog\ k, \tag{149}$$

$$\leq Q\left(\|\hat{\boldsymbol{\omega}}_{\backslash t}\|\right)\|\hat{\mathbf{F}}\|^4\ polylog\ k, \tag{150}$$

$$\leq k^{1-\epsilon}\ polylog\ k \tag{151}$$

where $Q(x)$ is some finite-degree polynomial, $\hat{\mathbf{F}} := \mathbf{F}(\mathbf{I}_n + (\sqrt{1+\theta} - 1)\boldsymbol{\gamma}\boldsymbol{\gamma}^T)$ and some $\epsilon > 0$ satisfying $\theta^2 \leq n^{1-\epsilon}$. We use (43) in the first step. Then, we reach the second line using Lemma 14. In the third step, we use Lemma 15. Finally, we use (58) and Lemma 18 to conclude the proof. $\qquad\square$

**Lemma 14.** *Suppose the definitions in Appendix A. Then, the following holds:*

$$\mathbb{E}_{\backslash \boldsymbol{F}}\left(\hat{\boldsymbol{\omega}}_{\backslash t}^T \mathbf{r} - y_t\right)^{2l} \leq \sqrt{\mathbb{E}_{\backslash \boldsymbol{F}}\left(\left|\hat{\boldsymbol{\omega}}_{\backslash t}^T \mathbf{r}\right| + 1\right)^{4l}}\ polylog\ k, \tag{152}$$

*for $l \in \mathbb{Z}^+$.*

*Proof.* Let $y_t := \sigma_*(s_t)$ where $s_t := \frac{\boldsymbol{\xi}^T \mathbf{x}_t}{\sqrt{1+\theta\alpha^2}}$. Then, for any $x \in \mathbb{R}$,

$$(x - y_t)^2 \leq |x|^2 + 2|x||y_t| + |y_t|^2, \tag{153}$$

$$\leq |x|^2 + 2|x|C \left(|s_t|^K + 1\right) + C^2 \left(|s_t|^K + 1\right)^2, \tag{154}$$

$$\leq \hat{C}(|x| + 1)^2 \left(|s_t|^K + 1\right)^2, \tag{155}$$

for some $C, \hat{C} > 0$. To reach the second line, we use $|y_t| \leq C \left(|s_t|^K + 1\right)$ due to assumption (A.3). Using this result, we get:

$$\mathbb{E}_{\backslash \boldsymbol{F}} \left(\hat{\boldsymbol{\omega}}_{\backslash t}^T \mathbf{r} - y_t\right)^{2l} \leq \hat{C}^l \mathbb{E}_{\backslash \boldsymbol{F}} \left[\left(\left|\hat{\boldsymbol{\omega}}_{\backslash t}^T \mathbf{r}\right| + 1\right)^{2l} \left(|s_t|^K + 1\right)^{2l}\right], \tag{156}$$

$$\leq \hat{C}^l \sqrt{\mathbb{E}_{\backslash \boldsymbol{F}} \left(\left|\hat{\boldsymbol{\omega}}_{\backslash t}^T \mathbf{r}\right| + 1\right)^{4l}} \sqrt{\mathbb{E}_{\backslash \boldsymbol{F}} \left(|s_t|^K + 1\right)^{4l}}. \tag{157}$$

$s_t \sim \mathcal{N}(0, 1)$ so $\mathbb{P}(|s_t| > \epsilon) \leq 2e^{-\epsilon^2/2}$ (Vershynin, 2018, Eq. (2.10)). Therefore, we can reach (152) by using moment bounds for $|s_t|$ from Lemma 16. $\qquad\square$

**Lemma 15.** *Suppose assumptions (A.1)-(A.6). Let $\mathbf{a} := \sigma(\mathbf{F}\mathbf{x})$. Furthermore, let $\hat{\mathbf{F}} := \mathbf{F}(\mathbf{I}_n + (\sqrt{1+\theta} - 1)\boldsymbol{\gamma}\boldsymbol{\gamma}^T)$. Then, there exist some $c, C > 0$ such that the following holds*

$$\mathbb{P}_{\backslash \mathbf{F}} \left(\left|\boldsymbol{\omega}^T \mathbf{a}\right| \geq \epsilon\right) \leq 2e^{\frac{-\epsilon^2}{c\|\boldsymbol{\omega}\|^2 \|\hat{\mathbf{F}}\|^2 \, polylog \, k}}, \tag{158}$$

$$\mathbb{E}_{\backslash \mathbf{F}} \left(\left|\boldsymbol{\omega}^T \mathbf{a}\right|^l\right) \leq (l!) \left(C\|\boldsymbol{\omega}\|^2 \|\hat{\mathbf{F}}\|^2 \, polylog \, k\right)^{l/2}, \tag{159}$$

*for any $l \in \mathbb{Z}^+$, any fixed $\boldsymbol{\omega} \in \mathbb{R}^k$ and $\epsilon \geq 0$.*

*Proof.* Let $\mathbf{g} \sim \mathcal{N}(0, \mathbf{I}_k)$. One can verify that the function $f(\mathbf{g}) := \boldsymbol{\omega}^T \mathbf{a} = \boldsymbol{\omega}^T \sigma(\hat{\mathbf{F}}\mathbf{g})$ is $(\|\sigma'\|_\infty \|\boldsymbol{\omega}\| \|\hat{\mathbf{F}}\|)$-Lipschitz continuous. Furthermore, $\mathbb{E}_{\backslash \mathbf{F}} \left(\boldsymbol{\omega}^T \mathbf{a}\right) = 0$ due to $\sigma(x)$ being an odd function. Furthermore, $\sigma'(z) \leq polylog \, k$ with high probability for $z \sim \mathcal{N}(0, \hat{C})$ with all $\hat{C} > 0$, which allows us to continue with $\|\sigma'\|_\infty \leq polylog \, k$ by using a truncation argument (similar to Lemma 19). Therefore, we can reach (158)-(159) using Lemma 16. $\qquad\square$

**Lemma 16** (Concentration of Lipschitz Functions - (Hu & Lu, 2023, Theorem 3))**.** *Consider $\mathbf{g} \sim \mathcal{N}(0, \mathbf{I}_k)$. Then, for any $\kappa$-Lipschitz function $f : \mathbb{R}^k \to \mathbb{R}$ and $\epsilon \geq 0$,*

$$\mathbb{P}(|f(\mathbf{g}) - \mathbb{E}[f(\mathbf{g})]| \geq \epsilon) \leq 2e^{-\epsilon^2/(4\kappa^2)}. \tag{160}$$

*Furthermore, let $x$ be a random variable. If $x$ satisfying $\mathbb{P}(|x| > v) \leq ce^{-Cv}$ for some $C, c > 0$,*

$$\mathbb{E}\left[|x|^l\right] \leq clC^{-l} \int_0^\infty e^{-v} v^{l-1} dv = c(l!)C^{-l} \quad \text{for any } l \in \mathbb{Z}^+. \tag{161}$$

*Similarly, if $x$ satisfying $\mathbb{P}(|x| > v) \leq ce^{-Cv^2}$ for some $C, c > 0$,*

$$\mathbb{E}\left[|x|^l\right] \leq 2c(l!)C^{-l/2} \quad \text{for any } l \in \mathbb{Z}^+. \tag{162}$$

*Proof.* See (Talagrand, 2010, Theorem 1.3.4). $\qquad\square$

**Lemma 17.** *Suppose the definitions in Appendix A. Let $\epsilon > 0$ be a constant satisfying $\theta \leq n^{1/2-\epsilon/2}$. For $\mathbf{r}_t = \mathbf{a}_t$ or $\mathbf{r}_t = \mathbf{b}_t$, we have*

$$\mathbb{P}(|\nu_t(\mathbf{r}_t) - \mathbb{E}_t[\nu_t(\mathbf{r}_t)]| > \delta) \leq ce^{-Ck^\epsilon \delta^2/polylog \, k}, \tag{163}$$

*for some $c, C > 0$. Furthermore,*

$$\mathbb{E}_t[|\nu_t(\mathbf{r}_t) - \mathbb{E}_t[\nu_t(\mathbf{r}_t)]|^l] \leq 2c(l!)(Ck^\epsilon/polylog \, k)^{-l/2}, \tag{164}$$

*for some $c, C > 0$ and any $l \in \mathbb{Z}^+$. Finally,*

$$\left|\mathbb{E}_t \nu_t(\mathbf{a}_t) - \mathbb{E}_t[\nu_t(\mathbf{b}_t)]\right| = o(1/polylog \, k). \tag{165}$$

*Proof.* Consider

$$\nu_t(\mathbf{r}_t) = \mathbf{r}_t^T \boldsymbol{H}_{\backslash t}^{-1} \mathbf{r}_t = \frac{\mathbf{r}_t^T \bar{\boldsymbol{H}}_{\backslash t}^{-1} \mathbf{r}_t}{k}, \tag{166}$$

where $\bar{\boldsymbol{H}}_{\backslash t}$ is defined as follows to take the $1/k$ scaling out

$$\bar{\boldsymbol{H}}_{\backslash t} := \frac{2}{k} \sum_{i=1}^{t-1} \mathbf{b}_i \mathbf{b}_i^T + \frac{2}{k} \sum_{i=t+1}^{m} \mathbf{a}_i \mathbf{a}_i^T + \frac{1}{k} \nabla^2 Q \left( \hat{\boldsymbol{\omega}}_{\backslash t} \right). \tag{167}$$

Then, using $s$-strong convexity of $Q(\boldsymbol{\omega})/m$ due to assumption (A.7), we have $\bar{\boldsymbol{H}}_{\backslash t} \succeq (m/k)s\mathbf{I}_k$, which lead to $\|\bar{\boldsymbol{H}}_{\backslash t}^{-1}\| \le (k/m)(1/s)$. Furthermore, we can rewrite $\mathbf{a}_i = \sigma(\mathbf{F}\mathbf{x}_i)$ equivalently as $\sigma(\hat{\mathbf{F}}\mathbf{g})$ for $\hat{\mathbf{F}} := \mathbf{F}(\mathbf{I}_n + (\sqrt{1+\theta}-1)\boldsymbol{\gamma}\boldsymbol{\gamma}^T)$ and $\mathbf{g} \sim \mathcal{N}(0, \mathbf{I}_n)$. We have $\|\hat{\mathbf{F}}\| \le k^{1/4-\epsilon/4}$ polylog $k$ (for some $\epsilon > 0$ satisfying $\theta \le n^{1/2-\epsilon/2}$), which is due to (58). We also have $\|\sigma'\|_\infty \le$ polylog $k$ by using a truncation argument (similar to Lemma 19) as mentioned in the proof of Lemma 15. Then, we use (Louart et al., 2018, Lemma 1) with the bounds for $\|\hat{\mathbf{F}}\|$ and $\|\bar{\boldsymbol{H}}_{\backslash t}^{-1}\|$ to reach (163). Finally, (164) can be obtained using Lemma 16. To show (165), we proceed as

$$\left| \mathbb{E}_t \nu_t(\mathbf{a}_t) - \mathbb{E}_t[\nu_t(\mathbf{b}_t)] \right| = \frac{1}{k} \left| \mathbb{E}_t \left[ \mathbf{a}_t^T \bar{\boldsymbol{H}}_{\backslash t}^{-1} \mathbf{a}_t \right] - \mathbb{E}_t \left[ \mathbf{b}_t^T \bar{\boldsymbol{H}}_{\backslash t}^{-1} \mathbf{b}_t \right] \right|, \tag{168}$$

$$= \frac{1}{k} \left| \text{Tr} \left( \bar{\boldsymbol{H}}_{\backslash t}^{-1} \left( \mathbb{E}_t \left[ \mathbf{a}_t \mathbf{a}_t^T \right] - \mathbb{E}_t \left[ \mathbf{b}_t \mathbf{b}_t^T \right] \right) \right) \right|, \tag{169}$$

$$= o(1/ \text{polylog } k), \tag{170}$$

where Tr denotes the trace while we use $\|\bar{\boldsymbol{H}}_{\backslash t}^{-1}\| \le (k/m)(1/s)$ and (10) to reach the last line. □

**Lemma 18.** *Suppose the definitions in Appendix A. Then, the following holds:*

$$\mathbb{E}_{\backslash \mathbf{F}} \|\hat{\boldsymbol{\omega}}_t\|^l \le polylog\ k, \tag{171}$$

$$\mathbb{E}_{\backslash \mathbf{F}} \|\hat{\boldsymbol{\omega}}_{\backslash t}\|^l \le polylog\ k, \tag{172}$$

*for any $l \in \mathbb{Z}^+$ and any $t \in \{0, \dots, m\}$.*

*Proof.* We show (171) here while (172) can be proved similarly. Let $\mathbf{r}_i := \mathbf{b}_i$ for $i \in \{1, \dots, k\}$ and $\mathbf{r}_i := \mathbf{a}_i$ for $t \in \{k+1, \dots, m\}$. Furthermore, let $Q(\boldsymbol{\omega}) := m\lambda\|\boldsymbol{\omega}\|_2^2 + \tau_1 k \boldsymbol{\omega}^T \Sigma_x \boldsymbol{\omega} + \tau_2 k \boldsymbol{\omega}^T \Sigma_{xy}$. Then,

$$\frac{1}{m} Q(\hat{\boldsymbol{\omega}}_t) \le \frac{1}{m} \sum_{i=1}^{m} \left( \hat{\boldsymbol{\omega}}_t^T \mathbf{r}_i \right)^2 - y_i \right)^2 + \frac{1}{m} Q(\hat{\boldsymbol{\omega}}_t) \le \frac{1}{m} \sum_{i=1}^{m} y_i^2 + \frac{1}{m} Q(\mathbf{0}), \tag{173}$$

since $\hat{\boldsymbol{\omega}}_t$ is the optimal solution defined in (30). Furthermore, $Q(\boldsymbol{\omega})/m$ is $s$-strongly convex by assumption (A.7) in Appendix A. Using this fact, we reach

$$\frac{1}{m} \mathbf{Q}(\hat{\boldsymbol{\omega}}_t) \ge \frac{1}{m} Q(\mathbf{0}) + \frac{1}{m} \nabla Q(\mathbf{0})^T \hat{\boldsymbol{\omega}}_t + (s/2)\|\hat{\boldsymbol{\omega}}_t\|^2, \tag{174}$$

$$\ge \frac{1}{m} Q(\mathbf{0}) - \frac{1}{m} \|\nabla Q(\mathbf{0})\| \|\hat{\boldsymbol{\omega}}_t\| + (s/2)\|\hat{\boldsymbol{\omega}}_t\|^2. \tag{175}$$

We may then combine the found lower and upper bounds for $Q(\hat{\boldsymbol{\omega}}_t)$

$$(s/2)\|\hat{\boldsymbol{\omega}}_t\|^2 - \frac{1}{m} \|\nabla Q(\mathbf{0})\| \|\hat{\boldsymbol{\omega}}_t\| \le \frac{1}{m} \sum_{i=1}^{m} y_i^2, \tag{176}$$

which leads to

$$\|\hat{\boldsymbol{\omega}}_t\| \le \frac{\|\nabla Q(\mathbf{0})\|/m + \sqrt{\|\nabla Q(\mathbf{0})\|^2/m^2 + 2(s/m) \sum_{i=1}^{m} y_i^2}}{s}, \tag{177}$$

$$\le \frac{2\|\nabla Q(\mathbf{0})\|/m + \sqrt{2(s/m) \sum_{i=1}^{m} y_i^2}}{s}, \tag{178}$$

$$\le \left( \frac{1}{m} \sum_{i=1}^{m} y_i^2 \right)^{1/2} \text{polylog } k, \tag{179}$$

where we use $\|\nabla Q(\mathbf{0})\|/m = |\tau_2|(k/m)\|\Sigma_{xy}\| \leq$ polylog $k$ since we have $(k/m) \in (0, \infty)$ by assumption (A.4) in Section 3, $\tau_2 = \mathcal{O}(1/\sqrt{k})$ by assumption (A.7) in Appendix A and one can easily show $\|\Sigma_{xy}\| \leq \sqrt{k}$ polylog $k$ using (112). Then, we consider $l$-th power of $\|\hat{\boldsymbol{\omega}}_t\|$ as

$$\|\hat{\boldsymbol{\omega}}_t\|^l \leq \left(\frac{1}{m}\sum_{i=1}^{m} y_i^2\right)^{l/2} \text{polylog } k. \tag{180}$$

Let $y_i := \sigma_*(s_i)$ where $s_i := \frac{\boldsymbol{\xi}^T \mathbf{x}_i}{\sqrt{1+\theta\alpha^2}}$. Using $|y_i| \leq C\left(|s_i|^K + 1\right)$ due to assumption (A.3), we get

$$\mathbb{E}_{\backslash\mathbf{F}}\|\hat{\boldsymbol{\omega}}_t\|^l \leq \mathbb{E}_{\backslash\mathbf{F}}\left(\frac{1}{m}\sum_{i=1}^{m}\left(|s_i|^K + 1\right)^2\right)^{l/2} \text{polylog } k. \tag{181}$$

Finally, $s_i \sim \mathcal{N}(0, 1)$ so we reach (171) by using moment bounds for $|s_i|$ from Lemma 16. Similarly, (172) can shown using analogous steps. $\square$

**Lemma 19.** *Let $f : \mathbb{R} \to \mathbb{R}$ be a function satisfying $|f(x)| \leq C(1 + |x|^K)$ for all $x \in \mathbb{R}$ and some constants $C > 0, K \in \mathbb{Z}^+$. Furthermore, define a truncated version of $f$ as*

$$f_\epsilon(x) := \begin{cases} f(x), & \text{if } |x| < \epsilon, \\ 0, & \text{otherwise,} \end{cases} \tag{182}$$

*for $\epsilon > 0$. If $\epsilon = 2logk$, the following holds for any constant $a > 0$ and $z \sim \mathcal{N}(0, 1)$,*

$$\mathbb{E}_z|f(az) - f_\epsilon(az)| = \frac{polylog\ k}{k^{logk}}. \tag{183}$$

*Proof.* We start by using $1 = \mathbb{1}_{|z|<\epsilon} + \mathbb{1}_{|z|\geq\epsilon}$ as follows

$$\mathbb{E}_z|f(az) - f_\epsilon(az)| \leq \mathbb{E}_z[|f(az) - f_\epsilon(az)|(\mathbb{1}_{|z|<\epsilon} + \mathbb{1}_{|z|\geq\epsilon})], \tag{184}$$

$$= \mathbb{E}_z[|f(az) - f_\epsilon(az)|\mathbb{1}_{|z|\geq\epsilon}], \tag{185}$$

$$= \mathbb{E}_z[|f(az)|\mathbb{1}_{|z|\geq\epsilon}] \tag{186}$$

$$\overset{(a)}{\leq} \sqrt{\mathbb{E}_z[f(az)^2]}\sqrt{\mathbb{E}_z[(\mathbb{1}_{|z|\geq\epsilon})^2]}, \tag{187}$$

$$= \sqrt{\mathbb{E}_z[f(az)^2]}\sqrt{\mathbb{P}(|z| \geq \epsilon)}, \tag{188}$$

$$\overset{(b)}{\leq} C\sqrt{\mathbb{E}_z[(a^{2K}z^{2K} + 2a^K|z|^K + 1)]}\sqrt{2e^{-\epsilon^2/2}}, \tag{189}$$

$$\overset{(c)}{\leq} \frac{\text{polylog } k}{e^{log^2(k)}} = \frac{\text{polylog } k}{k^{logk}}, \tag{190}$$

where we use Cauchy–Schwarz inequality to reach (a) while (b) is due to $|f(x)| \leq C(1+|x|^K)$ and $\mathbb{P}(|z| > \epsilon) \leq 2e^{-\epsilon^2/2}$ (Vershynin, 2018, Eq. (2.10)). Finally, we reach (c) by using moment bounds for $|z|$ from Lemma 16. $\square$

**Lemma 20.** *Let $H_i(x)$ be the probabilist's $i$-th Hermite polynomial and $z_1, z_2 \overset{i.i.d}{\sim} \mathcal{N}(0, 1)$. Then, for any $\rho \in [0, 1]$ and any $c \geq |\rho|$,*

$$\mathbb{E}_{z_1,z_2}[H_i(\rho z_1 + \sqrt{c^2 - \rho^2}z_2)H_j(z_1)] = (i!)\rho^i\delta_{i,j}, \tag{191}$$

*where $\delta_{ij}$ is Kronecker delta function.*

*Proof.* Let $a = (\rho/c)z_1 + \sqrt{1 - \rho^2/c^2}z_2$ and $b = z_1$. Observe that $(a, b)$ is jointly Gaussian with $\mathbb{E}[a^2] = \mathbb{E}[b^2] = 1$ and $\mathbb{E}[ab] = (\rho/c)$. Furthermore, let $p(a, b)$ be the joint probability density function of $(a, b)$ while $p(a)$ and $p(b)$ denoting the individual probability density functions. Using Mehler's formula (Kibble, 1945; Mehler, 1866), we get

$$p(a, b) = p(a)p(b)\sum_{l=0}^{\infty}\frac{(\rho/c)^l}{l!}H_l(a)H_l(b). \tag{192}$$

Then, it follows

$$\mathbb{E}_{z_1,z_2}[H_i(\rho z_1 + \sqrt{c^2 - \rho^2} z_2) H_j(z_1)] \tag{193}$$

$$= \mathbb{E}_{a,b}[H_i(ca) H_j(b)], \tag{194}$$

$$= \int_{-\infty}^{\infty} \int_{-\infty}^{\infty} H_i(ca) H_j(b) p(a,b) da db, \tag{195}$$

$$= \int_{-\infty}^{\infty} \int_{-\infty}^{\infty} H_i(ca) H_j(b) p(a) p(b) \sum_{l=0}^{\infty} \frac{(\rho/c)^l}{l!} H_l(a) H_l(b) da db, \tag{196}$$

$$= \sum_{l=0}^{\infty} \frac{(\rho/c)^l}{l!} \int_{-\infty}^{\infty} H_i(ca) H_l(a) p(a) da \int_{-\infty}^{\infty} H_j(b) H_l(b) p(b) db, \tag{197}$$

$$= \sum_{l=0}^{\infty} \frac{(\rho/c)^l}{l!} \mathbb{E}_a[H_i(ca) H_l(a)] \mathbb{E}_b[H_j(b) H_l(b)], \tag{198}$$

$$= \sum_{l=0}^{\infty} \frac{(\rho/c)^l}{l!} (i!)(j!)(c^i) \delta_{i,l} \delta_{j,l}, \tag{199}$$

$$= (i!) \rho^i \delta_{ij}, \tag{200}$$

where we use Lemma 21 to reach the line before the last one. □

**Lemma 21.** *Let $H_i(x)$ be the probabilist's $i$-th Hermite polynomial. Also, let $z \sim \mathcal{N}(0,1)$. Then, for any $c \in \mathbb{R}$,*

$$\mathbb{E}[H_i(cz) H_j(z)] = (i!)(c^i) \delta_{ij}. \tag{201}$$

*Proof.* It is known that $\mathbb{E}[H_i(z) H_j(z)] = (i!) \delta_{ij}$ (O'Donnell, 2014, Chapter 11.2). Here, we extend it to our case. For $i \neq j$, $\mathbb{E}[H_i(cz) H_j(z)] = 0$ by orthogonality. Let $a_l$ be the coefficient of $x^l$ in $H_i(x)$. Then, we have

$$\mathbb{E}[H_i(cz) H_i(z)] = \sum_{l=0}^{i} c^l \, \mathbb{E}[a_l z^l H_i(z)], \tag{202}$$

$$\overset{(a)}{=} c^i \, \mathbb{E}[a_l z^i H_i(z)], \tag{203}$$

$$\overset{(b)}{=} c^i \, \mathbb{E}[H_i(z) H_i(z)], \tag{204}$$

$$\overset{(c)}{=} (i!)(c^i), \tag{205}$$

where we use $\mathbb{E}[z^l H_i(z)] = \delta_{lj}$ due to orthogonality to reach (a) and (b) while (c) is because of $\mathbb{E}[H_i(z) H_i(z)] = (i!)$. Combining $i = j$ and $i \neq j$ cases, we get (201). □

## F    PLOTS FOR TRAINING ERRORS

For the sake of completeness, here, we provide the supplementary training error plots corresponding to the setting of Figure 1b and Figure 2 in Section 4.

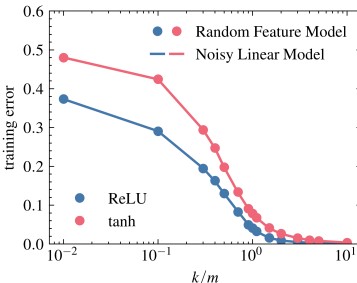

Figure 5: Training errors for the misaligned case (the setting in Figure 1b)

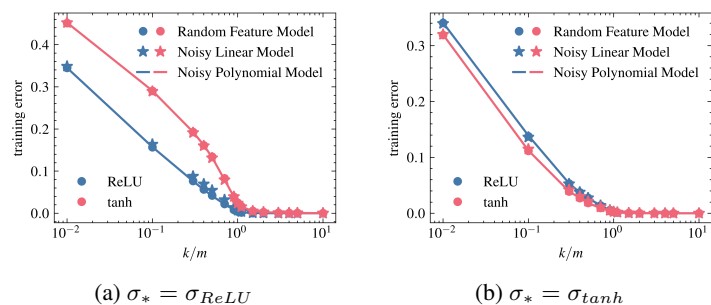

Figure 6: Training errors for the aligned case (the setting in Figure 2).

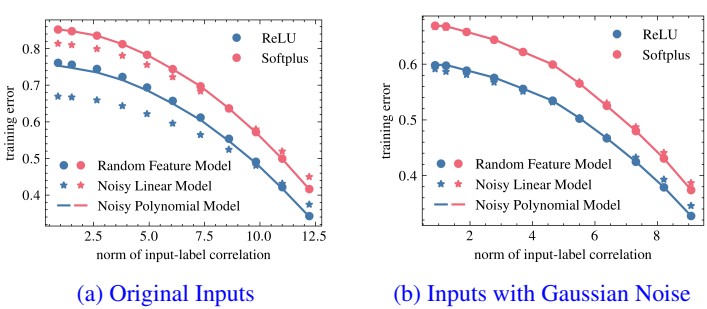

Figure 7: Training errors for CIFAR-10 experiments (the setting in Figure 4).

## G   DETAILS FOR CIFAR-10 EXPERIMENTS

In our experiments, we focus on binary classification between airplanes and automobiles using the CIFAR-10 dataset (Krizhevsky et al., 2009) to demonstrate how our findings translate to real-world applications. We randomly select 2,000 samples from each class for training, while a separate set of 2,500 samples (distinct from the training set) is used to calculate the test (generalization) error. To prepare the input samples, we normalize the pixel values to achieve zero mean and unit variance for each color channel (R, G, B). Specifically, the pixel values are first scaled to the range [0, 1] by dividing by 255, followed by subtracting channel-wise means and dividing by their respective standard deviations. The images are then flattened into vectors for input into the model.

For the feature matrix $\mathbf{F}$, we sample entries independently from a normal distribution $\mathcal{N}(0, 1/\text{Tr}(\mathbb{E}[\mathbf{x}\mathbf{x}^T]))$, ensuring that $\mathbb{E}[(\mathbf{f}_i^T\mathbf{x})^2] = 1$ for all $i$, where $\mathbf{f}_i$ denotes the $i$-th row of $\mathbf{F}$.

Our results are presented in two scenarios: first with the original normalized inputs (Figure 4a), and second after adding standard Gaussian noise (variance of one for each feature) to create a more isotropic covariance structure (Figure 4b). This addition of noise allows us to assess the performance equivalence between the linear model and RFM under conditions of weak input-label correlation.

For labels, we use $\{-1, 1\}$ for the first class and the second class, respectively. To control input-label correlation, we introduce a label flipping mechanism with probability $p$, where $p = 0$ corresponds to true labels (maximum correlation) and $p = 0.5$ represents random labels (minimum correlation). By varying $p$ within the range $[0, 0.5]$, we interpolate between these two extremes and analyze how this affects model performance. The training errors for these experiments are illustrated in Figure 7.

