# OpenReview forum: "Random Features Outperform Linear Models: Effect of Strong Input-Label Correlation in Spiked Covariance Data"
_ICLR.cc/2025/Conference — Submitted to ICLR 2025_

### Official Review · Reviewer_vcz1 · 2024-10-24

**Soundness:** 3
**Presentation:** 2
**Contribution:** 2
**Rating:** 6
**Confidence:** 3

**Summary:**

This paper examines the performance of random feature models (RFMs) with anisotropic input data, characterised by spiked covariance. Whilst RFMs and linear models are asymptotically equivalent for isotropic data, RFMs tend to perform better in practical applications – an observation that the authors put down to the structured, anisotropic nature of real world data. They establish a ‘universality theorem’, extending the work of Hu and Lu (2023) to prove that there exists two different activation functions with equivalent performances if the first two statistical moments of $\sigma(\mathbf{F} \mathbf{x},y)$ match. They use this result to show that higher-order polynomial models are equivalent to RFMs for spiked covariance data, and show that in particular limits for this data linear models become sufficient (roughly, when input correlations and covariance spiking becomes small). Input-label correlation is found to be crucial in determining whether RFMs outperform linear models. The authors provide detailed theoretical guarantees and experimental validation on synthetic data.

**Strengths:**

The paper makes solid technical contributions, making a careful detailed case that, for spiked covariance data (Eq. 6), RFMs will perform better except when the spike magnitude, alignment parameter, or cosine similarity between the rows of $\mathbf{F}$ and $\gamma$ are sufficiently small (how should I interpret the last condition?). It provides a nice extension to the work of Hu (2023) beyond the linear regime, and it seems possible that this type of data structure/anisotropy might be responsible for the superior performance of RFMs in the wild. The experiments (especially, measuring generalisation error vs. number of samples, alignment and spike magnitude for different models in Fig. 3) appear to back up the authors’ central claims.

**Weaknesses:**

1. *Applicability to real-world data.* I appreciate that the specific parameterisation of Eq. 6 is needed for analytic tractability, but can the authors provide evidence that this kind of anisotropy in real-world datasets is responsible for the performance gap between RFMs and linear models? I think that the detailed analysis on this toy model is valuable in its own right, and would be of especial interest for readers of e.g. IEEE transactions on information theory (where Hu 2023, which this work builds on, was published). But to be suitable for ICLR, I think it would be helpful to provide some evidence that similar behaviour holds on real-world data. As a straightforward example, the authors could show performance on a real dataset with the polynomial kernel, ablating over different $l$ and inspecting when performance approaches RFM. Do any properties of real-world distributions roughly corresponds to $\theta$ and $\alpha$, spike magnitude and alignment? Can we see a difference in the gap for datasets where these properties differ? Another way to better orient the work in existing ML literature would be to go further explaining the relationship to double descent, beyond the brief empirical observations around line 490, or the relationship (if any) to feature learning in neural networks.
2. *Other points.*
- *Thm 1*. An activation function $\sigma$ and its Hermite expansion $\widehat{\sigma}$ will perform identically if we include every term since the Hermite basis is complete (line 280). Under suitable asymptotic conditions (needed to bound the contribution from higher order terms), it’s not too surprising that it becomes sufficient to consider just the first two terms. This appears to be the central conclusion of Thm. 1. I don’t doubt that the authors’ derivations are correct and can see that this is an essential corollary for the arguments that follow, but I wonder whether emphasising this as a `key advancement of this work’ (line 263) is overstated/misplaces emphasis.
- *Assumption 7*. A.7, which is required to keep the perturbed objective convex (line 742), seems to be essential to Thm. 1. Is there any evidence that this will hold in practice? This seems like an important and nontrivial assumption, needed to make claims about the generalisation error, so I wonder why it is deferred to the appendix. Perhaps it would be better to include it in the main text, and probe whether it holds in practice in the experiments.
- *Odd activation functions*. Assumption A.6. requires that activation functions are odd. The authors empirically show that their arguments hold even when this is relaxed, e.g. for ReLU. Can the authors give some idea of where their arguments break down for non-odd activation functions (I guess around line 1006)? And why doesn’t Fig. 3 include any odd activations, e.g. tanh, given that the proof only holds in this case?

As a stylistic comment, Fig. 2a is difficult to read since the noisy linear and random feature models for $\tanh$ are overlaid. Perhaps this could be presented a little differently to make things clearer.

**Questions:**

1. To what extent would these findings generalise to other anisotropic data settings, and to what extent are they expected to be particular to the spiked covariance model?
2. Are there any practical implications from this work for model selection, given access to a labelled dataset?
3. Given the odd activation function assumption (A.6), do the authors see any empirical difference in results depending on the parity of $\sigma$? As mentioned above, including $\tanh$ in Fig. 3 might help answer this question.

I sincerely thank the authors for their time and efforts. It’s great to see such detailed, careful mathematical work, which is sometimes lacking in the field! If they can convince me of the broader applicability of their findings (especially, relating to real-world datasets) and clarify assumption A.7 I will be happy to raise my score.

---

> ### Author Response · Authors · 2024-11-24
> **Part 1/2**
>
> We thank the reviewer for their thoughtful and constructive feedback. Below, we address each of the points raised.
>
> > Applicability to real-world data $\cdots$  But to be suitable for ICLR, I think it would be helpful to provide some evidence that similar behaviour holds on real-world data $\cdots$ Do any properties of real-world distributions roughly corresponds to $\theta$ and $\alpha$, spike magnitude and alignment? $\cdots$
>
> We appreciate the reviewer’s concern regarding the real-world applicability of our theoretical findings. In response, we have conducted additional experiments on the CIFAR-10 dataset, specifically focusing on the binary classification of automobiles versus airplanes. This experiment involved systematically interpolating between random and true labels to analyze the impact of input-label correlation on model equivalence. Our results confirm a transition between regimes where linear models suffice and those requiring high-order polynomial models for equivalence with Random Feature Models (RFMs). These findings provide empirical evidence that input-label correlation is a crucial factor in real-world scenarios, aligning with our theoretical predictions. We observed that as the correlation increases, the performance of RFMs improves relative to linear models, supporting the hypothesis that anisotropic data structures significantly influence model performance. We incorporated these new results into the revised manuscript.
>
>
> > Thm 1. An activation function $\sigma$ and its Hermite expansion $\hat{\sigma}$ $\cdots$
>
> We thank the reviewer for the feedback regarding the framing of Theorem 1. Under the assumption of spiked covariance data, Theorem 1 establishes an important equivalence between Random Feature Models (RFMs) for different activation functions under matched statistical moments. However, we would like to clarify that $\hat{\sigma}$ is not simply the Hermite expansion of $\sigma$; rather, $\sigma$ and $\hat{\sigma}$ can be any functions that satisfy our assumptions. We believe that the equivalence of activation functions is broader than just prior equivalence results in the literature, and this generality is significant on its own. Theorem 1 serves as a foundational result that supports the proof of Theorem 2, which addresses equivalence to noisy polynomial models. While we specifically considered polynomial activation functions in Theorem 2, Theorem 1 can be utilized to demonstrate equivalence for a broader range of parametric function families, enhancing its relevance compared to other results.
>
> > Assumption 7. A.7, which is required to keep the perturbed objective convex (line 742)$\cdots$
>
> We appreciate the reviewer’s insightful comment regarding the significance of Assumption A.7 in maintaining the convexity of the perturbed objective. We acknowledge that this assumption is crucial for making claims about generalization error. While similar additional assumptions can be found in the works of Hu and Lu (2023) (e.g., Theorem 1) and Montanari and Saeed (2022) (e.g., Theorem 3), we recognize that Assumption A.7 is indeed nontrivial. To clarify, Assumption A.7 primarily serves a theoretical purpose, specifying a condition about the perturbed objective introduced in equation (23) for proof purposes. It is not directly relevant to our numerical results, which is why we deferred to Appendix A where the perturbed objective is introduced and used for the proof.
>
> > Odd activation functions. Assumption A.6. requires that activation functions are odd. $\cdots$
>
> We appreciate the reviewer’s observations regarding Assumption A.6 and the limited representation of odd activation functions in Figure 3. The reviewer correctly points out that the odd function assumption is used for our argument, particularly around line 1006. The argument breaks down when we attempt to bound the spectral norm of the difference between covariance matrices. This challenge has been noted in prior work, such as Hu and Lu (2023), which also assumes odd activation functions under the isotropic data assumption. Furthermore, Goldt et. al., 2021 (for the case of isotropic data again) introduced a different assumption to relax the requirement for odd activation functions, as detailed in their Example 1. While we maintain the assumption of odd activation functions in our theoretical development for simplicity, we have conducted numerical experiments with both odd and non-odd activation functions. Regarding Figure 3, we did not include tanh because its generalization performance was significantly worse than that of other activation functions in the tested setting.

---

> > ### Author Response · Authors · 2024-11-24
> > **Part 2/2**
> >
> > > As a stylistic comment, Fig. 2a is difficult to read since the noisy linear and random feature models for tanh are overlaid.  Perhaps this could be presented a little differently to make things clearer.
> >
> > Thank you for your valuable feedback regarding Figure 2a. We recognize that the overlay of the noisy linear and random feature models for tanh can make the figure difficult to read. To enhance clarity, we revised Figure 2 by using a bigger marker for the linear model, which improves the readability of the figure.
> >
> > > 1.To what extent would these findings generalise to other anisotropic data settings, and to what extent are they expected to be particular to the spiked covariance model?
> >
> > We hypothesize that our results will extend to various forms of anisotropic data beyond the spiked covariance model. In particular, we plan to investigate scenarios involving more complex alignment and eigenvalue distributions in future work. This exploration will help us better understand the broader applicability of our theoretical insights and their implications for different types of anisotropic data.
> >
> > > 2. Are there any practical implications from this work for model selection, given access to a labelled dataset?
> >
> > Thank you for your question regarding the practical implications of our work for model selection, particularly in the context of having access to a labeled dataset. We believe that our findings can inform model selection by providing insights into the performance differences between Random Feature Models (RFMs) and linear models under various conditions, including input-label correlations. Specifically, our theoretical results suggest that understanding the characteristics of the data—such as its anisotropy and the alignment of features—can guide practitioners in choosing the most appropriate model. For instance, if the data exhibits strong input-label correlation, RFMs may outperform linear models, making them a preferable choice. Please also see our response to "Applicability to real-world data" above for more details.
> >
> >
> > > 3. Given the odd activation function assumption (A.6), do the authors see any empirical difference in results depending on the parity of $\sigma$? As mentioned above, including tanh in Fig. 3 might help answer this question.
> >
> > Thank you for your question regarding the empirical effects of the parity of activation functions. We would like to clarify that Figures 2 and Remark 4 address this issue. Specifically, if the target function is even, using an even activation function in the RFM can lead to better performance compared to using an odd activation function. Conversely, if the target function is odd, an odd activation function would be more suitable. Remark 4 provides a detailed technical explanation of this phenomenon. In simpler terms, if the target function has a bias (i.e., a non-zero mean), it is important to use an even activation function to effectively capture this bias, as our model does not include a learnable bias parameter.
> >
> > > I sincerely thank the authors for their time and efforts. It’s great to see such detailed, careful mathematical work, which is sometimes lacking in the field! $\cdots$
> >
> > Thank you for your kind words and constructive feedback! We appreciate your recognition of our efforts in providing detailed mathematical work. We are committed to addressing your concerns regarding the broader applicability of our findings, particularly in relation to real-world datasets, and we will clarify Assumption A.7 in our revisions. We hope these improvements will enhance the manuscript and meet your expectations. Thank you again for your thoughtful review!

---

> > > ### Comment · Reviewer_vcz1 · 2024-11-25
> > >
> > > Thanks for the detailed response.
> > >
> > > I think the additional experiments on CIFAR-10 -- interpolating between random and 'correlated' labels and showing how this affects RFMs, linear models and polynomial models -- are helpful. Thanks for adding these. It would be nice if this could be related a bit more concretely to properties of the data distribution, but this at least gives the reader a heuristic sense of how your work might apply to real-world data.
> > >
> > > Thanks also for clarifying assumption A.7 and its importance to make theoretical claims about generalisation error. Even if it is only required for proof purposes, I'm still not entirely convinced that it should be deferred to the appendix. The fact that numerical results are still good *despite* this presumably not holding seems important and might be explored more carefully.
> > >
> > > On balance, given the improvements, I'll raise my score from 5 to 6 and recommend acceptance.

---

> > > > ### Author Response · Authors · 2024-11-25
> > > >
> > > > Thank you for your thoughtful and detailed response, and for raising your score. We truly appreciate your feedback regarding the new experiments and the clarification of Assumption A.7. In the camera-ready version (upon acceptance), we will extend our discussions to more concretely relate these findings to the properties of the data distribution. We also appreciate your insights on Assumption A.7 and its role in making theoretical claims about the generalization error. We recognize its importance and will ensure that we explore its implications more thoroughly in the main text. Thank you once again for your constructive feedback.

---

### Official Review · Reviewer_QG5Y · 2024-10-31

**Soundness:** 2
**Presentation:** 3
**Contribution:** 3
**Rating:** 5
**Confidence:** 3

**Summary:**

This paper explores the conditions when and how random feature models (RFM) outperform linear models, focusing on the anisotropic conditions with spiked covariance data with strong input-label correlations. Specifically, they prove that RFM performs equivalent to noisy polynomial models with polynomial degrees influenced by input-label correlation. Numerical simulations are conducted to validate the theoretical results.

**Strengths:**

1. This paper looks good and well-written. Random features model and its connection to neural networks is of interest in machine learning theory.
2. The main contribution is its theoretical aspect, which extends the previous work in [1] to the anisotropic data structure. The universality theorem extends the understanding of RFMs, demonstrating their performance across different activation functions.
3.  Authors suggest that exploiting the non-linearities of RFMs could lead to significant performance gains in high correlation scenarios, which is interesting and confirms practitioners’ intuition.

**Weaknesses:**

1. **Assumptions**: The author only discusses these assumptions from a technical or the math proof aspect, the practical aspects and their limitations should be clarified. And it seems hard to verify these assumptions in practice.
2. **Comparisons with previous work**:  The comparison with previous work is not enough, the proof technique in this paper is based on [1], and more comparisons and detailed differences should be remarked in the paper.
3.  **Limited experiments**:  While the focus of this paper is theory, the experiments in this paper are not adequate, only evaluated in some simulated data, while the real data application with unknown data structure is missing. The detailed setup of the experiment is not clearly presented in this paper, which should be put in a separate section.
3.  **Scope of Application**: While detailed, the focus is specifically on spiked covariance data, which may not generalize to all data structures and might not hold in all real-world scenarios.

**Questions:**

1. While the authors say *"it is noteworthy that while ReLU (9) does not conform to the odd function assumptions stipulated in(A.6)"* in line 208, the simulation uses ReLU as an activation function and gets good results, can you explain it more intuitively?
2. The alignment parameter $\alpha$ seems to be a simple multiplication of two parameters in the structure of x and y, what is the real meaning of the input-label correlation in practice?
3. The main theorem in this paper is only an asymptotic results, i.e., equation (56), can you provide some non-asymptotic results such as the converge speed between the corresponding generalization errors $G_{\sigma}$ and $G_{\hat{\sigma}}$?
4. While this paper is the extension to [1], and [1] considers a general loss function, can the framework from square loss extend to a general loss?

[1]  *Universality Laws for High-Dimensional Learning with Random Features*. (2023), Hu and Lu, TIT.

**Details Of Ethics Concerns:**

None details of concerns beyond what is described above.

---

> ### Author Response · Authors · 2024-11-24
> **Part 1/2**
>
> We appreciate the reviewer's thoughtful feedback and detailed suggestions. We thank the reviewer for recognizing the strengths in our work, particularly regarding its theoretical contributions and relevance to the field of machine learning. We would like to address your concerns in detail:
>
> > Assumptions: The author only discusses these assumptions from a technical $\cdots$
>
> Thank you for emphasizing the need for clarification regarding the practical aspects of our assumptions. We recognize that while most of the assumptions align with practice, the spiked covariance model warrants further explanation. In prior works, such as Hu and Lu (2023), the isotropic data assumption was used, which can be significantly limiting in practical applications. In real-world settings, it is often observed that inputs exhibit an additional low-rank structure, as noted by Facco et al. (2017). To capture this phenomenon, we adopted the spiked covariance model, characterized by a covariance structure that combines an identity matrix with a rank-one spike. This choice allows us to derive theoretical insights that are more applicable to real-world scenarios. To validate our findings, we have conducted additional experiments on the CIFAR-10 dataset, demonstrating the applicability of our results in real-world contexts. You may find the new experimental results in the revised paper.
>
>
> [Facco et al., 2017] Estimating the intrinsic dimension of datasets by a minimal neighborhood information (Nature - Scientific Reports).
>
> > Comparisons with previous work: The comparison with previous work is not enough, $\cdots$
>
> Thank you for your valuable feedback regarding the comparison with previous work, particularly with [Hu and Lu, 2023]. We appreciate the opportunity to clarify this point. While our proof techniques build upon those in [Hu and Lu, 2023], our key contribution is the relaxation of the isotropic data assumption to establish universality for anisotropic data with spiked covariance. Specifically,  the result of [Hu and Lu, 2023] primarily demonstrates the equivalence between Random Feature Models (RFMs) and linear models, whereas our results provide insights into the conditions under which RFMs outperform linear models in anisotropic data settings.
>
> [Hu and Lu 2023], Universality Laws for High-Dimensional Learning with Random Features, IEEE TIT.
>
> > Limited experiments: While the focus of this paper is theory, the experiments in this paper $\cdots$
>
> Thank you for your feedback regarding the need for real data applications in our experiments. To address this concern, we have recently conducted experiments on the CIFAR-10 dataset, focusing on a binary classification task (automobile vs. airplane). In these experiments, we interpolated between random and true labels to demonstrate the practical impact of input-label correlation on our equivalence results. Our findings indicate that the noisy polynomial model performs equivalently to the RFM even in real data scenarios with unknown data structure. Moreover, we observed that the RFM can outperform linear models in cases of strong input-label correlation. We included these experiments in the revised manuscript.
>
> > $\cdots$ The detailed setup of the experiment is not clearly presented in this paper,$\cdots$
>
> Thank you for your suggestion to provide more details about the experimental setup. Given the theoretical focus of our paper, we kept the experimental setup straightforward, with details included in the figure captions. However, we recognize the importance of clarity and will take your feedback into account. Additionally, we will make our code publicly available to facilitate reproducibility and further exploration of our results.

---

> > ### Author Response · Authors · 2024-11-24
> > **Part 2/2**
> >
> > > Scope of Application: While detailed, the focus is specifically on spiked covariance data,$\cdots$
> >
> > Thank you for your insightful comment regarding the scope of our application, particularly concerning the focus on spiked covariance data. While we acknowledge that our findings may not generalize to all data structures, we believe this study serves as a foundational step in understanding the performance of RFMs under anisotropic conditions. Our simplified setting effectively demonstrates how RFMs can outperform linear models in scenarios characterized by strong input-label correlations. This is a significant contribution to the theoretical understanding of generalization errors in nonlinear models, especially since RFMs represent one of the simplest forms of nonlinear modeling. We are optimistic about the potential for future research to extend these findings to more complex data structures, such as Gaussian mixtures and other anisotropic distributions.
> >
> > > While the authors say "it is noteworthy that while ReLU (9) does not conform to the odd $\cdots$"
> >
> > Thank you for your question regarding the odd function assumption in relation to ReLU. ReLU can be decomposed into an even component and an odd component, where the even part behaves like a constant (similar to a bias) in our framework. In our equivalent models, this constant is represented by the additional term $\mu_0 \mathbf{1}$, which captures the effect of the even component. However, incorporating this bias term complicates the proofs, as noted in prior work by Hu and Lu (2023). For this reason, we maintain the odd function assumption for our theoretical development while still utilizing both odd and non-odd activation functions, including ReLU, in our numerical experiments. This approach allows us to explore the practical performance of RFMs without compromising the clarity of our theoretical results. We hope this explanation clarifies the relationship between ReLU and our assumptions.
> >
> > > The alignment parameter $\alpha$ seems to be a simple multiplication of two parameters $\cdots$
> >
> > Thank you for your question regarding the alignment parameter $\alpha$ and its practical implications.  The alignment parameter models the geometric relationship between the structures of the inputs and labels, capturing how well they correspond to each other. This alignment is crucial for characterizing the degree of dependency between the data and labels, which in turn influences model complexity and generalization performance. In practical terms, we can utilize the norm of the input-label correlation as a measure related to the alignment parameter. (Note that input-label correlation is a vector since the input $\mathbf{x}$ is a vector while the label $y$ is scalar.) This measure provides insights into how aligned the features are with the target labels. We have incorporated this concept in our new experimental results on the CIFAR-10 dataset, demonstrating its relevance in real-world applications.
> >
> > > The main theorem in this paper is only an asymptotic results, i.e., equation (56), $\cdots$
> >
> > Thank you for your question regarding non-asymptotic results. While extending our analysis to include convergence speed between the corresponding generalization errors $G_{\sigma}$ and $G_{\hat{\sigma}}$ would require different techniques that are beyond the scope of this submission, we want to clarify that our theoretical results remain valid even for moderate dimensions (on the order of hundreds or thousands), as demonstrated by our numerical results. We appreciate your interest in this aspect of our work and will consider exploring non-asymptotic results in future research.
> >
> > > While this paper is the extension to [1], and [1] considers a general loss function, $\cdots$
> >
> > Thank you for your question regarding the extension of our framework to general loss functions. Our theoretical framework primarily focuses on squared loss due to its analytical tractability. However, we believe that extending these results to broader loss functions is feasible, as discussed in the paper.

---

> > ### Comment · Reviewer_QG5Y · 2024-11-25
> > **Thank you**
> >
> > Thank the authors for their effort and very detailed response to my comments. After going through the other reviewers' comments and the authors' responses, considering the limited novelty compared to the related literature, I maintain my rates.

---

> > > ### Author Response · Authors · 2024-11-25
> > >
> > > Thank you for your response and acknowledgment of our efforts. We are sorry to hear that your judgment about our contributions was affected by other reviewers' misleading comments. Indeed, the misleading comments make indirect connections between our work and prior work from the feature learning literature. One can even understand the difference between our work and the mentioned prior work by reading the abstracts. Our insights (such as the effect of input-label correlation on the (Gaussian) equivalence) DO NOT EXIST in the prior work. Therefore, we believe it is unfair to evaluate the contributions of a work based on indirect connections to prior work.

---

### Official Review · Reviewer_xJdD · 2024-11-04

**Soundness:** 3
**Presentation:** 3
**Contribution:** 3
**Rating:** 6
**Confidence:** 3

**Summary:**

This paper targets to address the question of "When and how does the RFM outperform linear models?". To this end, the paper considers the setting where the data are distributed according to a spike covariance model, which is more general than the isotropic setting existing analyses based on. The author(s) show that RFM can outperform linear models when there is a high correlation between inputs and labels. They also show that RFM is equivalent to noisy polynomial models, where the polynomial degree depends on the input-output correlations.

**Strengths:**

- The paper is clearly written and motivated
- The topic of study is important, since RFMs are closely related to other machine learning models like neural networks.
- The work extents existing studies by performing analysis on a more general data distribution, which is a step toward more general analysis.
- The theoretical analysis is derived in detail.
- The theoretical analysis is verified with numerical experiments

**Weaknesses:**

- Is it unclear how realistic the assumptions in p.4 are. It would be helpful to verify them using real data and compare the assumptions with the ones used in related analyses
- It seems that every single equation in the paper is numbered. It would be more readable to remove the numbers of the unreferenced equations.

**Questions:**

- How realistic is the spike covariance model? Are there any datasets that (approximately) follow such distribution?
- For the assumptions (equation 10) and (equation 11), do existing datasets satisfy them? If so, is there any references or empirical plots? Also, how does these assumptions compared to those of Hu and Lu (2023)'s?
- In (10), is the equation independent of y?
- It is stated in the literature review that RFMs are used in explaining the double descent phenomenon. Can this study on generalization performance provide information about the double descent phenomenon for data drawn from the spike covariance model?

---

> ### Author Response · Authors · 2024-11-24
>
> Thank you for your thorough review and constructive feedback on our paper. We appreciate your recognition of our contributions and the strengths of our work. We would like to address the weaknesses and questions you raised:
>
> > Is it unclear how realistic the assumptions in p.4 are. It would be helpful to verify them using real data and compare the assumptions with the ones used in related analyses
>
> Thank you for raising the important question regarding the practicality of our assumptions.
>
> The assumptions outlined on page 4 are essential for deriving mathematically rigorous results within the spiked covariance model. While they may seem restrictive, they effectively capture key anisotropic properties that are often present in real-world data. The spiked covariance model is widely recognized in high-dimensional statistics and machine learning as a valuable framework for studying structured data.
>
> To address concerns about the realism of these assumptions, we have conducted additional experiments using the CIFAR-10 dataset (specifically, classification between automobiles and airplanes). In these experiments, we interpolated between random and true labels. Our findings demonstrate that while a linear model aligns with the RFM under random labels, a noisy polynomial model becomes necessary to accurately represent the RFM as we approach true labels. These results substantiate the relevance of our assumptions in practical scenarios.
>
> > It seems that every single equation in the paper is numbered. It would be more readable to remove the numbers of the unreferenced equations.
>
> Thanks for your feedback about the equation numbering. We agree that excessive numbering might reduce readability. In the camera-ready version, we will remove the numbering for equations that are not directly referenced, to enhance the presentation.
>
>
> > How realistic is the spike covariance model? Are there any datasets that (approximately) follow such distribution?
>
> Thank you for your question regarding the realism of the spike covariance model. This model is inspired by scenarios where dominant latent factors, such as principal components, significantly influence data in high-dimensional settings. Datasets like CIFAR-10 exhibit structures consistent with this model, as correlations between inputs and labels often stem from a few dominant features. To explore this further, we conducted experiments using CIFAR-10 for binary classification tasks (e.g., distinguishing between automobiles and airplanes), which support our theoretical findings and demonstrate the model's applicability.
>
> > For the assumptions (equation 10) and (equation 11), do existing datasets satisfy them? If so, is there any references or empirical plots?
>
> Thank you for your question regarding the assumptions in Equations (10) and (11) of Theorem 1.  We would like to clarify that these equations represent statistical consistency conditions necessary for establishing equivalence results for random features with different activation functions. They are not designed to model specific datasets directly but rather to outline the conditions under which various activation functions yield similar performance for the random feature model.
>
> > Also, how does these assumptions compared to those of Hu and Lu (2023)'s?
>
> While Hu and Lu's framework assumes isotropic Gaussian inputs, our approach utilizes a spiked covariance model, which introduces structured anisotropy. This choice aligns our data assumptions more closely with real-world datasets, where dominant latent factors often influence the data distribution. The remaining assumptions are similar to those of Hu and Lu (2023), allowing for a meaningful comparison while addressing the complexities of real-world scenarios.
>
> > In (10), is the equation independent of y?
>
> Yes, thank you for pointing this out. In the revised paper, we clarified that $y$ is not included in the expectations in Equation (10).
>
> > It is stated in the literature review that RFMs are used in explaining the double descent phenomenon. Can this study on generalization performance provide information about the double descent phenomenon for data drawn from the spike covariance model?
>
> Our study enhances the understanding of generalization performance in structured data, but it does not directly address the double descent phenomenon. However, the equivalence established between RFMs and polynomial models suggests a potential pathway for analyzing double descent in structured settings. Specifically, the RFM can emulate complex model behavior under certain conditions, which may provide insights into how the double descent phenomenon manifests in data drawn from the spike covariance model. We appreciate your inquiry and will consider discussing this connection further in the revised manuscript.

---

> > ### Comment · Reviewer_xJdD · 2024-11-26
> > **Response**
> >
> > I thank the author(s) for their detailed response and additional experiments. They have addressed my concerns. After reading both the positive and negative comments from other reviewers, I still believe that this work has notable contributions. I will keep my score of 6.

---

> > > ### Author Response · Authors · 2024-11-26
> > >
> > > Thank you for your thoughtful feedback and for recognizing the detailed response and additional experiments we provided. We’re glad to hear that your concerns have been addressed and that you appreciate the contributions of our work, even in light of mixed reviews. Your score of 6 is valued, and we remain committed to improving our research based on insights from all reviewers.

---

### Official Review · Reviewer_RZ3A · 2024-11-04

**Soundness:** 2
**Presentation:** 3
**Contribution:** 4
**Rating:** 6
**Confidence:** 3

**Summary:**

This paper studied whether, and if so under what conditions, the performance of the random feature method (RFM) is better than that of a
 simpler linear model. The authors considered the spiked covariance data model, where the anisotropic characteristics of $x$ relax the previous work based on a more restrictive isotropic data assumption, thereby broadening the applicability of their established results. One important discovery in this paper is that in the case of a strong correlation between inputs and labels, the RFM outperforms the linear model in the sense that the latter has worse generalization performance.

**Strengths:**

This paper solidly verified an interesting phenomenon within the framework of the random feature model that the performance of a learning algorithm has essential dependence on the input or the input-label correlation.  I find it interesting that in the case of a strong correlation between inputs and labels, the RFM outperforms the linear model.

**Weaknesses:**

**Writing:**

 I strongly suggest that the authors include some definitions, symbols, and notations (e.g. the training and generalization errors) in the main text so that the paper can be easily followed. I think that there is significant room for improvement in the paper's arrangement.

**Typographical remarks:**

The first expectation in Eq.(10) should not be taken over $(x,y)$ since the quantity involved does not contain $y$.  Some similar concerns also appear elsewhere.

**Questions:**

I am a bit confused about why the RFM outperforms the linear model in mathematical expression. Could the authors provide a clear comparison of their learning rates in the presence of strong input-label correlation?

---

> ### Author Response · Authors · 2024-11-24
>
> We thank the reviewer for their constructive feedback and the detailed evaluation of our work. We appreciate the reviewer's recognition of our contributions and their suggestion for improving the presentation. Below, we address these points in detail:
>
> > I strongly suggest that the authors include some definitions, symbols, and notations (e.g. the training and generalization errors) in the main text so that the paper can be easily followed. I think that there is significant room for improvement in the paper's arrangement.
>
> Thank you for your suggestions regarding the readability of our paper. We recognize the importance of clear definitions and notations for enhancing comprehension. To clarify, the training and generalization errors are defined in Equations 3 and 4, respectively, and the relevant notations are introduced at the beginning of Section 3. However, we appreciate your feedback and will consider incorporating additional definitions and explanations in the main text to further improve clarity. If you have any further suggestions or specific areas where you feel additional detail would be beneficial, please let us know.
>
> > The first expectation in Eq.(10) should not be taken over $(\mathbf{x}, y)$ since the quantity involved does not contain $y$. Some similar concerns also appear elsewhere.
>
> Thank you for pointing this out. In the revised paper, we clarified that $y$ is not included in the expectations in Equation (10). Also, we clarified similar points appearing elsewhere.
>
> > I am a bit confused about why the RFM outperforms the linear model in mathematical expression. Could the authors provide a clear comparison of their learning rates in the presence of strong input-label correlation?
>
> Thank you for your question regarding the performance comparison between the RFM and the linear model.  The superiority of the RFM stems from its ability to capture higher-order interactions between inputs and labels through nonlinear activation functions. In scenarios with strong input-label correlation, a conventional linear model struggles to leverage these interactions, resulting in suboptimal generalization performance. In contrast, the RFM aligns closely with high-order noisy polynomial models under such conditions, enabling it to outperform the linear model when an appropriate activation function is employed. Regarding your mention of "learning rates," we would like to clarify that our comparison focuses on the generalization errors after complete training rather than on learning rates during training. If you could provide further context on what you mean by learning rates, we would be happy to elaborate on this aspect.

---

> > ### Comment · Reviewer_RZ3A · 2024-11-24
> >
> > Thank you for your response and for bringing up additional insight! I will raise my score to $6$.
> >
> > Regarding the learning rate, I meant the excess risk, a measure of generalization error. To be more clear, the excess risk of any estimator $\hat{f}$ is defined as the prediction error relative to the true target $f^*$, and for the mean regression task, that is $E[ (y-\hat{f}(x))^2- (y- {f^*}(x))^2] = E[ ( \hat{f}(x) - {f^*}(x))^2]$, where $E$ takes expectation over the random pair $(x,y)$ or $x$.
> >
> > I think it could be interesting and helpful, or perhaps more important, to capture the explicit dependence of excess risk for the RFM and linear models on the interaction among sample size and input-label correlation.

---

> > > ### Author Response · Authors · 2024-11-24
> > >
> > > Thank you for your thoughtful response and for raising your score. We appreciate your clarification regarding the learning rate in particular the excess risk as a measure of generalization error. However, notice that in our case, $y$ is equal to the true target; thus, the generalization error is indeed the excess risk. Moreover, while we acknowledge that it could be interesting to develop an explicit characterization of the generalization error, it is outside the scope of the current submission. If you have any further questions or suggestions, we would be happy to discuss them.

---

### Official Review · Reviewer_w9RG · 2024-11-05

**Soundness:** 4
**Presentation:** 4
**Contribution:** 2
**Rating:** 3
**Confidence:** 4

**Summary:**

The paper studies a random features (RF) model trained using ridge regression on a data with a spiked covariance matrix where the covariance is aligned to the target function. Conditions under which such alignment break the gaussian university  (and hence, RF outperforms linear models) is studied.

**Strengths:**

- The paper is well written and well organized.

- Theorem 1 is something new and interesting. Although the proof technique is not novel, and is similar to the approach of Hu and Lu, the such universality across models is not something explicitly studied before (to the best of my knowledge). It can be of independent interest.

**Weaknesses:**

1- Although the analysis of RF models with a spiked covariance assumption for the covariates is new, unlike what is stated in line 143-144, it can still be seen as "feature-learning": One can think of spiked covariance as a model for feature learning in the first layers of a deep neural network where the second-to-last layer is random (i.e., the matrix F) and the last layer is trained with ridge regression (i.e., the vector \omega). In particular, the connections to the following papers should be discussed. These papers study layer-wise updates for a three-layer neural network.

- [R1] Eshaan Nichani, Alex Damian, Jason D. Lee, Provable Guarantees for Nonlinear Feature Learning in Three-Layer Neural Networks.

- [R2] Zihao Wang, Eshaan Nichani, Jason D Lee, Learning hierarchical polynomials with three-layer neural networks

2- Ba et al, (2023); Mousavi-Hosseini et al. (2023) study a problem similar to the one studied here, but for kernel ridge regression instead of random features regression. There needs to be a detailed comparison of the results presented here to the results of these two papers

3- What happens when we set \beta = 1/2 in Assumption A.2? Also, as the paper studies squared losses, there is probably no need to make the odd activation function assumption (A.6); there can be a easier direct proof of universality using Lindeberg exchange without the need to use the results of Hu & Lu.

4- A precise characterization of the training/test errors will improve the quality of this work significantly.

**Questions:**

Please see the weakness section.

---

> ### Author Response · Authors · 2024-11-24
> **Part 1/2**
>
> We thank the reviewer for their detailed feedback and constructive comments on our paper. We appreciate your recognition of the strengths of our paper, particularly regarding the novelty of Theorem 1 and the clarity of our writing. We would like to address your concerns and provide additional context regarding our contributions.
>
> > 1- Although the analysis of RF models with a spiked covariance assumption for the covariates is new, unlike what is stated in line 143-144, it can still be seen as "feature-learning": One can think of spiked covariance as a model for feature learning in the first layers of a deep neural network where the second-to-last layer is random (i.e., the matrix F) and the last layer is trained with ridge regression (i.e., the vector $\omega$). In particular, the connections to the following papers should be discussed. These papers study layer-wise updates for a three-layer neural network.
>
> > [R1] Eshaan Nichani, Alex Damian, Jason D. Lee, Provable Guarantees for Nonlinear Feature Learning in Three-Layer Neural Networks.
>
> > [R2] Zihao Wang, Eshaan Nichani, Jason D Lee, Learning hierarchical polynomials with three-layer neural networks
>
> We appreciate the reviewer’s insightful comments regarding the framing our data assumption with spiked covariance as a form of feature learning. To address the reviewer's feedback, in the revised paper, we included the mentioned papers ([R1] and [R2]) in our discussion about feature learning to provide a more comprehensive context for our work. While we acknowledge this connection from a mathematical perspective, our primary focus is on identifying the conditions under which random feature models (RFMs) outperform linear models.
> Specifically, we demonstrate how strong input-label correlation enables the RFM to achieve superior performance compared to linear models, which is a distinct contribution from the existing feature learning literature. To further emphasize the novel insights of our study, we conducted additional experiments using the CIFAR-10 dataset during the rebuttal period. These experiments illustrate the practical effects of strong input-label correlation on model performance. We present these new experimental results in the revised manuscript.
>
> > 2- Ba et al, (2023); Mousavi-Hosseini et al. (2023) study a problem similar to the one studied here, but for kernel ridge regression instead of random features regression. There needs to be a detailed comparison of the results presented here to the results of these two papers
>
> Thank you for your insightful comments regarding the relationship between our work and the studies by Ba et al. (2023) and Mousavi-Hosseini et al. (2023).  While our data assumptions share similarities with those in the mentioned papers, the focuses of our research are significantly different. Ba et al. and Mousavi-Hosseini et al. primarily investigate sample complexity in the context of kernel ridge regression and neural networks. In contrast, our paper specifically examines Gaussian equivalence for Random Feature Models (RFMs) under anisotropic data conditions, emphasizing how strong input-label correlation allows RFMs to outperform linear models. We recognize that the framework of spiked covariance can be viewed through a lens of feature learning, particularly in relation to deep neural networks. However, our analysis is distinct in that it provides a theoretical foundation for understanding when and why RFMs excel compared to traditional linear models, rather than focusing on feature learning dynamics.

---

> > ### Author Response · Authors · 2024-11-24
> > **Part 2/2**
> >
> > > 3- What happens when we set $\beta = 1/2$ in Assumption A.2?
> >
> > Thank you for your question regarding the assumption on the parameter $\beta$, which governs the scale of spike magnitude $\theta \asymp n^\beta$. In our proofs, we utilize the condition $\beta < 1/2$ to establish asymptotic equivalence between the models.  When setting $\beta = 1/2$, our current proofs yield terms that do not vanish as $n \to \infty$, which presents a limitation of our proof technique, as discussed in the "Discussion of Assumptions." Despite this theoretical constraint, we have observed in our numerical simulations that the results remain valid for $\beta = 1/2$.
> >
> > > $\cdots$ Also, as the paper studies squared losses, there is probably no need to make the odd activation function assumption (A.6); there can be a easier direct proof of universality using Lindeberg exchange without the need to use the results of Hu and Lu.
> >
> >
> > Thank you for your suggestion regarding the relaxation of the odd activation function assumption (A.6). We agree that this assumption could potentially be relaxed due to the properties of squared loss. However, we have chosen to retain it because it simplifies the universality proof using existing techniques. Additionally, when a function is decomposed into its even and odd components, the even part effectively behaves like a constant (similar to a bias) in our context. The additional term $\mu_0 \mathbf{1}$ in the equivalent models captures the influence of the even component, meaning that there is no further insight to be gained from including non-odd parts of the activation functions beyond this bias effect. Therefore, while we maintain the odd function assumption for our theoretical development, we also utilize both odd and non-odd activation functions (e.g., ReLU) in our numerical results. This approach allows us to explore practical implications while keeping our theoretical framework robust.
> >
> > > 4- A precise characterization of the training/test errors will improve the quality of this work significantly.
> >
> > We appreciate the reviewer’s suggestion regarding the precise characterization of training and test errors, and we agree that this would enhance the quality of our work. We would like to clarify that our results can indeed be utilized to derive a more precise characterization of training and test errors. However, this analysis falls outside the primary scope of our current study, which focuses on establishing the asymptotic equivalence of models and identifying the conditions under which the RFM outperforms linear models based on specific data properties.

---

> ### Comment · Reviewer_w9RG · 2024-11-24
>
> I thank the authors for their response. I still believe that there is a huge overlap between the prior work Ba et al, (2023); Mousavi-Hosseini et al. (2023) and your results. From a mathematical perspective, Ba et al, (2023); Mousavi-Hosseini et al. (2023) also study how the gaussian universality breaks as the target-input alignment is increased. (See e.g., Section A.1 in Ba et al. (2023)). The difference is that the prior work consider kernel regression, whereas this paper considers random features regression. The overall message is very similar, so is the proof idea.
>
> Also, mathematically, there is a clear correspondence between result of the paper under review and the result of [R1, R2]. In [R1, R2], a RF model (layer 2 and 3) is trained on top a nonlinear layer (layer 1) which is updated by one step of gradient descent. This updated nonlinear layer (layer 1) is creating target-input alignment for the RF model. They are also showing that the RF model (layers 2 and 3) is beating linear models when the input-target alignment (i.e., the strength of the feature learned by the GD on their 1st layer) is large. In your paper, these correspondences need to be discussed in a lot of detail. At this stage, it is not clear whether the results are novel or they are a special case of the prior work.
>
> As a result, I cannot recommend the acceptance of the current version of the paper.

---

> > ### Author Response · Authors · 2024-11-24
> >
> > Thank you for your response. However, we respectfully disagree with the reviewer's assessment, as we believe it does not accurately reflect the true nature of our work. This misunderstanding may have led to an unfair evaluation of our contributions. We clarify the distinctions between our work and the aforementioned studies as follows.
> >
> > > ... I still believe that there is a huge overlap between the prior work Ba et al, (2023); Mousavi-Hosseini et al. (2023) and your results. ...
> >
> > Ba et al. (2023) studied kernel regression (together with two-layer neural networks) in a spiked covariance setting that **DOES NOT include an alignment parameter** controlling the input-label correlation. Furthermore, Mousavi-Hosseini et al. (2023) only considered two-layer neural networks so it **DOES NOT include kernel regression or random feature models**. However, we study random feature models in a spiked covariance setting with an alignment parameter controlling the input-label correlation. This allows us to illustrate how data properties (e.g., input-label correlation) affect the performance of random feature models in comparison to linear models. Also, the proof techniques are significantly different. For example, we use Lindeberg's method as the main technique in the proof of Theorem 1 while there is no application of Lindeberg's method in the mentioned papers.
> >
> > > ... In [R1, R2], a RF model (layer 2 and 3) is trained on top a nonlinear layer (layer 1) which is updated by one step of gradient descent. This updated nonlinear layer (layer 1) is creating target-input alignment for the RF model. They are also showing that the RF model (layers 2 and 3) is beating linear models when the input-target alignment (i.e., the strength of the feature learned by the GD on their 1st layer) is large ...
> >
> > [R1,R2] studied feature learning in three-layer neural networks in comparison to two-layer networks. Furthermore, **[R1,R2] DOES NOT include (Gaussian) equivalence results**. However, our work provides equivalence results for the random feature model under spiked covariance data. Therefore, research questions and results are significantly different. Finally, we would like to note that the effects of data (e.g., input-label alignment) and the effects of feature learning are two different topics of interest. For the latter, feature learning may introduce some alignment between learned features and targets but this depends on the feature learning algorithm. On the other hand, input-label alignment is an intrinsic property of data that affects the performances of models. To explain practical results, the effects of data should also be studied separately.

---

> ### Comment · Reviewer_w9RG · 2024-11-24
>
> I'm not saying that there are no differences between your result and Ba et al. (2023) and Mousavi-Hosseini et al. (2023). Yes, e.g., for example in Ba et al. (2023) they set (in your notation) $\alpha = 1$. But the extension of their proof to general $\alpha$ is rather straightforward. At this stage, I cannot fully understand what the new insight offered by this paper.
>
> Also regarding the use of Lindeberg exchange arguments: for example Ba et al. (2023) use the very strong results of [EK10] so they find no need to reprove some statements (which in more modern times is often done using Lindeberg exchange).
>
> [EK10] Noureddine El Karoui. The spectrum of kernel random matrices. The Annals of Statistics, 38(1):1–50, 2010.
>
> ---------
>
> Also, it is true that [R1,R2] do not directly mention Gaussian universality, however they indirectly show that this universality is broken because they show that their methods beats kernel/linear methods.

---

> > ### Author Response · Authors · 2024-11-24
> >
> > Thanks for the quick response. We appreciate your recognition of the distinctions stated by us. There are many papers stating that the RFM performs equivalent to linear models while ignoring the data aspect. Our work highlights the importance of data assumptions, which is the new insight. Furthermore, we assert that our recent results with real data mark a pioneering effort in this area, establishing a baseline for future research. We are confident that these contributions will be acknowledged as valuable additions to the existing body of knowledge. Thank you for considering these points as we move forward in the discussion.

---

> > > ### Author Response · Authors · 2024-11-25
> > > **Further Clarification**
> > >
> > > While we have already provided a detailed discussion of the distinctions earlier, presenting such an extensive comparison might inadvertently suggest a stronger basis for comparison than actually exists. We would like to emphasize that the reviewer's comments draw indirect connections between our work and prior studies in the feature learning literature. However, the differences between our work and the cited prior studies are evident, even at the level of their abstracts. Crucially, the novel insights we provide—such as the role of input-label correlation in (Gaussian) equivalence—are entirely absent in the prior work. Consequently, we believe it is not appropriate to assess the contributions of our work based on these indirect connections.

---

> ### Comment · Reviewer_w9RG · 2024-11-26
>
> For example, lemma 6 in  (Ba et al., 2023) demonstrates breaking of universality (a.k.a., gaussian equivalence, etc.) in the exact setup of the paper where the input covariance matrix is spiked. They show that in the case where this spike is present, gaussian universality breaks; i.e., the kernel matrix can no longer be linearized, and depending on the strength of the spike, the kernel matrix can instead be approximated with polynomials of higher degree. The setup is identical with two differences:
>
> 1. in your model, the spike can be made misaligned with the target direction --- however, this is not a major limitation of (Ba et al., 2023), as their proof can be adopted to this setting.
>
> 2. you consider RF instead of kernel regression
>
> For this reason, I don't think this is just "indirect connection". However, not new insight arises with this tweak, and the analysis is fairly similar in nature.
>
> (Ba et al., 2023) Learning in the Presence of Low-dimensional Structure: A Spiked Random Matrix Perspective

---

> > ### Author Response · Authors · 2024-11-27
> >
> > Thank you for engaging in further discussion. We see that you acknowledge the distinctions between our work and Ba et al. (2023), but you argue that the proof in the mentioned paper can be extended to cover our setting (except the difference between kernel regression and random features regression). We understand your concern and address it in detail as follows.
> >
> > ## Clarification on Ba et al. (2023)
> >
> > You mentioned that **Lemma 6** in Ba et al. (2023) demonstrates a breaking of universality in the presence of a spiked covariance matrix, indicating that the kernel matrix cannot be linearized under these conditions.
> >
> > ### Key Distinctions:
> > - **Applicability to Random Features**: While Lemma 6 provides insights relevant to our work, it does not directly apply to random feature models, as their proof techniques are specialized for kernel methods and do not account for the nuances introduced by random features.
> > - **Spike Misalignment**: You correctly note that our model allows for misalignment between the spike and target signals. While it is true that the proof by Ba et al. (2023) can be extended to cover spike-target misalignment, their results do not address it at their current stage.
> >
> > ## On Breaking Universality
> >
> > Regarding your assertion about breaking universality, we interpret your statement as suggesting that if polynomial expansions are necessary, then linear models are insufficient, indicating a breakdown of universality. However, our **Theorem 1** counters this by demonstrating that universality still holds in our setting. We find that only the first two statistical moments of the random features are crucial for performance characterization.
> >
> > ### Our Findings:
> > Our research presents several significant results that advance the understanding of random feature models in the context of spiked covariance:
> >
> > - **Linear Model Approximation**: We demonstrate that the covariance matrix $\mathbb{E}[\sigma(\mathbf{F} \mathbf{x}) \sigma(\mathbf{F} \mathbf{x})^T]$ can still be effectively approximated by a linear model, even when the input $\mathbf{x}$ has spiked covariance. This finding challenges the prevailing notion that spiked covariance inherently disrupts linear approximations.
> >
> > - **Role of Cross-Covariance**: We identify that the cross-covariance $\mathbb{E}[\sigma(\mathbf{F} \mathbf{x}) y]$ plays a crucial role in the emergence of higher-order terms within the equivalent model. This insight reveals a previously unexplored mechanism through which input-label correlations influence model behavior.
> >
> > - **Regimes of Equivalence**: Our analysis delineates specific regimes where the random feature model behaves equivalently to a linear model, as well as conditions under which this equivalence breaks down. We establish an analytical boundary based on spike magnitude and alignment parameters, providing a framework for understanding when and why these models diverge (see Figure 1(a) in our paper).
> >
> > These contributions are novel and fill critical gaps in the Gaussian equivalence literature.
> >
> > ## Technical Challenges in Our Work
> > We recognize your interest in understanding the technical challenges in our work. A significant challenge arises from relaxing the isotropic data assumption utilized by Hu and Lu (2023). In our scenario, we consider inputs distributed as $\mathbf{x} \sim \mathcal{N}(0, \mathbf{I} + \theta \boldsymbol{\gamma} \boldsymbol{\gamma}^T)$. Thus, we can study $\sigma(\mathbf{F} (\mathbf{I} + (\sqrt{\theta + 1} -1)  \boldsymbol{\gamma} \boldsymbol{\gamma}^T) \mathbf{z})$ for $\mathbf{z} \sim \mathcal{N}(0, \mathbf{I})$ instead of $\sigma(\mathbf{F} \mathbf{x})$. However, the spectral norm $ ||\mathbf{F}(\mathbf{I} + (\sqrt{\theta + 1} - 1) \boldsymbol{\gamma} \boldsymbol{\gamma}^T)|| $ diverges as $n \to \infty$. This divergence contrasts sharply with the proof presented by Hu and Lu (2023), where the corresponding spectral norms are assumed to be bounded. This divergence complicates the bounding of terms appearing in the interpolation path when applying Lindeberg's method to prove our Theorem 1.
> >
> > This complexity underscores the unique challenges we face in our work, highlighting the need for careful consideration of these effects in our analysis.
> >
> > ### Addressing Proof Techniques:
> > You suggested that simpler proof methods could have been employed, given our focus on squared loss. While we acknowledge that an easier proof may exist for squared loss case, our objective was to demonstrate results applicable under general loss functions. This necessitated starting with Hu and Lu's proof technique. After establishing key findings related to input-label correlations, we opted to simplify our proofs for squared loss to expedite the sharing of our results with the community.
> >
> > ## Conclusion
> >
> > We genuinely value your comments and hope this response clarifies how our work diverges from the prior work. Your support would mean a lot to us as we seek to advance this area of research. We look forward to your response.

---

### Official Review · Reviewer_ormf · 2024-11-06

**Soundness:** 1
**Presentation:** 2
**Contribution:** 1
**Rating:** 3
**Confidence:** 4

**Summary:**

This work examines Random Feature Models (RFMs) under the assumption of spiked covariance for the input data. The authors establish a Gaussian Equivalence (GE) principle for the errors achieved by RFMs in this setting by revealing an equivalence with noisy polynomial equivalent models. When the alignment between the spike and the target is small, the classical GE principle holds, but for larger alignments, an extended version is required. The analysis is supported by numerical simulations that validate the theoretical findings.

**Strengths:**

The paper is nicely written and the authors provide a nice introduction to the related literature.

**Weaknesses:**

The main weakness for this work is the strong relationship with already publsihed works, namely [Moniri et al. 2024, Cui et al. 2024]. The authors overclaim the depth of their contribution in different parts of the manuscript.

**Questions:**

My main concern for the present submission is the lack of clear elements of novelty that do not meet the high ICLR standards.

As the auhtors correctly report, (Moniri et al. 2024) have already provided the rigorous Random Matrix Theory characterization when the spike appears in the weights of the Random Feature map. These results have been extended up to the maximal learning rate scaling regime by (Cui et al. 2024) which describes the emergence of a fully non-polynomial equivalent feature map in this regime.

I fail to see notable differences between the setting of the present submission and the one in the above-mentioned works.

In different parts of the manuscript, the authors significantly overclaim their contribution. For example, on page 5 "new universality theorem", "this result is notably more general than previous findings". Could the authors clarify what are the novel aspect in their contribution and distinguish them clearly from the results in (Moniri et al. 2024)?

**Details Of Ethics Concerns:**

N/A.

---

> ### Author Response · Authors · 2024-11-24
> **Part 1/2**
>
> We thank the reviewer for their time and feedback. While we appreciate the reviewer's comments, we respectfully disagree with the assessment regarding the novelty of our contributions. We acknowledge the concerns raised and would like to clarify the distinct aspects of our research that set it apart from the existing works. Below, we address these concerns in detail.
>
> > The main weakness for this work is the strong relationship with already publsihed works, namely [Moniri et al. 2024, Cui et al. 2024]. The authors overclaim the depth of their contribution in different parts of the manuscript.
>
> We respectfully disagree with the reviewer's assertion that our work lacks notable novelty compared to the previously published works by Moniri et al. (2024) and Cui et al. (2024). We appreciate the opportunity to clarify the key distinctions between our contributions and those cited:
>
> 1. **Extension to Spiked Covariance Data**
>    While Moniri et al. and Cui et al. primarily focus on isotropic data settings and specific aspects of spiked covariance in first-layer weight structures, our work broadens the analysis to encompass *spiked covariance in input data*. This shift significantly impacts the theoretical foundations of Random Feature Models (RFMs) in real-world scenarios where such input structures are prevalent. To demonstrate how our findings translate to practical applications, we conducted additional experiments on the CIFAR-10 dataset during the rebuttal period, which we detail below.
>
> 2. **Input-Label Correlation**
>    Our research explicitly establishes the critical role of input-label correlation in determining when RFMs outperform linear models. This perspective diverges from the approaches taken by Moniri et al. and Cui et al., which primarily focus on feature learning without considering this specific and practically relevant condition.
>
> 3. **High-Order Noisy Polynomial Models**
>    We provide a novel insight by showing that RFMs require equivalence with *high-order noisy polynomial models* as input-label correlation increases. This result extends prior Gaussian equivalence findings, which have predominantly concentrated on linear regimes (as noted in Hu and Lu, 2023).
>
>
> 4. **Universality Theorem Under Anisotropic Conditions**
>    Our paper introduces a new universality theorem that generalizes previous results to include anisotropic spiked covariance data. This contribution addresses a significant gap in the literature by considering realistic non-isotropic data assumptions, thereby enhancing the applicability of RFMs in practical settings.
>
> In summary, we believe that our work contributes valuable insights into the performance dynamics of RFMs under spiked covariance conditions, particularly regarding input-label correlation and its implications for model superiority.

---

> > ### Author Response · Authors · 2024-11-24
> > **Part 2/2**
> >
> > > As the auhtors correctly report, (Moniri et al. 2024) have already provided the rigorous Random Matrix Theory characterization when the spike appears in the weights of the Random Feature map. These results have been extended up to the maximal learning rate scaling regime by (Cui et al. 2024) which describes the emergence of a fully non-polynomial equivalent feature map in this regime.
> >
> > > I fail to see notable differences between the setting of the present submission and the one in the above-mentioned works.
> >
> > We would like to further clarify the distinctions between our work and those of Moniri et al. (2024) and Cui et al. (2024). While these studies focus on two-layer neural networks operating under isotropic data assumptions, where a spike emerges in the covariance of the first-layer weights after a gradient step, our research shifts the emphasis to the practical performance of Random Feature Models (RFMs) in relation to the properties of input data. Specifically, we investigate how spiked covariance in input data influences RFM performance, particularly under conditions of strong input-label correlation. This nuanced perspective is not addressed in the aforementioned works, which primarily explore feature learning without considering the critical role of input-label correlation in determining model efficacy. Furthermore, we extend the equivalence of RFMs to high-order polynomial models, a significant contribution that enhances our understanding of model performance dynamics. Our new experimental results using a real-world dataset (CIFAR-10) validate our theoretical insights and demonstrate that our findings are applicable beyond synthetic data settings. This comprehensive approach underscores the novelty of our work and its relevance to real-world applications, setting it apart from existing literature.
> >
> > > In different parts of the manuscript, the authors significantly overclaim their contribution. For example, on page 5 "new universality theorem", "this result is notably more general than previous findings". Could the authors clarify what are the novel aspect in their contribution and distinguish them clearly from the results in (Moniri et al. 2024)?
> >
> >
> > We respectfully disagree with the assertion that our work lacks distinctiveness. Our Theorem 1, described as "new and more general," is significant for two main reasons. First, it relaxes the isotropic data assumption established by Hu and Lu (2023), allowing our universality result to apply in a broader context that includes spiked covariance in input data. This is a crucial advancement, as it enhances the applicability of Random Feature Models (RFMs) in real-world scenarios where data often exhibit anisotropic characteristics. Second, we demonstrate the equivalence of RFMs across different activation functions without requiring any specific form, which, to our knowledge, has not been previously explored in the literature. This aspect of our work provides a new perspective on the universality of RFMs, distinguishing it from existing studies that primarily focus on specific architectures or conditions without addressing the performance-wise equivalence of various activation functions.
> >
> > In contrast, Moniri et al. concentrate on replacing the feature matrix with its polynomial equivalent in their analysis, which does not encompass the broader implications of activation function equivalence that we explore. Our findings underscore the critical role of input-label correlation in determining when RFMs outperform linear models—an important nuance that is not addressed by the cited works.
> >
> > To further validate our theoretical insights, we conducted additional experiments using a real-world dataset (CIFAR-10), demonstrating that our results translate effectively to practical applications. This empirical evidence reinforces the relevance and applicability of our contributions beyond synthetic data settings.
> >
> > In summary, we believe that our work offers valuable new insights into the behavior of RFMs under spiked covariance conditions, particularly regarding input-label correlation and its implications for model performance.

---

> > > ### Comment · Reviewer_ormf · 2024-11-26
> > >
> > > I warmly thank the authors for their time in addressing my concerns. However, I would like to maintain my score as I believe that the technical novelty concerning previously published works is still limited.

---

> > > > ### Author Response · Authors · 2024-11-26
> > > >
> > > > Thank you for your response. We respectfully disagree with your assessment that our work lacks novelty.
> > > >
> > > > Your critique appears to be based on an indirect connection between our setting—Random Feature Models (RFMs) under spiked data—and the feature learning scenario involving a single gradient step. While there is an abstract conceptual link, the results in the referenced works (Moniri et al., 2024; Cui et al., 2024) are specifically tailored to feature learning and do not directly apply to our unique setting. Rather than reiterating existing results, our work introduces a new perspective by exploring the implications of anisotropic data in RFMs.
> > > >
> > > > To clarify the novel contributions of our work, we would like to highlight the following key points:
> > > >
> > > > 1. *Universality Theorem under Spiked Data*: We establish a universality theorem for random features of the form $\sigma(\mathbf{F} \mathbf{x})$ when $\mathbf{x} \sim \mathcal{N}(0, \mathbf{I} + \theta \boldsymbol{\gamma} \boldsymbol{\gamma}^\top)$, which is a significant advancement in understanding RFMs.
> > > >
> > > > 2. *High-Order Equivalent Models*: Our theorem providing high-order polynomial models equivalent to the RFM demonstrates how input-label correlation influences the degree of the equivalent model, adding depth to the theoretical framework.
> > > >
> > > > 3. *Effects of Anisotropic Data*: We investigate how anisotropic data affects model behavior, revealing intriguing cases where equivalence between the RFM and noisy linear models persists, which has not been previously explored.
> > > >
> > > > 4. *Beyond Linear Models*: Our extension of analysis to noisy polynomial models and comparison of different activation functions indicates that nonlinearity enhances learning from anisotropic data, marking a departure from traditional linear assumptions.
> > > >
> > > > 5. *Practical Implications*: In response to your feedback, we have included new experimental results on CIFAR-10 that illustrate how input-label correlation affects our equivalence results in practice, further demonstrating the relevance and applicability of our findings.
> > > >
> > > > We believe these advancements provide significant and novel insights into the field and should be considered when assessing the contribution of our work.
> > > >
> > > > We hope that upon reconsideration, you will recognize the novelty and significance of our contributions and agree that our work aligns with the high standards of ICLR.

---

### Author Response · Authors · 2024-11-24

Dear Area Chairs and Reviewers,

Thank you for your detailed feedback on our work. In response to reviewers' feedback, we conducted new experiments on the CIFAR-10 dataset to evaluate the practical relevance of our findings. The new experimental results confirm our claims about the equivalence to the noisy polynomial model and the impact of input-label correlation.

Note that we revised our paper during the rebuttal period and highlighted the revisions in blue.

---

### Author Response · Authors · 2024-12-04
**Summary of Discussion**

Dear Reviewers, AC, and SAC,

We thank you for your thorough evaluation and insightful comments on our manuscript. Below, we summarize the discussion and our responses:

### Key Concerns Raised:
1. **Applicability to Real-World Data**:
   - Most of the reviewers expressed concerns about the practical relevance of our theoretical assumptions, particularly the spiked covariance model.
   - To address this, we conducted additional experiments on CIFAR-10 (binary classification: automobiles vs. airplanes) during the discussion period. These experiments demonstrated that our theoretical findings (e.g., equivalence of RFMs to noisy polynomial models) hold in real-world settings with structured data. These results were included in the revised manuscript.

2. **Novelty Compared to Existing Literature**:
   - Reviewers ormf and w9RG highlighted the need to better differentiate our work from prior studies (e.g., Moniri et al., 2024; Ba et al., 2023).
   - We clarified that our work goes beyond prior studies by focusing on the role of **input-label correlation** and its effect on model performance. We showed that RFMs outperform linear models under conditions of strong input-label correlation, a perspective not covered by the cited studies.
   - We also addressed the mathematical novelty of our approach, particularly the use of Lindeberg’s method to extend Gaussian equivalence results to anisotropic data.

3. **Clarity and Readability**:
   - Reviewers RZ3A and vcz1 suggested improving the presentation, such as including Assumption A.7 in the main text, revising figure layouts, and simplifying notation.
   - In response, we updated figures (e.g., improved visibility in Fig. 2a) and clarified notations. Also, we will discuss Assumption A.7 in the main text in the final version of the paper.

### Remaining Concerns:
- **Reviewer ormf**: Maintained their score due to perceived limited novelty despite detailed clarifications on the distinctions from prior work.
- **Reviewer w9RG**: Argued that the connections to prior kernel regression work reduce the novelty. We respectfully disagreed, emphasizing that our focus on random feature models and input-label correlation is distinct.

### Outcomes:
- Reviewers RZ3A, xJdD, and vcz1 raised their scores after reviewing our responses and additional experiments.
- Despite maintaining their score, Reviewer QG5Y acknowledged the theoretical and practical insights of the work as strengths in their review.
- Reviewers ormf, and w9RG maintained their scores, citing limited novelty despite acknowledging our detailed responses.

### Conclusion:
The discussion reflects recognition of the technical rigor, novel insights, and practical relevance of our work. We believe the revisions and responses have significantly addressed key points.

Thank you for your time and consideration.

Best regards.

---

### Meta-Review · Area_Chair_puMw · 2024-12-20

**Metareview:**

(a) Summary:

The paper explores the conditions under which Random Feature Models (RFMs) outperform linear models in scenarios with spiked covariance data and strong input-label correlation. The authors demonstrate that RFMs can exceed linear models by leveraging this correlation and establish an equivalence between RFMs and noisy polynomial models, with polynomial degree determined by the input-label correlation. They present a universality theorem to support their claims and validate their results with numerical experiments.

(b) Strengths:

The paper addresses an important and timely topic in machine learning theory, providing insights into the advantages of RFMs under structured data conditions.
It introduces a new perspective on how input-label correlation influences model performance, which is a valuable addition to the existing literature.
The use of spiked covariance as a model enriches the theoretical analysis and broadens applicability to realistic datasets.
Numerical simulations are consistent with the theoretical findings.

(c) Weaknesses:

Novelty: Despite the efforts to differentiate the work, the contribution feels incremental relative to prior studies, such as those by Ba et al. (2023) and Mousavi-Hosseini et al. (2023), which addressed similar phenomena in kernel regression contexts. The parallels to existing studies raise questions about the novelty of the results.
Practical Relevance: While theoretical results are well-grounded, the assumptions about spiked covariance data and strong alignment between input and target seem restrictive and may limit real-world applicability.
Empirical Evaluation: The experimental validation lacks breadth, relying heavily on synthetic data and a single dataset (CIFAR-10 binary classification). This undermines the claim of practical relevance.
Clarity and Scope: Reviewers noted room for improvement in presentation, especially in defining key assumptions and notations, and explaining connections to related works.

(d) Decision:

The decision is to reject the paper. While the paper contributes to understanding RFMs, the incremental nature of the contribution and the limited empirical validation weigh against its acceptance. I encourage the authors to address these issues in a future submission by expanding empirical validation to real-world datasets, providing stronger distinctions from related works, and improving the clarity of assumptions and their practical implications.

**Additional Comments On Reviewer Discussion:**

The discussion period addressed several critical points:

Novelty: Reviewers, particularly ormf and w9RG, emphasized the perceived overlap with prior work. The authors responded by clarifying distinctions, especially their focus on input-label correlation. However, these arguments did not fully convince reviewers about the work’s novelty.

Empirical Validation: Concerns about the limited scope of experiments were raised by multiple reviewers. The authors attempted to mitigate this with additional CIFAR-10 experiments, but the overall empirical foundation remains insufficient for a robust validation.

Assumption Realism: Reviewers highlighted the restrictive nature of assumptions. The authors acknowledged this and provided further context and justifications, but practical relevance remains uncertain.

Presentation: Reviewers noted issues in notation and clarity, which the authors promised to address in revisions. These improvements, while helpful, do not resolve the more substantive concerns about the paper's contribution.

The final decision weighed heavily on the consensus regarding limited novelty and scope, despite recognition of the technical rigor and some interesting theoretical insights.

---

### Decision · Program_Chairs · 2025-01-22

Reject